# Rapid binding to protofilament edge sites facilitates tip tracking of EB1 at growing microtubule plus-ends

**Samuel J Gonzalez[1], Julia M Heckel[1], Rebecca R Goldblum[2,3], Taylor A Reid[1], Mark McClellan[1], Melissa K Gardner[1]***

[1]Department of Genetics, Cell Biology, and Development, University of Minnesota, Minneapolis, United States; [2]Department of Biophysics, Molecular Biology, and Biochemistry, University of Minnesota, Minneapolis, United States; [3]Medical Scientist Training Program, University of Minnesota, Minneapolis, United States

**Abstract** EB1 is a key cellular protein that delivers regulatory molecules throughout the cell via the tip-tracking of growing microtubule plus-ends. Thus, it is important to understand the mechanism for how EB1 efficiently tracks growing microtubule plus-ends. It is widely accepted that EB1 binds with higher affinity to GTP-tubulin subunits at the growing microtubule tip, relative to GDP-tubulin along the microtubule length. However, it is unclear whether this difference in affinity alone is sufficient to explain the tip-tracking of EB1 at growing microtubule tips. Previously, we found that EB1 binds to exposed microtubule protofilament-edge sites at a ~70 fold faster rate than to closed-lattice sites, due to diffusional steric hindrance to binding. Thus, we asked whether rapid protofilament-edge binding could contribute to efficient EB1 tip tracking. A computational simulation with differential EB1 on-rates based on closed-lattice or protofilament-edge binding, and with EB1 off-rates that were dependent on the tubulin hydrolysis state, robustly recapitulated experimental EB1 tip tracking. To test this model, we used cell-free biophysical assays, as well as live-cell imaging, in combination with a Designed Ankyrin Repeat Protein (DARPin) that binds exclusively to protofilament-edge sites, and whose binding site partially overlaps with the EB1 binding site. We found that DARPin blocked EB1 protofilament-edge binding, which led to a decrease in EB1 tip tracking on dynamic microtubules. We conclude that rapid EB1 binding to microtubule protofilament-edge sites contributes to robust EB1 tip tracking at the growing microtubule plus-end.

**\*For correspondence:**
klei0091@umn.edu

**Competing interest:** The authors declare that no competing interests exist.

## Editor's evaluation

This paper represents an important study for the microtubule cytoskeleton research community. By employing computational simulation, cell-free biophysical assays, and live-cell imaging, Gonzalez et al. convincingly reveal a mechanistic insight into the EB1 tip-tracking activity at the growing microtubule plus ends, preferential binding of GTP- over GDP-microtubule protofilaments does not fully explain the plus tip tracking of EB1. The authors show a binding preference of EB1 for protofilament edges over the closed lattice, which together with the nucleotide-state dependent dissociation rate of EB1 from the closed lattice successfully recapitulates the efficiency of EB1 tip tracking.

## Introduction

Microtubules are important cellular filaments that are comprised of αβ tubulin heterodimers. The tubulin heterodimers are stacked end-to-end to form structures known as protofilaments, which associate laterally to form the hollow-tube structure of the microtubule (*Mitchison and Kirschner,*

*1984*). The α/β polarity of the tubulin dimer induces microtubule polarity, such that the microtubule end with β-tubulin exposed forms the fast-growing, dynamic 'plus-end' of the microtubule (*Desai and Mitchison, 1997*; *Mitchison and Kirschner, 1984*). In solution, the β-tubulin subunit binds to a GTP nucleotide, which then hydrolyzes to GDP after incorporation into the microtubule lattice. This delayed hydrolysis leads to a high concentration of GTP tubulin at the growing microtubule plus-end, commonly referred to as the 'GTP-cap' (*Desai and Mitchison, 1997*). The presence of the GTP-cap creates a distinct region that is present exclusively at growing microtubule ends (*Maurer et al., 2011*; *Maurer et al., 2014*; *Zanic et al., 2009*).

The localization of proteins along different regions of the microtubule is central to the role of microtubules in cell migration, intracellular transport, and cell division. EB1 is a key cellular protein that autonomously localizes to the growing ends of microtubules ('tip tracks') and recruits other important proteins that have little or no affinity to growing microtubule ends (*Bieling et al., 2007*; *Dixit et al., 2009*; *Morrison et al., 1998*; *Mustyatsa et al., 2017*). It has been shown that improper localization of EB1 at growing microtubule plus-ends can lead to disruptions in both cell division and cell migration (*Dema et al., 2023*; *Dong et al., 2010*; *Honoré et al., 2008*; *Mustyatsa et al., 2017*; *Rogers et al., 2002*; *van Haren et al., 2018*).

EB1 binds a small pocket within the microtubule lattice that is created by four tubulin dimers. It has been shown that EB1 binds with a higher affinity to GTP-tubulin subunits as compared to GDP-tubulin subunits. (*Maurer et al., 2011*; *Maurer et al., 2012*; *Maurer et al., 2014*; *Zanic et al., 2009*; *Zhang et al., 2015*). This difference in affinity likely increases the enrichment of EB1 within the GTP-cap at growing microtubule plus-ends.

Recent work has demonstrated that EB1 can bind to a partial binding pocket composed of 2–3 tubulin subunits, either at the tip of a protofilament, along the side of an exposed protofilament, or at lattice openings within the microtubule (*Reid et al., 2019*). We describe these exposed, partial binding pockets as 'protofilament-edge' sites. Specifically, we use the term 'protofilament-edge' to describe any partial EB1 binding site on the microtubule lattice, as opposed to closed (4-tubulin) binding sites. Importantly, we recently reported that the arrival rate of EB1 to 2-tubulin protofilament-edge sites was ~70 fold faster than to closed 4-tubulin pockets, due to a reduced diffusional steric hindrance to binding (*Reid et al., 2019*). Here, a partial EB1 binding site on the microtubule lattice led to a dramatic reduction in the diffusional steric hindrance that EB1 encounters in order to become properly oriented and then to slide into a closed, 4-tubulin binding pocket. In other words, the expanded physical access that is afforded by EB1 binding to a partial, 2-tubulin binding pocket (as compared to a closed 4-tubulin binding pocket) led to a ~70 fold increase in the EB1 on-rate. Because protofilament-edge sites are present at growing microtubule plus-ends (*Atherton et al., 2018*; *Gudimchuk et al., 2020*; *Guesdon et al., 2016*), we hypothesized that this large difference in EB1 arrival rates could have important repercussions for the efficiency of EB1 tip tracking at growing microtubule plus-ends. We thus predicted that the rapid binding of EB1 to protofilament-edge sites at the growing microtubule plus-end could increase the efficiency of EB1 plus-end tip tracking.

In this work, we generated a single-molecule stochastic simulation that incorporated the assembly and hydrolysis of individual tubulin subunits, as well as the binding and unbinding of EB1 molecules. Importantly, in our simulation, EB1 bound rapidly to protofilament-edge sites, and bound more slowly to closed-lattice sites. In addition, consistent with previous affinity measurements, the off-rate of EB1 from GTP-tubulin sites was low, with higher EB1 off-rates from GDP-tubulin sites. The simulation predicted that rapid binding to protofilament-edge sites increased the efficiency of EB1 tip-tracking at growing microtubule plus-ends. To test this prediction, we used cell-free biophysical assays, as well as live-cell imaging, in combination with a DARPin that binds exclusively to protofilament-edge sites, and whose binding site partially overlaps with the EB1 binding site (*Pecqueur et al., 2012*). We found that DARPin suppressed EB1 protofilament-edge binding on stabilized microtubules, and led to a disruption of EB1 tip tracking on dynamic microtubules plus-ends, both in cell-free experiments and in cells. Together, our work predicts that protofilament-edge binding, along with a differential EB1 binding affinity for GTP vs GDP tubulin, facilitates efficient EB1 tip tracking.

## Results

### A stochastic simulation that simultaneously incorporates tubulin assembly and EB1 on-off dynamics

**Video 1.** Simulated EB1 tip tracking. EB1-GFP in green, Microtubule in red. 2 μm scale bar.
https://elifesciences.org/articles/91719/figures#video1

In previous work, we found that the arrival rate of EB1 to exposed protofilament-edge sites on the sides and/or tips of microtubule protofilaments was ~70 fold faster than to closed four-tubulin pockets, due to a diffusional steric hindrance to binding (*Reid et al., 2019*). To ask whether rapid EB1 protofilament-edge binding could contribute to EB1 tip tracking, we created a stochastic simulation in which there was an increased on-rate of EB1 to protofilament-edge sites relative to closed-lattice sites. This simulation combined the assembly of individual tubulin subunits with EB1 binding and unbinding from the dynamic microtubule.

The microtubule assembly portion of the simulation utilized a previously published model, in which individual tubulin subunits were allowed to arrive and depart from the growing microtubule plus-end (*Margolin et al., 2011*; *Margolin et al., 2012*). Once a tubulin subunit arrived at the growing microtubule plus-end, a longitudinal bond was immediately formed with its penultimate tubulin dimer. Then, lateral bonds were stochastically formed in subsequent time steps (*Margolin et al., 2011*; *Margolin et al., 2012*). Finally, lattice-incorporated GTP-tubulin subunits were stochastically hydrolyzed to GDP-tubulin. In general, the on-rate of new tubulin subunits to the microtubule plus-end depended on the simulated tubulin concentration, and the off-rate of an individual tubulin subunit from the plus-end depended on its hydrolysis state and bonding state, where a GTP-tubulin subunit with two lateral bonds had the lowest off-rate in the simulation. All of the parameter values for the microtubule assembly simulation matched a previously published parameter set (*Margolin et al., 2012*; *Supplementary file 1*), with the exception of (1) the tubulin on-rate constant, which was lowered in order to match our (slow) experimental growth rates, and (2) one additional rule was added to ensure that the tip taper at the microtubule plus-end matched our experimental values (*Figure 1—figure supplement 1A, B*). Here, if the difference between the longest and the penultimate shortest protofilament exceeded 600 nm (75 dimers), the tubulin subunit off-rate and the lateral bond breakage rate were dramatically increased, quickly leading to a catastrophe event.

In addition to tubulin assembly, individual EB1 molecules were allowed to bind and unbind from their binding pockets at any position on the growing microtubule (*Figure 1A*). However, the EB1 on-rates and off-rates depended on the individual binding pocket chemistry and configuration. Specifically, the on-rates for individual EB1 molecules depended on the structure of the binding pocket, such that EB1 arrivals to protofilament-edge sites were substantially faster than to closed-lattice sites, regardless of the hydrolysis state (*Figure 1A*, top, see Methods) (*Reid et al., 2019*). In contrast, the off-rate of EB1 molecules depended on the hydrolysis state of the EB1 binding site. Here, the tubulin subunits towards the minus end of the microtubule dictated the 'hydrolysis state' of the EB1 binding site. If 1–2 of these tubulin subunits were hydrolyzed to GDP, the binding site was considered to be a 'GDP' binding site, leading to an increased EB1 off-rate (*Figure 1A*, bottom, see Methods). All parameter values in the EB1 model were constrained by previously published experimental values (*Maurer et al., 2011*; *Maurer et al., 2014*; *Reid et al., 2019*), with the exception of the EB1 off-rate from protofilament-edge sites, which has not been experimentally measured, but was constrained using bond energy arguments (see *Supplementary file 2*). To evaluate the uncertainty of each model parameter in impacting simulation results, the success of simulated tip tracking was plotted over a broad range of values for each parameter (see *Figure 1—figure supplement 2* and *Figure 1—figure supplement 3*).

### Simulations with rapid binding at protofilament-edge sites can recapitulate EB1 tip tracking

We first asked whether EB1 'tip tracked' growing microtubule plus-ends in the simulation, similar to experimental observations. Qualitatively, our simulated EB1 behaved similarly to experiments – strongly targeting growing microtubule plus-ends, while detaching from shortening ends (*Figure 1B*; *Video 1*). To quantitatively confirm that the simulated EB1 tip tracking was similar to experimental

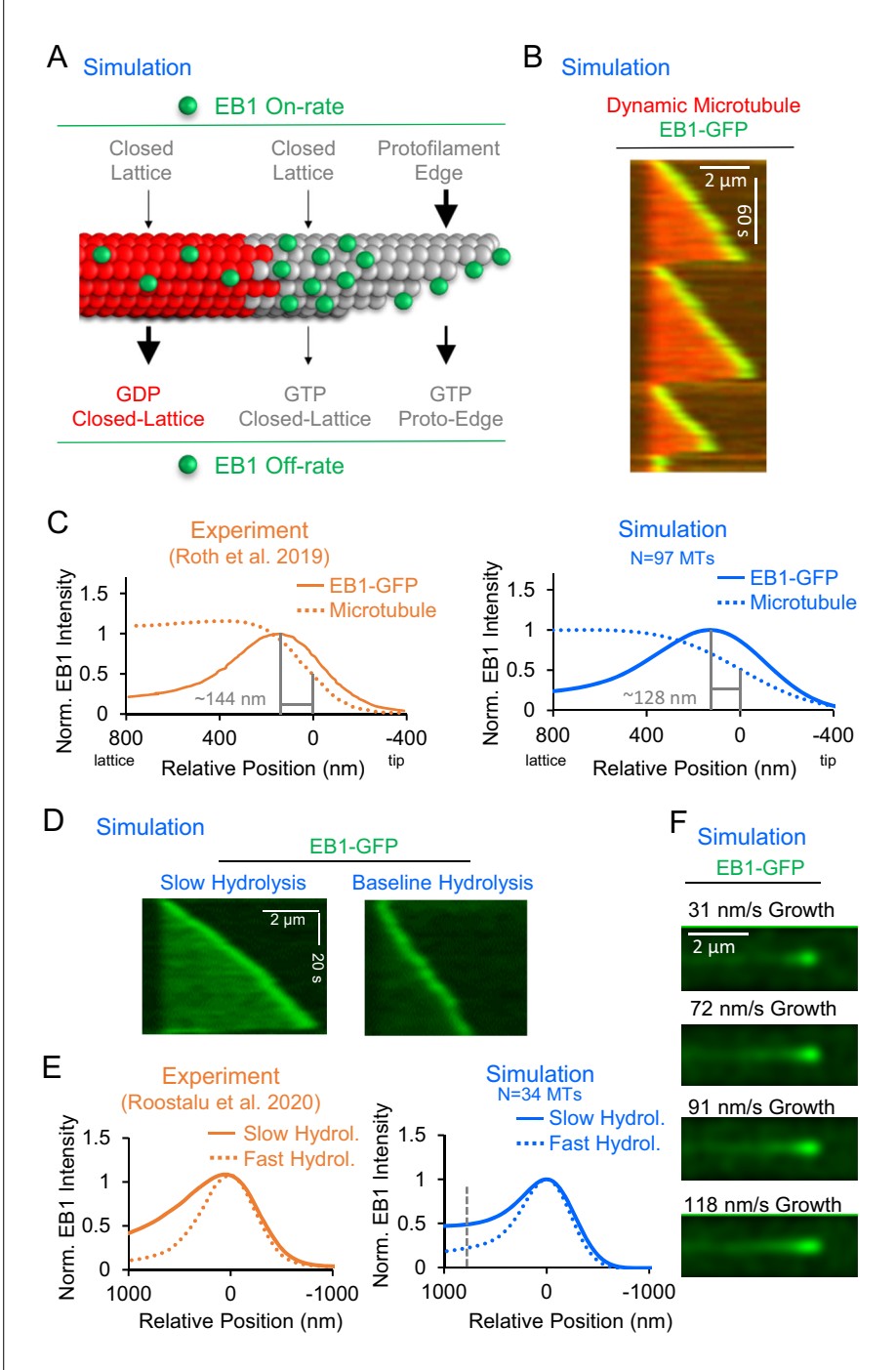

**Figure 1.** Development and validation of a stochastic simulation for EB1 tip tracking. (**A**) Rules for a molecular-scale stochastic simulation that incorporates both tubulin subunit assembly and EB1 arrivals to and departures from the growing microtubule (See Methods). In the simulation, the EB1 protofilament-edge on-rate (top-right) is 50–100 fold higher than the EB1 closed-lattice on-rate (top-left and top-center). The EB1 off-rate is 6–12 fold faster for closed-lattice GDP-tubulin binding pockets (bottom-left) than for closed-lattice GTP-tubulin binding pockets (bottom-center). (**B**) Simulated EB1 tip tracking at growing microtubule ends. (**C**) Left: Line scans of EB1-GFP intensity (solid line), and microtubule intensity (dotted line) from experimentally reported data (**Roth et al., 2019**) (orange). Right: Line scans of EB1-GFP intensity (solid line), and microtubule intensity (dotted line) from the simulation (blue, see Methods). (**D**) Left: Simulated EB1 tip tracking with a slow GTP-tubulin hydrolysis rate (0.05 s$^{-1}$) Right: Simulated EB1 tip tracking with the baseline GTP-tubulin hydrolysis rate (0.55 s$^{-1}$). (**E**) Left: Experimental

*Figure 1 continued on next page*

*Figure 1 continued*

line scan quantification of EB3-GFP intensity along the length of the microtubule with two different hydrolysis rates (orange, *Roostalu et al., 2020*). Right: Simulated data line scan quantification of EB1-GFP intensity along the length of the microtubule with two different hydrolysis rates (blue). Slower hydrolysis leads to a ~ twofold increase in binding along the lattice at 768 nm distal of the peak EB1 position (dashed line). (**F**) Increasing the microtubule growth rate in the simulation increases the EB1 comet length, similar to reports in the literature (*Bieling et al., 2007*).

The online version of this article includes the following source data and figure supplement(s) for figure 1:

**Source data 1.** Line scan data for EB1 intensity for *Figure 1C, E* (right).

**Figure supplement 1.** Additional simulation results.

**Figure supplement 1—source data 1.** Data for panels in *Figure 1—figure supplement 1*.

**Figure supplement 2.** Parameter sensitivity testing for the EB1 tip tracking model I.

**Figure supplement 2—source data 1.** Data for panels in *Figure 1—figure supplement 2*.

**Figure supplement 3.** Parameter sensitivity testing for the EB1 tip tracking model II.

**Figure supplement 3—source data 1.** Data for *Figure 1—figure supplement 3*.

results, we next compared the peak EB1 position from our simulation data to results reported in the literature. The peak EB1 position refers to the distance between the highest EB1 intensity location on the microtubule, and the tip of the growing microtubule plus-end (*Maurer et al., 2012*; *Maurer et al., 2014*; *Nakamura et al., 2012*; *Roth et al., 2019*). At a microtubule growth rate of 10–30 nm/s, the peak EB1 position has been reported to be ~144 nm distal of the microtubule tip (*Figure 1C*, left, orange) (*Roth et al., 2019*). To quantify the peak EB1 position in the simulation, line scans of simulated EB1 comets were obtained and averaged over 97 simulated growth events. We found that the simulation produced a peak EB1 position of ~128 nm distal of the microtubule tip, similar to experimental observations (*Figure 1C*, right, blue). We note that the growth rate and time to catastrophe for the simulated microtubules were similar to experimentally reported values (*Figure 1—figure supplement 1D, E*), and so the simulated peak EB1 position likely reflects an appropriately sized GTP-cap. Importantly, our model with EB1 protofilament-edge binding reproduced the peak EB1 position without requiring a predetermined EB1 'exclusion zone,' as has been previously hypothesized (*Maurer et al., 2014*). Rather, EB1 tip tracking in our current model depended solely on EB1 on/off rates and a growing microtubule plus-end.

To ensure that the configuration of the microtubule plus-end was similar between experiments and simulation, we compared the fitted tip standard deviation in simulated microtubule images to our experimental values. Here, the 'tip standard deviation' reflects the range of protofilament lengths at the tip of the growing microtubule, such that a 'tapered tip' would have a large tip standard deviation (*Coombes et al., 2013*; *Demchouk et al., 2011*). We found that the average tip standard deviation of our simulated microtubules was 191±6 nm (mean ± SEM), similar to our experimental measurements of 180 ± 17 nm (*Figure 1—figure supplement 1A, B*; *Video 2*, mean ± SEM).

It has been previously suggested that growing microtubule plus-ends could be 'flared,' such that they have bent protofilaments that are curved (or flared) away from the central microtubule axis (*McIntosh et al., 2018*). Thus, we asked how a flared microtubule tip structure would affect tip tracking in our simulation. To approximate microtubule tip flaring in the model, we assumed that, with a flared end, all EB1 binding sites in front of the most distal lateral bond would be considered protofilament-edge sites. We found that the microtubule flaring approximation in the simulation had no discernible effect on EB1 tip tracking (*Figure 1—figure supplement 1C*, left/center). Furthermore, we introduced increased tip flaring into the simulation by moving the most distal lateral bond farther away from the growing microtubule tip, which led to increased EB1 targeting to the flared growing microtubule plus-end (*Figure 1—figure supplement 1C*, right). Thus, flared microtubule tips in the simulation behaved similarly to tapered tips, both in EB1 intensity and in peak EB1 location.

It has been shown that a slower GTP-tubulin hydrolysis rate increases EB1 binding along the microtubule, likely due to an increased concentration of GTP-tubulin within the microtubule lattice (*Roostalu et al., 2020*). Thus, to ask whether the simulation could recapitulate this phenomenon, we ran simulations with a slower GTP-tubulin hydrolysis rate (*Figure 1—figure supplement 1F*). We found that a slower hydrolysis rate ($0.05\ s^{-1}$) led to an increased concentration of EB1 on the microtubule,

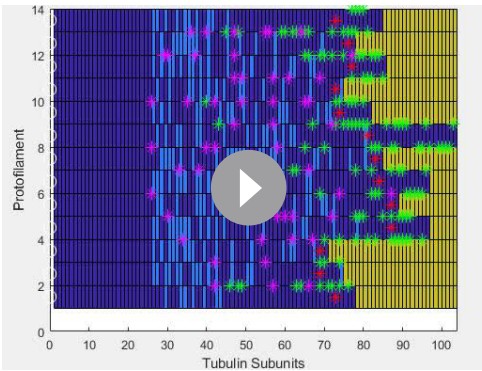

**Video 2.** Animated simulation output for a growth event of one microtubule. Green asterisks: EB1 that originally bound to a protofilament-edge site. Purple asterisks: EB1 that originally bound to a closed-lattice site. Red crosses: location of most distal lateral bond for each protofilament. Protofilament #13 shares a lateral bond with protofilament #1. Dark blue rectangles: GTP-tubulin. Light blue rectangles: GDP-tubulin. A seed of 25 dimers was maintained as GTP-tubulin to represent a GMPCPP seed. The Video is updated every 1000 steps within the simulation.
https://elifesciences.org/articles/91719/figures#video2

**Video 3.** Simulated EB1 tip tracking for different hydrolysis rates. EB1-GFP in green, microtubule in red. 2 μm scale bar. Slower hydrolysis rate on left (0.05 s$^{-1}$), baseline hydrolysis rate on right (0.55 s$^{-1}$).
https://elifesciences.org/articles/91719/figures#video3

as compared to the baseline simulation (0.55 s$^{-1}$) (*Figure 1D*, *Video 3*). By quantifying the localization of EB1 at growing microtubule plus-ends in these simulations, we observed a ~twofold increase in EB1 binding along the lattice of simulated microtubules with a slower hydrolysis rate, relative to the baseline simulation (*Figure 1E*, right, blue; calculated at position 768 nm, gray dashed line), similar to previously reported experimental results (*Figure 1E*, left, orange; *Roostalu et al., 2020*). This result demonstrates that EB1 tip tracking in the simulation depends on the tubulin hydrolysis rate, similar to previous experimental results (*Roostalu et al., 2020*).

Previous work has demonstrated that EB1 monomers tip track less effectively than their dimer counterparts (*Komarova et al., 2009*; *Skube et al., 2010*). In the model, we employed experimentally determined on and off rates for EB1 (see *Supplementary file 2*). Therefore, because the relevant experiments were performed using EB1 in its normal state as a dimer, the baseline simulations represent the simulation results for EB1 dimers. To determine how the model results would be impacted by including monomers in the model, rather than dimers, we turned to previous work, which demonstrated that the EB1 monomer off-rate was ~fourfold larger than the off-rates for dimers (*Song et al., 2020*). Thus, we increased all off-rates in the model by fourfold from their baseline values, and thus ran 'monomer' simulations. We found that EB1 tip tracking was decreased by ~threefold in the monomer simulations (*Figure 1—figure supplement 1G*), consistent with previous reports (*Komarova et al., 2009*; *Skube et al., 2010*).

Finally, it has been widely reported that an increased microtubule growth rate leads to a longer EB1 'comet' (*Farmer et al., 2021*; *Maurer et al., 2014*; *Reid et al., 2019*). Thus, we ran simulations with increasing microtubule growth rates, keeping the GTP-tubulin hydrolysis rate constant. Similar to experimental reports, we found that, as the microtubule growth rate was increased in the simulation, the comet length was increased (*Figure 1F*).

## Protofilament-edge binding increases the efficiency and robustness of tip tracking

We next asked how simulated EB1 tip tracking would be affected if EB1 bound exclusively to the canonical closed-lattice sites on the microtubule. Thus, we set the EB1 protofilament-edge on-rate to zero, and then slowly increased the EB1 closed-lattice on-rate, while leaving all EB1 off-rates constant and at their baseline values (*Figure 2A*, left; *Video 4*; *Supplementary file 2*, see Methods). We found that, while a higher EB1 closed-lattice on-rate led to EB1 accumulation at the growing microtubule end, it also led to EB1 accumulation along the length of the microtubule (*Figure 2A*, right), thus reducing the specificity of EB1 localization to the growing microtubule end.

We next explored the effect on EB1 tip tracking of increasing the EB1 protofilament-edge on-rates. Thus, we set the EB1 closed-lattice on-rate to its baseline (non-zero) value, and then slowly increased the EB1 protofilament-edge on-rate, while leaving all EB1 off-rates constant (*Figure 2B*, left; *Video 5*; *Supplementary file 2*, See Methods). We found that an increasingly intense EB1-GFP puncta appeared

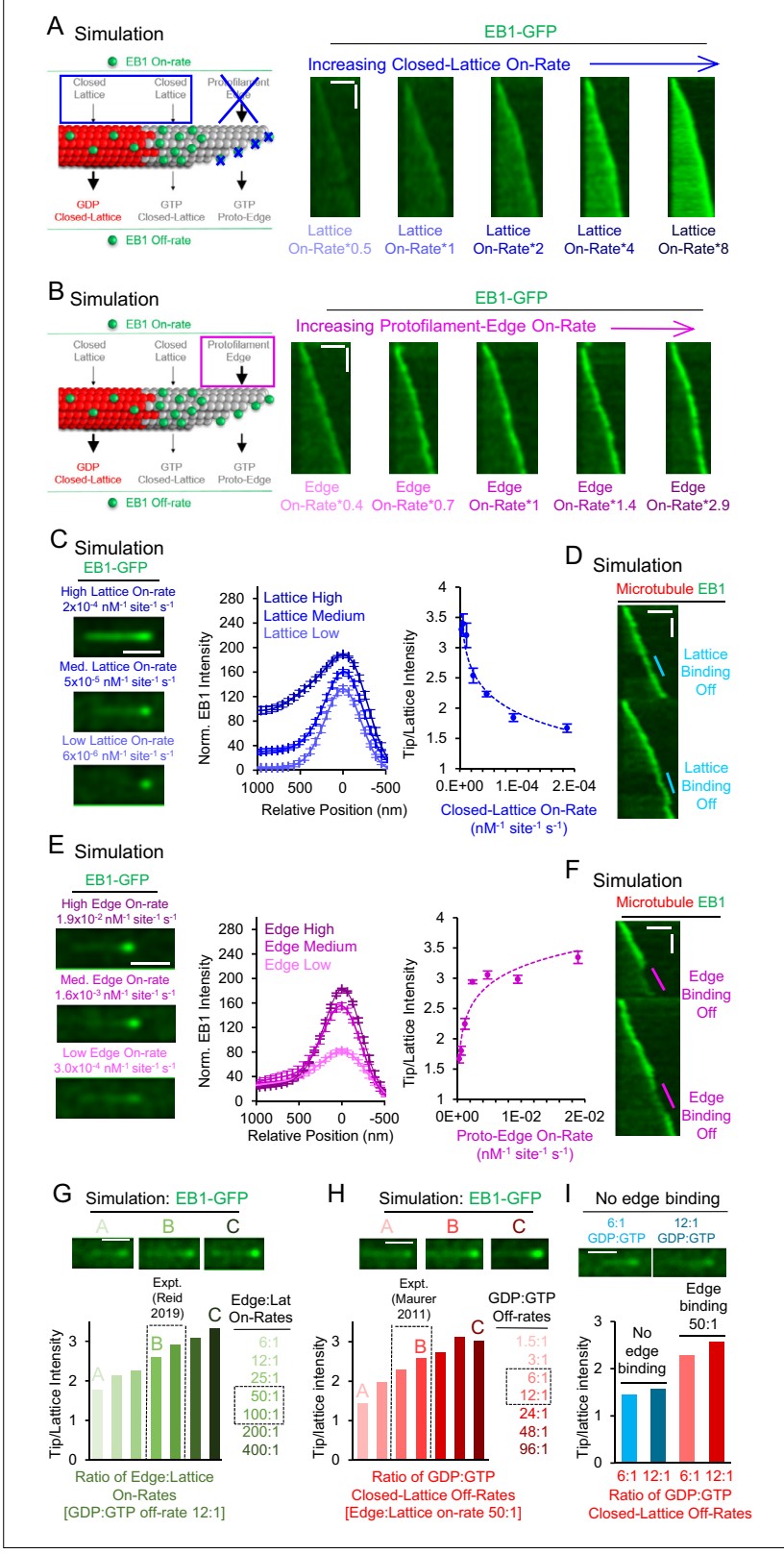

**Figure 2.** Simulations predict that protofilament-edge binding facilitates robust EB1 tip tracking. (**A**) Left: Simulations were performed in which the EB1 protofilament-edge on-rate was set to zero, and the closed-lattice on-rate was gradually increased. Right: Simulated kymographs in which the EB1 protofilament-edge on-rate was set to zero, and the on-rate at closed-lattice sites was gradually increased (scale bars: 2 μm and 10 s). (**B**) Left:

*Figure 2 continued on next page*

*Figure 2 continued*

Simulations were performed in which the closed-lattice on-rate remained constant at its baseline (non-zero) value, and the protofilament-edge on-rate was gradually increased. Right: Simulated kymographs in which the closed-lattice on-rate remained constant at its baseline (non-zero) value, and the protofilament-edge on-rate was gradually increased (scale bars: 2 µm and 10 s). (**C**) Left: Simulated images of EB1-GFP tip tracking over a range of closed-lattice on-rates (scale bar: 1 µm). Center: Line scans from simulated images of EB1-GFP intensity for a range of closed-lattice on-rates (error bars, SEM). Right: Tip:Lattice EB1-GFP intensity ratio vs closed-lattice on-rates in the simulation (error bars, SEM). The tip:Lattice EB1-GFP intensity ratio decreases with increasing closed-lattice on-rates. (**D**) Simulated kymograph in which the closed-lattice on-rate is set to zero partway through the simulation, and later returned to its baseline value (scale bars: 2 µm and 20 s). (**E**) Left: Simulated images of EB1-GFP tip tracking over a range of protofilament-edge on-rates (scale bar: 1 µm). Center: Line scans from simulated images of EB1-GFP intensity for a range of protofilament-edge on-rates (error bars, SEM). Right: Tip:Lattice EB1-GFP intensity ratio vs protofilament-edge on-rates in the simulation (error bars, SEM). Localization to the microtubule tip increases with increasing protofilament-edge on-rates. (**F**) Simulated kymograph in which the protofilament-edge on-rate is set to zero partway through the simulation and later returned to its baseline value (scale bars: 2 µm and 20 s). (**G**) Top: Representative images of EB1-GFP tip tracking for increasing Protofilament-edge:Closed-lattice on-rate ratios. Bottom: Tip:Lattice EB1-GFP intensity ratio for increasing Protofilament-edge:Closed-lattice on-rate ratios (GDP:GTP off-rate ratio is constant and set to 12:1). The experimentally measured Protofilament-edge:Closed-lattice on-rate ratio is 50–100 (*Reid et al., 2019*) (gray dashed boxes). (**H**) Top: Representative images of EB1-GFP tip tracking for increasing GDP:GTP closed-lattice off-rate ratios. Bottom: Tip:Lattice EB1-GFP intensity ratio for increasing GDP:GTP closed-lattice off-rate ratios (Protofilament-edge:Closed-lattice on-rate ratio is constant and set to 50:1). The experimentally measured GDP:GTP closed-lattice off-rate ratio is 6–12 (*Maurer et al., 2011*) (gray dashed boxes). (**I**) Top: Representative images of EB1-GFP tip tracking without protofilament-edge binding. Bottom: Tip:Lattice EB1-GFP intensity ratio for experimentally measured GDP:GTP closed-lattice off-rate ratios, with (red) or without (blue) EB1 binding at protofilament-edge sites.

The online version of this article includes the following source data for figure 2:

**Source data 1.** Data for *Figure 2C, E, G, H, and I*.

at the growing microtubule end as the protofilament-edge on-rate was increased (*Figure 2B*, right; *Video 5*).

## Specificity of EB1 targeting to growing microtubule tips is reduced with higher EB1 closed-lattice binding site on-rates

To quantitatively dissect the relative role of closed-lattice binding on EB1 localization, we ran simulations over a range of EB1 closed-lattice on-rates, while keeping all other EB1 on-rates and off-rates constant and set to their baseline values, including rapid EB1 protofilament-edge binding (*Supplementary file 2*). We found that a low EB1 closed-lattice on-rate led to a clear EB1 puncta at the tip of the microtubule (*Figure 2C*, left-bottom). Here, EB1 accumulation is dominated by protofilament-edge binding. However, increasing the EB1 closed-lattice on-rate by 32-fold led to a ~1.6 fold increase in EB1 intensity at the microtubule tip, but, importantly, also led to a ~25 fold increase in EB1 intensity along the length of the microtubule (*Figure 2C*, center), even in the presence of EB1 protofilament-edge binding. By plotting the ratio of Tip:Lattice EB1 intensity (see Methods), we found that, with increasing EB1 closed-lattice on-rates, the EB1 intensity at the microtubule tip was decreased relative to the lattice (*Figure 2C*, right). Thus, the efficiency of simulated EB1 tip tracking was reduced with faster EB1 binding to closed-lattice sites, due to increased EB1 accumulation along the length of the microtubule.

We then performed a simulation in which the closed-lattice on-rate was set to zero partway through the simulation, to observe in real-time the effect of closed-lattice binding on EB1 tip tracking. We found that EB1 tip tracking was similar whether closed-lattice binding was on or off during the dynamic microtubule simulation (*Figure 2D*, cyan; *Video 6*).

## Simulations with increasing EB1 protofilament-edge on-rates lead to EB1 accumulation exclusively at the growing microtubule plus-end

Next, to quantitatively assess the role of protofilament-edge binding on EB1 localization, we ran simulations over a range of protofilament-edge on-rates, while keeping all other EB1 on-rates and off-rates constant and set at their baseline values, including the closed-lattice on-rate (see Methods). We

**Video 4.** Simulated EB1 tip tracking over a range of EB1 closed-lattice on rates (1.2 x 10⁻⁵ – 1.9 x 10⁻⁴ nM⁻¹ sites ⁻¹ s⁻¹). EB1 protofilament-edge binding on-rate was set to 0. Then, the remaining EB1 on and off rates were kept constant except for the closed-lattice on-rate, which increases from left to right for each subset of the panel. EB1-GFP in green, microtubule in red. 2 µm scale bars.

https://elifesciences.org/articles/91719/figures#video4

**Video 5.** Simulated EB1 tip tracking over a range of protofilament-edge on-rates (5.9 x 10⁻⁴ – 4.7 x 10⁻³ nM⁻¹ sites⁻¹ s⁻¹). All EB1 on-and off-rates were kept constant except for the protofilament-edge on-rate, which increases from left to right for each subset of the panel. EB1-GFP in green, microtubule in red. 2 µm scale bars.

https://elifesciences.org/articles/91719/figures#video5

found that, by decreasing the protofilament-edge on-rate, the intensity of EB1 at the growing microtubule tip was dimmed (*Figure 2E*, left-bottom). Upon increasing the protofilament-edge on-rate, the intensity of EB1 at the growing tip was increased, without an increase in EB1 intensity along the length of the microtubule (*Figure 2E*, left-top). Here, a 32-fold increase in the protofilament-edge on-rate led to a ~2.2 fold increase in EB1 intensity at the tip of the microtubule, and, importantly, no change in the EB1 intensity along the length of the microtubule (*Figure 2E*, center). By plotting the ratio of Tip:Lattice EB1 intensity (see Methods), we found that, with increasing EB1 protofilament-edge on-rates, the EB1 intensity at the microtubule tip was increased relative to the lattice (*Figure 2E*, right). Thus, the efficiency of simulated EB1 tip tracking was enhanced by higher EB1 on-rates to incomplete, protofilament-edge binding sites.

Finally, we performed a simulation in which the protofilament-edge on rate was set to zero partway through a simulation. We found that EB1 tip tracking was rapidly diminished when protofilament-edge binding was shut off during a dynamic microtubule simulation, and returned quickly when the EB1 protofilament-edge on-rate was reset to its baseline value (*Figure 2F*, magenta; *Video 7*).

## Dimensionless variables demonstrate key parameters that control simulated EB1 tip tracking

To quantitatively interrogate the model parameter sensitivity, we defined two key dimensionless variables that control tip tracking in the model. First, as described above, the ratio of the on-rate of EB1 at protofilament-edge sites relative to closed-lattice sites, which is independent of the hydrolysis state of the associated tubulin molecules, directly alters the EB1 tip tracking efficiency in the model (*Figure 2G*). Importantly, distinct tip tracking was observed using the experimentally measured on-rate ratio for protofilament-edge sites relative to closed-lattice sites (50-100:1, *Reid et al., 2019*; *Figure 2G*, image B, gray dashed boxes).

Second, as has been previously described, the ratio of the off-rate of EB1 from closed-lattice GDP-tubulin sites, relative to closed-lattice GTP-tubulin sites, also influenced EB1 tip tracking in the model (*Figure 2H*; note that the model is comparatively insensitive to protofilament-edge off-rates, regardless of hydrolysis state *Figure 1—figure supplement 2G–I*, *Figure 1—figure supplement 3D–F*). Similar to the on-rate ratio, clear tip tracking was observed using the experimentally measured off-rate ratio for GDP-tubulin relative to GTP-tubulin (calculated as 6–12, based on $K_D$ values reported in *Maurer et al., 2011*; *Figure 2H*, image B, gray dashed boxes).

Finally, we evaluated the relative importance of the two dimensionless variables: one that dictates relative EB1 on-rates, and the other that dictates relative EB1 off-rates, in influencing simulated EB1 tip tracking (*Figure 2I*). We found that, in the absence of protofilament-edge binding, the experimentally observed range of closed-lattice GDP:GTP off-rate ratios did not reproduce EB1 tip tracking (*Figure 2I*, top: representative simulated images; bottom: blue bars). However, by including a 50:1 protofilament-edge to closed-lattice on-rate ratio in the simulation, robust tip tracking was reproduced, with an increase in EB1 tip localization for a higher ratio of GDP:GTP off-rates (*Figure 2I*, red). Thus, based on the experimentally measured EB1 on and off rates, both a hydrolysis-state dependent

**Video 6.** Simulated EB1 tip tracking, with EB1 closed-lattice binding dynamically set to 0 during the run. Simulation is run with all baseline parameter values (***Supplementary file 2***). At times 57 s and 209 s, EB1 closed-lattice on-rate is set to zero. Then, at times 87 s and 239 s, the EB1 closed-lattice on-rate is reset to its baseline value. EB1-GFP in green, microtubule in red. 2 μm scale bar.

https://elifesciences.org/articles/91719/figures#video6

**Video 7.** Simulated EB1 tip tracking, with EB1 protofilament-edge binding dynamically set to 0 during the run. Simulation is run with all baseline parameter values (***Supplementary file 2***). At times 56 s and 203 s, EB1 protofilament-edge on-rate is set to zero. Then, at times 85 s and 233 s, the EB1 protofilament-edge on-rate is reset to its baseline value. EB1-GFP in green, microtubule in red. 2 μm scale bar.

https://elifesciences.org/articles/91719/figures#video7

EB1 off-rate, as well as a rapid protofilament-edge EB1 on-rate, were critical to reproduce EB1 tip tracking in the model.

## Split EB1 comets have increased EB1 binding relative to single EB1 comets

It has been previously reported that EB1-GFP can split into multiple comets that track the growing microtubule end (***Doodhi et al., 2016***). Thus, a 'split comet' refers to the phenomenon in which there are two or more distinct EB1 puncta that track a growing microtubule end (***Doodhi et al., 2016***; ***Farmer et al., 2021***). A split comet likely occurs when one or more protofilaments lag behind the growing microtubule tip, thus producing an extended, highly tapered tip (***Figure 3A***, left (gray)). In the canonical model in which tip tracking relies exclusively on a higher EB1 affinity for GTP-tubulin relative to GDP-tubulin, it is expected that, for a single microtubule growth event with a constant growth rate (and thus a constant total GTP-cap size), the total summed intensity of EB1-GFP at split-comet tips would be similar to the intensity of EB1-GFP at single-comet tips. However, in a model with preferential EB1 binding to protofilament-edge sites, we predicted that the additional protofilament-edge binding sites on the sides of exposed protofilaments, afforded by a large difference in proto-filament lengths at the tip of growing microtubules with 'split comets,' would lead to a net increase in the summed intensity of EB1-GFP (***Figure 3A***, left) (***Farmer et al., 2021***). Thus, if EB1 binds to protofilament-edge sites, we predicted that there would be an increase in the summed EB1-GFP intensity at growing microtubule tips with split comets, due to the increased number of protofilament-edge sites that are available to recruit EB1.

We first tested this prediction using our simulation. Thus, we asked whether there was an increase in the summed EB1-GFP intensity at growing microtubule tips with split comets. To generate split comets in the simulation, we altered the microtubule assembly simulation parameters to allow for an increase in taper at the growing microtubule tips (from ≤~600 nm in our standard simulation, to ≤~3 μm in the split comet simulation (see Methods)). By increasing the taper at the microtubule tip, the simulation was able to recapitulate split comets (***Figure 3A***, right (orange arrow: pre-split; cyan arrows: post-split)). We then asked whether there was an increase in the summed EB1-GFP intensity on individual growing microtubule tips after an EB1 comet split, relative to prior to the split. Thus, we measured the total intensity of EB1-GFP both before and after the comet split on individual simulated growing microtubules (***Figure 3B***, top: pre-split; middle: post-split). We subtracted the green background intensity both before and after the comet split (***Figure 3B***, bottom). We found that the split comets had a~40% increase in the summed intensity of EB1-GFP at the growing tip, relative to single comets on the same growth events (***Figure 3C***, p<0.001, paired t-test,). Therefore, consistent with our prediction, the simulation data indicates that an increase in protofilament-edge sites on the sides of exposed protoflaments during split-comet growth events leads to an increase in EB1 recruitment to the microtubule plus-end.

Next, to test this prediction experimentally, we examined experimental microtubule growth events with split comets (***Figure 3D***, right; orange arrow: pre-split; cyan arrows: post-split). We measured the summed Mal3-mCherry (yeast EB1-homolog) intensity both before and after the comet split on individual growing microtubules (***Figure 3E***; top: pre-split; middle: post-split). We subtracted the green background intensity both before and after the comet split (***Figure 3E***, bottom). We found that split

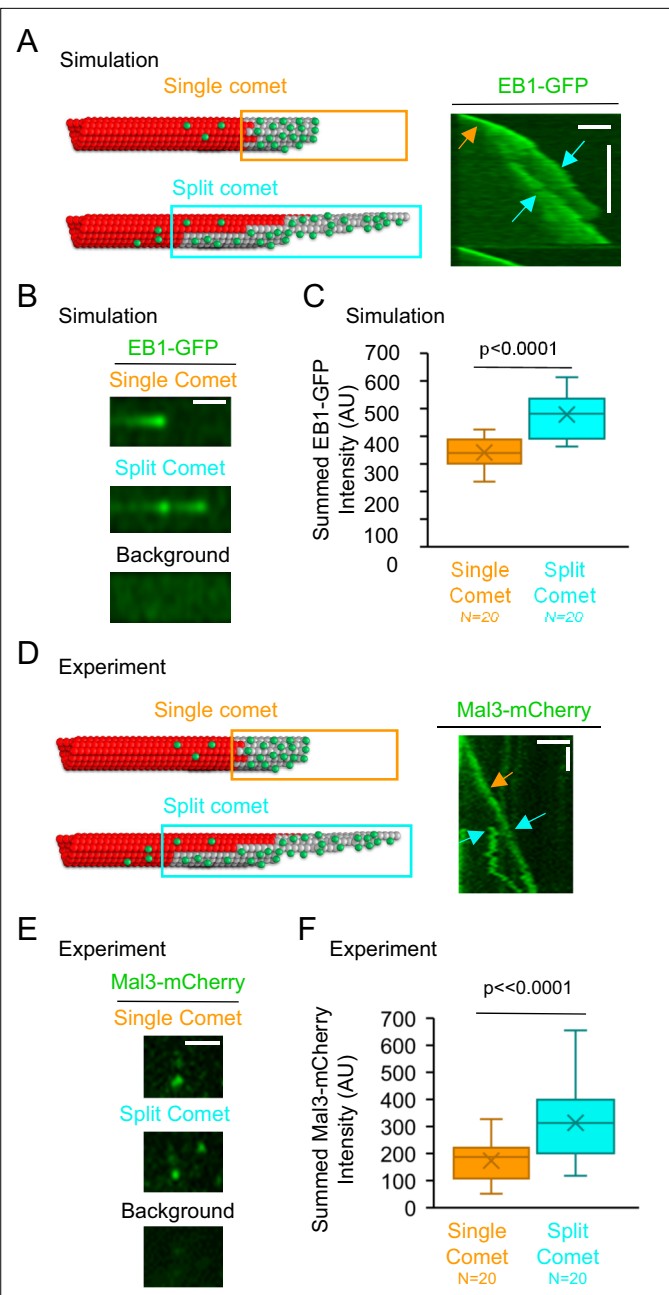

**Figure 3.** Summed Mal3 comet intensity is increased in split comets relative to single comets. (**A**) Left: Schematic of a single comet (top), and a split comet (bottom). Right: Simulated kymograph with a single comet (orange arrow), and split comet (cyan arrows), within one common microtubule growth event (scale bars: 2 μm and 60 s). (**B**) The summed comet intensity can be measured for a single simulated comet (top) and, later on, the same simulated microtubule growth event, for a split simulated comet (middle). The background (bottom) was subtracted from both the single comet and split comet summed intensity measurements. (**C**) Simulated split comets have a higher summed EB1 intensity at the microtubule tip than their single comet counterparts on the same growth event (p<0.0001, paired t-test,). (**D**) Left: Schematic of a single comet (top), and a split comet (bottom). Right: Typical experimental kymograph with a single comet (orange arrow), and split comet (cyan arrows), within one common microtubule growth event (scale bars: 2 μm and 60 s). (**E**) The summed comet intensity can be measured for a single comet (top) and, later on, the same microtubule growth event, for a split comet (middle). The background (bottom) was subtracted from both the single comet and split comet summed intensity measurements. (**F**) Split comets had a higher summed Mal3-mCherry intensity at the microtubule tip than their single comet counterparts from the same growth event (p<0.0001, paired t-test).

*Figure 3 continued on next page*

*Figure 3 continued*

The online version of this article includes the following source data for figure 3:

**Source data 1.** Data for *Figure 3C, F*.

comets had an ~80% increase in the summed intensity of Mal3 at the growing microtubule tip relative to the single comets on the same microtubule growth events (*Figure 3F*, p<0.0001, paired t-test,). Thus, the experimental results are consistent with the simulation results, and suggest that an increase in protofilament-edge sites on the sides of exposed protofilaments during split-comet growth events leads to an increase in EB1 recruitment to the microtubule plus-end.

## DARPin suppresses EB1 binding to protofilament-edge sites on stabilized microtubules

Because the simulation predicted that protofilament-edge binding is integral to EB1 tip tracking, we reasoned that EB1 tip tracking would be disrupted by a protein that could block EB1 binding to protofilament-edge sites. Thus, to test this prediction, we leveraged a DARPin D1 that binds exclusively to protofilament-edge sites, and also partially overlaps with the EB1 binding site on microtubules (*Figure 4A*; *Video 8*; *Pecqueur et al., 2012*). Here, DARPin could bind to protofilament-edge sites, and thus suppress EB1 binding in these locations, which would, in turn, disrupt proper EB1 tip tracking.

We first asked whether DARPin could block EB1 binding at protofilament-edge sites. Thus, we generated stabilized GTP-analogue (GMPCPP) microtubules that were damaged, such that portions of the microtubule contained openings and defects. Damaging the microtubules leads to an increased number of protofilament-edge sites along the microtubule length (*Coombes et al., 2016*; *Gupta et al., 2013*; *Reid et al., 2017*). Damaged microtubules can be generated by briefly exposing stabilized GMPCPP microtubules to $CaCl_2$ (*Coombes et al., 2016*; *Gupta et al., 2013*; *Reid et al., 2017*). Thus, coverslip-adhered, rhodamine-labeled GMPCPP microtubules were briefly incubated in 10 mM $CaCl_2$, followed by a wash to remove the $CaCl_2$ (*Figure 4B*, left). Then, the damaged microtubules were incubated with the yeast EB1 homolog Mal3-GFP, in the absence or presence of DARPin (*Figure 4B*). Total Internal Reflection Fluorescence (TIRF) microscopy was used to visualize the microtubules, and Mal3 binding to the microtubules was assessed. Qualitatively, we observed a reduction in Mal3-GFP binding to the damaged microtubules in the presence of DARPin, as compared to the no-DARPin controls (*Figure 4C*, top). By using a custom MATLAB script to measure the Mal3-GFP binding area on the damaged microtubules (*Reid et al., 2017*), we found that the fraction of microtubule area bound by Mal3-GFP was ~2.7 fold lower in the presence of DARPin as compared to the no-drug controls (*Figure 4C* bottom-left, and *Figure 4—figure supplement 1A*; p<0.001, t-test). In contrast, by using undamaged GMPCPP microtubules, which did not have openings and defects to generate protofilament-edge binding sites along the microtubule length, there was no significant difference in Mal3-GFP binding in the presence and absence of DARPin (microtubules not treated with $CaCl_2$; *Figure 4C* bottom-right, and *Figure 4—figure supplement 1B*; p=0.58, t-test). These results suggest that DARPin acts to block Mal3-GFP binding specifically on protofilament-edge sites.

## In cell-free experiments, DARPin suppresses Mal3 tip tracking on dynamic microtubule plus-ends

We next asked whether suppression of protofilament-edge binding would disrupt EB1 tip tracking. First, we ran simulations to quantitatively predict how EB1 tip tracking would be altered by suppressing its protofilament-edge on-rate (*Figure 5A*, left). Thus, we gradually reduced the protofilament-edge on-rate and generated simulated images to detect the relative localization of EB1-GFP at growing microtubule plus-ends (*Figure 5A*, center). To evaluate EB1-GFP localization to growing microtubule plus-ends in the simulation, we measured the EB1-GFP 'Tip Specificity.' Here, we defined Tip Specificity ($S$) as:

$$S = \frac{\left(I_{tip} - I_{background}\right)}{\left(I_{lattice} - I_{background}\right)} \tag{1}$$

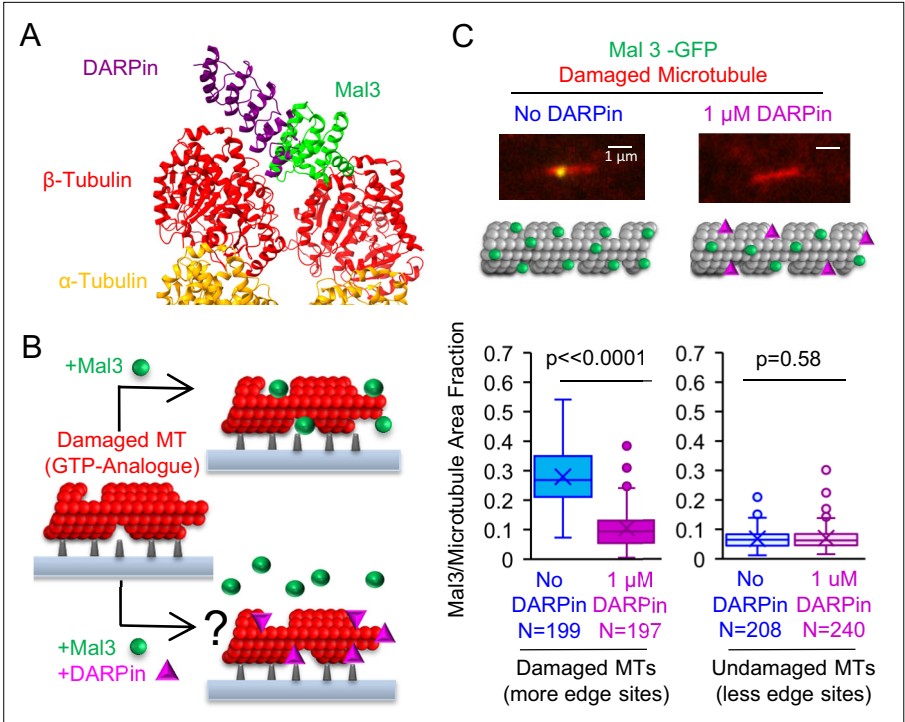

**Figure 4.** Synthetic Designed Ankyrin Repeat Protein (DARPin) peptide blocks EB binding to protofilament-edge sites. (**A**) Structure of DARPin and Mal3, together with α and β tubulin at a microtubule tip was created in ChimeraX (*Pettersen et al., 2021*) using the crystal structures 4drx (DARPin) and 4abo (Mal3). (**B**) Left: To test whether DARPin blocks EB binding to protofilament-edge sites, we first generated stabilized GTP-analogue (GMPCPP) microtubules that were damaged with CaCl$_2$ treatment, thus creating protofilament-edge sites along the length of the microtubule. Right: The damaged microtubules were incubated with Mal3-GFP in the presence (bottom) or absence (top) of 1 μM DARPin. Suppression of Mal3-GFP binding to the damaged microtubule in the presence of DARPin suggests that DARPin blocks Mal3 from binding to protofilament-edge sites (right, bottom). (**C**) Top: Representative images of Mal3-GFP binding to damaged microtubules, in the absence (blue) or presence (magenta) of 1 μM DARPin. Middle: Cartoons depicting the relative binding of Mal3 and DARPin in each experiment. Bottom-Left: Quantification of the fraction of microtubule area bound by Mal3-GFP for damaged microtubules in the absence or presence of DARPin (p<0.0001, t-test; sample size indicates number of images). Bottom-Right: Quantification of the fraction of microtubule area bound by Mal3-GFP for undamaged microtubules in the absence or presence of DARPin (p=0.58, t-test; sample size indicates number of images).

The online version of this article includes the following source data and figure supplement(s) for figure 4:

**Source data 1.** Data for *Figure 4C*.

**Figure supplement 1.** Additional experimental data for binding of Mal3-GFP to stabilized GMPCPP microtubules.

**Figure supplement 1—source data 1.** Data for *Figure 4—figure supplement 1*.

---

Where $I_{tip}$ is the EB1 intensity at the growing microtubule tip, $I_{lattice}$ is the EB1 intensity on the microtubule lattice, and $I_{background}$ is the background EB1 intensity just outside of the growing microtubule tip. By definition, a lower Tip Specificity value indicates that there is less efficient tip tracking. In addition, a Tip Specificity value equal to one (e.g. $S=1$) means that the EB1 intensity at the growing microtubule tip is equal to the EB1 intensity along the length of the microtubule, and therefore EB1 is not tip tracking. We found that, in the simulation, a decreased protofilament-edge on-rate led to a decrease in Tip Specificity (*Figure 5A*, right inset: y-axis is absolute Tip Specificity). Specifically, a ~twofold reduction in protofilament-edge on-rate led to a ~25% reduction in Tip Specificity (*Figure 5A*, right, large plot y-axis shows fold-change in tip specificity; gray dotted lines is ~25% reduction in tip specificity).

Thus, to test this simulation prediction, we performed a cell-free assay in which dynamic microtubules were grown from stabilized seed templates in the presence of Mal3-mCherry (*Figure 5B*, left). We visualized the growing microtubules using TIRF microscopy, in the presence of increasing

concentrations of DARPin (*Figure 5B*, center). We found that Mal3 tip tracking was increasingly disrupted as the DARPin concentration was increased (p<0.001, Kruskal Wallis) (*Figure 5B*, right inset: y-axis is absolute Tip Specificity). Interestingly, 1 µM DARPin led to a~25% reduction in Tip Specificity, consistent with the simulation prediction of a twofold reduction in protofilament-edge on-rate (*Figure 5B*, right, large plot y-axis shows fold-change in tip specificity; gray dotted lines is ~25% reduction in tip specificity).

We then asked whether the suppression of tip tracking in DARPin could be due to a drop in microtubule growth rate, leading to a reduced concentration of GTP-tubulin at the growing microtubule plus-end (*Farmer et al., 2021*; *Maurer et al., 2014*; *Reid et al., 2019*). We found that the suppression of tip tracking was more substantial than would be predicted based on the small changes in microtubule growth rate at 1 µM DARPin (*Figure 5C*, left, blue dotted line: control; purple: 1 µM DARPin; p<0.001, Kruskall Wallis comparing DARPin data to Control data at ~0.012 um/s growth rate). Furthermore, we found no significant increase in the time to catastrophe with increasing DARPin concentrations, suggesting that DARPin does not affect the GTP hydrolysis rate or the associated GTP-cap size (*Figure 5C*, center; p=0.09–0.4, Tukey's post-hoc test).

Finally, we asked whether DARPin could indirectly disrupt Mal3 tip tracking by altering the configuration of the growing microtubule plus-end. Here, a more blunt microtubule tip structure could reduce the number of available protofilament-edge sites, and thus indirectly disrupt tip tracking. In contrast, a more extended, tapered tip structure would naturally allow for increased numbers of protofilament-edge sites, similar to the split comet phenotype as described above (*Figure 3A*), which increased Mal3 targeting to the growing microtubule tip. We found that 1 µM DARPin led to a ~40% increase in tip tapering at the growing microtubule end, which reflects a moderate increase in available protofilament-edge sites (*Figure 5C*, right; *Coombes et al., 2013*; *Demchouk et al., 2011*). However, despite the increased availability of protofilament-edge sites, tip tracking was suppressed in DARPin (*Figure 5B*). Thus, DARPin does not suppress Mal3 tip tracking by indirectly reducing the number of available protofilament-edge sites. Rather, Mal3 is likely excluded from the protofilament-edge sites that are occupied by DARPin, which in turn suppresses tip tracking.

## DARPin suppresses EB1 tip tracking on growing microtubules in LLC-Pk1 cells

Finally, we asked whether DARPin could block EB1 binding to protofilament-edge sites, and thus suppress EB1 tip tracking, inside of cells. Thus, we cloned the DARPin sequence into a vector with an N-terminal Turbo RFP followed by a self-cleaving P2A peptide, which allowed us to examine cells for RFP expression to detect successful plasmid transfection into the cell, while at the same time allowing DARPin to function in its native, unlabeled form.

To first determine whether the DARPin protein could bind microtubule protofilament-edge sites in cells, we transfected the RFP-P2A-DARPin construct into LLC-Pk1 cells that expressed Tubulin-GFP (*Rusan et al., 2001*). Here, we reasoned that, if the expressed DARPin protein was binding protofilament-edge sites, a high concentration of DARPin could potentially suppress new tubulin subunit binding to growing microtubule plus-ends, and thus reduce the overall microtubule density in the transfected cells. Indeed, in comparing the microtubule density in cells that were not transfected (as identified by a lack of red fluorescence, *Figure 6A*, left), to transfected cells (with red fluorescence, *Figure 6A*, right), we observed a reduction in microtubule density in the DARPin-transfected cells (*Figure 6B*; p=0.017, Mann-Whitney U Test).

We then transfected the RFP-P2A-DARPin construct into LLC-Pk1 cells that overexpressed EB1-GFP, to examine the effect of the DARPin on EB1 tip tracking (*Piehl et al., 2004*). We observed fewer EB1 comets in the presence of DARPin, as would be expected due to a reduction in the microtubule network density (*Figure 6B*). However, a sufficient number of EB1 comets were

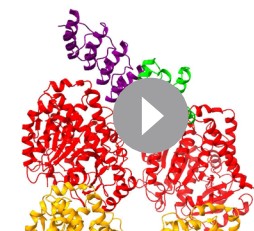

**Video 8.** Mal3 and Designed Ankyrin Repeat Protein (DARPin) overlap along an exposed tubulin plus-end. Created in ChimeraX using the structures 4DRX (DARPin) and 4ABO (Mal3).
https://elifesciences.org/articles/91719/figures#video8

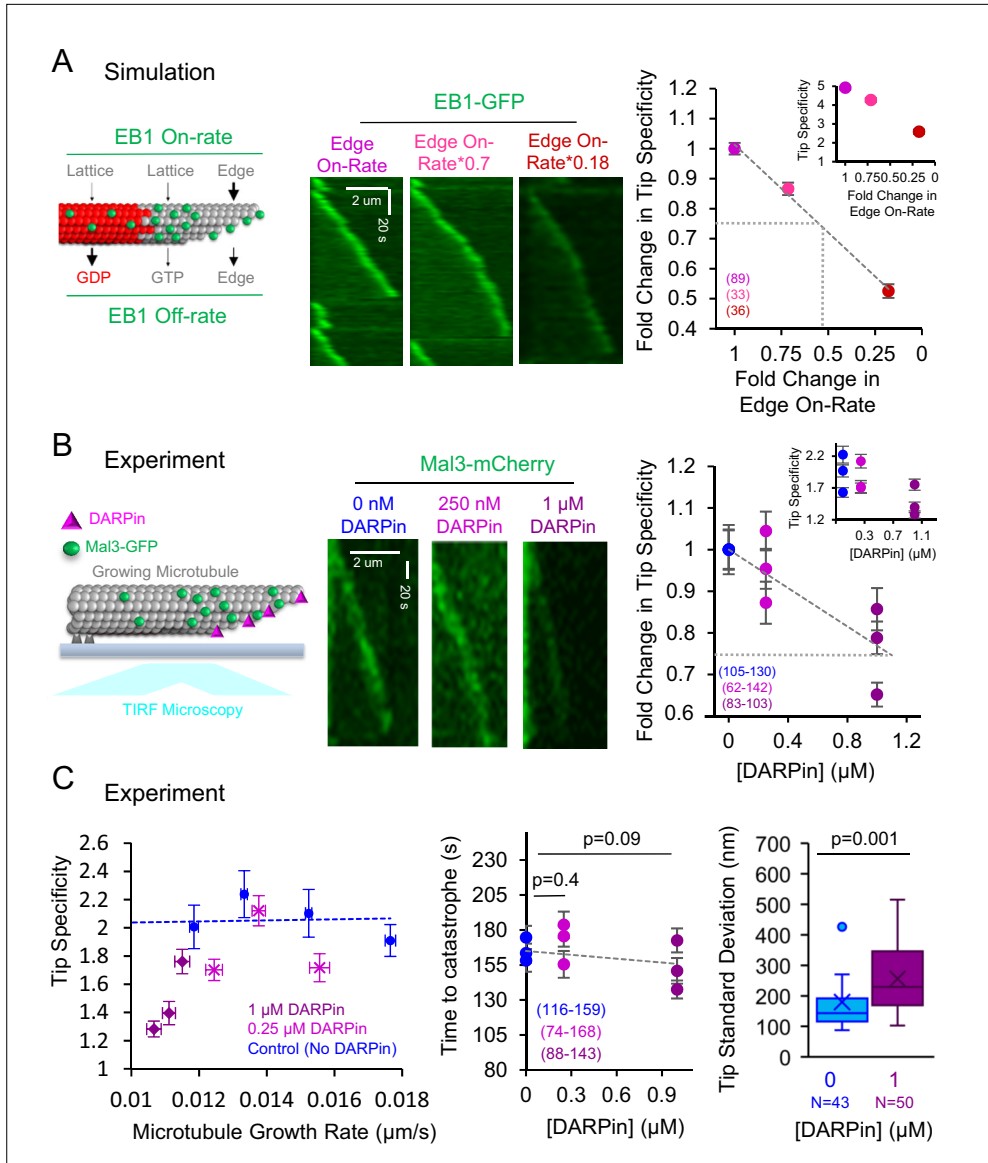

**Figure 5.** In cell-free experiments, Designed Ankyrin Repeat Protein (DARPin) disrupts tip tracking by blocking EB1 access to protofilament-edge sites. (**A**) Left: Cartoon depicting simulation rules for EB1 binding and unbinding to growing microtubules. Center: Simulated kymographs of EB1-GFP at growing microtubule tips, with decreasing protofilament-edge on-rates. Right: Simulation prediction for the fold change in the EB1 Tip Specificity as a function of the fold-change in the protofilament-edge on-rate (error bars: SEM; p<0.001, Kruskall Wallis). Gray dotted lines correspond to a twofold decrease in protofilament-edge on-rate, which leads to a ~25% decrease in predicted Tip Specificity. Inset: Absolute EB1 tip Specificity as a function of the fold-change in the protofilament-edge on-rate. (**B**) Left: Cartoon depicting experimental setup for dynamic microtubules with Mal3-mCherry, in the presence of DARPin, and visualized using TIRF microscopy. Center: Experimental kymographs of Mal3-mCherry tip tracking along dynamic microtubules in the presence of increasing DARPin concentrations. Right: Experimental results for the fold change in the Mal3-GFP Tip Specificity as a function of DARPin concentration (error bars: SEM; p<0.001, Kruskall Wallis). Gray dotted line represents a~25% decrease in Mal3-GFP Tip Specificity, which corresponds to a simulation prediction of a ~two-fold decrease in protofilament-edge on-rate. Inset: Absolute Mal3-GFP Tip Specificity as a function of DARPin concentration. (**C**) Left: The decrease in Mal3-GFP Tip Specificity in DARPin is not due to a decrease in microtubule growth rate (p<0.001, Kruskall Wallis comparing DARPin data to control data at a ~0.012 μm/s growth rate,). Center: The time to catastrophe is not altered by DARPin, suggesting that the GTP-cap size remains similar in the presence and absence of DARPin (Tukey's post-hoc analysis after an ANOVA). Right: Tip standard deviation in the presence and absence of DARPin (p=0.001, t-test,).

The online version of this article includes the following source data for figure 5:

*Figure 5 continued on next page*

*Figure 5 continued*

**Source data 1.** Data for *Figure 5*.

present to allow for an analysis of the relative comet brightness in the presence and absence of DARPin. Cells that were not transfected had a ~31% higher average tip:lattice EB1 intensity ratio as compared to the cells that were transfected with DARPin (*Figure 6C*, left vs right; *Video 9*; *Video 10*; *Figure 6D*; p<0.0001, t-test). In examining the microtubule growth rate in cells with and without DARPin (as determined by the EB1 comet velocity), we found that the microtubule growth rate for the comets that were visible in the presence of DARPin was similar to the cells without DARPin (*Figure 6E*; p=0.027, t-test), suggesting that the reduced comet intensity in DARPin was not due to a slowed microtubule growth rate.

## Discussion

In this work, we developed a molecular-scale computational simulation that incorporated both tubulin assembly dynamics, and EB1 on-off dynamics. Our simulation predicted that the binding of EB1 to protofilament-edge sites contributes to efficient tip-tracking of EB1 at growing microtubule plus-ends. To test this prediction, we used DARPin, a synthetic peptide, which binds to protofilament-edge sites on microtubules and partially overlaps the EB1 binding site. We found that DARPin blocks EB1 binding at protofilament-edge sites on stabilized microtubules, and importantly, this blocking of EB1 binding to protofilament-edge sites led to a disruption of EB1 tip tracking in dynamic microtubule cell-free assays, and in cells. We conclude that the rapid binding of EB1 to protofilament-edge sites facilitates the tip tracking of EB1 at growing microtubule ends. We note that, while we found that DARPin is not likely to indirectly suppress EB1 tip tracking by altering the plus-end tip conformation or the GTP-tubulin hydrolysis rate (*Figure 5C*), it remains possible that DARPin treatment could alter the conformation of the tubulin that composes the microtubule tip to indirectly suppress EB1 tip tracking. Furthermore, while the model leads to robust tip tracking by leveraging our previous experimental and simulation results that demonstrate the on-rate of EB1 molecules is rapid to protofilament-edge sites (*Reid et al., 2019*), we cannot exclude that another end-specific feature, that was not considered here, may be possible.

Previously, a model was developed to explain the peak position of EB1 on the growing microtubule tip, which is slightly distal from the tip of the growing microtubule (*Figure 1C*; *Maurer et al., 2014*). Because both our currently described model and the previously described model were able to reproduce the localization of EB1 on the microtubule, we sought to compare and contrast the described mechanisms in each of the two models.

In the previously described work by *Maurer et al., 2014*, a model was developed that relied on a constant length microtubule template with three EB1 binding 'zones.' Here, explicit tubulin assembly dynamics were not included in the model, but rather a constant length microtubule template was employed, in which there was an EB1 binding 'exclusion zone' at the tip of the microtubule. A tubulin subunit maturation rate was included in the model, which led to a second zone, slightly distal from the tip of the microtubule, in which EB1 binding was allowed. Finally, a second tubulin subunit maturation rate was employed, which led to a third zone, far from the tip and along the microtubule lattice, in which EB1 disassociation was allowed. The second tubulin subunit maturation rate likely corresponds to GTP-tubulin to GDP-tubulin hydrolysis, which is similar both in the magnitude of the hydrolysis rates employed, and in the EB1 off-rates employed, between the Maurer model and our newly described model (*Supplementary file 3*).

Thus, the primary difference between the two models was in the binding of EB1 to the microtubule lattice. Here, we predict that the key features of the Maurer model that involved exclusion of EB1 binding to the tip of the microtubule, along with a binding zone just distal to the microtubule tip, are incorporated into our newly described model by the ability of newly arriving tubulin subunits to 'lock in' protofilament-edge bound EB1 molecules into a stable 4-tubulin pocket, and by binding of EB1 to protofilament-edge sites along exposed protofilament sides that are distal from the tip of the microtubule. Specifically, in our new model, EB1 molecules arrive rapidly to easily accessible protofilament-edge sites at the growing tip of the microtubule, and to exposed protofilament sides (*Figure 7*, step 1). Then, upon new tubulin subunit addition, protofilament-edge bound EB1 molecules at the tip of

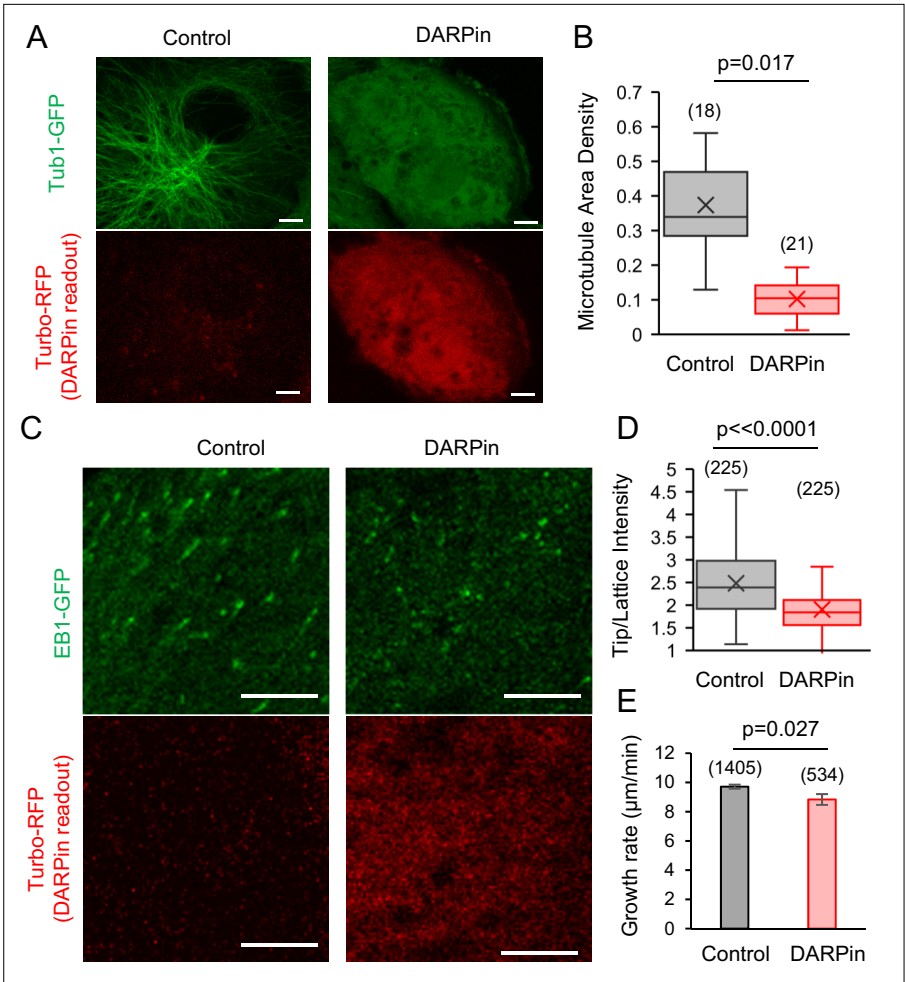

**Figure 6.** EB1 tip tracking is suppressed in cells that expressed Designed Ankyrin Repeat Protein (DARPin). (**A**) Left: Tubulin-GFP (top) and Turbo-RFP expression (bottom) in LLC-Pk1 cells that were not transiently transfected (scale bar: 5 µm). Right: Tubulin-GFP (top) and Turbo-RFP expression (bottom) in LLC-Pk1 cells that were transiently transfected (scale bar: 5 µm). (**B**) Microtubule density is lower in cells transfected with DARPin (p=0.017, Mann-Whitney U Test). (**C**) Left: EB1-GFP (top) and Turbo-RFP expression (bottom) in LLC-Pk1 cells that were not transiently transfected (scale bar: 5 µm). Right: EB1-GFP (top) and Turbo-RFP expression (bottom) in LLC-Pk1 cells that were transiently transfected (scale bar: 5 µm). (**D**) EB1-GFP Tip:Lattice intensity ratio is lower in cells transfected with DARPin as compared to control cells (p<0.0001, t-test). (**E**) Microtubule growth rate is similar in cells transfected with DARPin as compared to control cells (p=0.027, t-test).

The online version of this article includes the following source data for figure 6:

**Source data 1.** Data for *Figure 6B, D, and E*.

---

the microtubules are 'locked in' to a stable, 4-GTP-tubulin pocket (*Figure 7*, step 2), leading to a low EB1 off-rate. Thus, in our new model, EB1 accumulates on the GTP-cap at the growing microtubule end (*Maurer et al., 2014*; *Roth et al., 2019*). Finally, upon the hydrolysis of GTP-tubulin to GDP-tubulin, the affinity of EB1 for the GDP-tubulin subunits is reduced, and EB1 dissociates from the microtubule (*Figure 7*, step 3). Thus, we predict that the primary difference between our new model, and the previous Maurer et al model, may be in (1) the inclusion of tubulin assembly dynamics, and (2) rapid EB1 binding to protofilament-edge sites. These features eliminate the requirement for an explicit EB1 binding 'exclusion zone' at the tip of the microtubule, and naturally lead to a decrease in signal at the tip of the microtubule.

The tubulin assembly portion of the model was built on earlier work, in which individual tubulin subunits were allowed to arrive and depart from the growing microtubule plus-end (*Margolin et al., 2011*; *Margolin et al., 2012*) (see Methods). Future work will involve examining the effects of a

slower tubulin association rate on EB1-occupied protofilament edge sites, and whether EB1 binding to protofilament edge sites could assist in neighboring protofilament zippering on flared microtubule tips. Furthermore, microtubule targeting drugs that suppress the kinetics of tubulin assembly at the growing microtubule plus-end, such as Taxol (*Castle et al., 2017*), could potentially disrupt EB1 tip tracking by slowing the capture and 'lock in' of EB1 to 4-tubulin pocket binding sites (*Figure 7*, step 2), an idea that could be explored in future work.

As described above, our model predicts that rapid protofilament-edge binding increases the efficiency of EB1 tip tracking. In the simulation, the peak EB1 location is slightly distal from the tip of the growing microtubule (*Figure 1C*), similar to previous reports (*Maurer et al., 2014*). We surmise that the peak EB1 location in the simulation is heavily influenced by EB1 molecules that are stably bound to GTP-tubulin closed-lattice sites on the growing microtubule tip. Indeed, by reporting the fraction of EB1 molecules that are bound to GTP-Tubulin protofilament-edge sites as compared to GTP-Tubulin closed lattice sites, we found that there are ~twofold more EB1 molecules bound to closed-lattice GTP-tubulin sites, as compared to protofilament-edge sites, at any one time in the simulation (*Figure 1—figure supplement 1H*). Furthermore, the number of EB1 binding sites at the tip of each protofilament is explicitly limited by the number of protofilaments in the microtubule (13 binding sites). Thus, EB1 binding to numerous protofilament-edge sites along exposed protofilament sides that are distal from the tip of the microtubule may also contribute to the peak EB1 location. This idea is consistent with results from the 'split comet' simulations (*Figure 3A and B*). Here, by substantially increasing the taper at the tip of the simulated growing microtubule (≤~3 µm), the EB1 comet was greatly extended in length, and altered in configuration, thus shifting the location of EB1 binding (*Figure 3B*). However, the location of the simulated EB1 peak position was insensitive to small changes in tip taper (*Figure 1—figure supplement 1B*).

A key aspect of the simulation is that EB1 molecules arrive rapidly to protofilament-edge sites at the tip of the growing microtubule. We propose that, because the on-rate of new tubulin molecules is also rapid (simulated arrival rate for tubulin: ~85 s⁻¹ at 10 µM tubulin), the simulated EB1 molecules that bind to protofilament-edge sites at the tip of the growing microtubule are quickly 'locked in' to a closed-lattice GTP-tubulin binding configuration (*Figure 7*). Thus, while most of the EB1 molecules at the microtubule tip are indeed bound to closed-lattice GTP-tubulin sites (*Figure 1—figure supplement 1H*), many of these EB1 molecules likely originated as arrivals to protofilament-edge sites (*Figure 1—figure supplement 1A*). To test this idea, we ran simulations in which we recorded the initial binding location of EB1, to determine the fraction of EB1 molecules that initially bound to protofilament-edges as compared to closed-lattice positions (*Figure 1—figure supplement 1I*). We found that ~50% of all EB1 binding events occurred at protofilament-edge sites (*Figure 1—figure supplement 1I*). However, importantly, the EB1 molecules that initially bound to protofilament-edge sites were heavily concentrated at the growing microtubule tip, while EB1 molecules that bound to closed-lattice sites were more uniformly distributed throughout the microtubule (*Figure 1—figure supplement 1J and A*; *Video 2*). This is because closed-lattice binding occurs throughout the microtubule, rather than specifically near to the growing microtubule plus-end.

While protofilament-edge binding is a key aspect of our model, it is important to emphasize that both rapid binding of EB1 to protofilament-edges (50-100:1 edge:lattice), as well as a differential GDP-to GTP-tubulin off-rate (6-12:1 GDP:GTP), were critical to produce robust tip tracking in the model. Because both of these factors contribute to tip tracking, this leads to a highly robust model, that does not require a narrow set of parameter values for either effect, in order to reproduce experimental results (*Figure 2G–I*). Thus, based on the experimentally measured EB1 on and off rates, a differential GDP- to GTP-tubulin off-rate, together with rapid protofilament-edge binding, was required for EB1 tip tracking in the model. Importantly, we observed robust tip tracking in the model without requiring narrow parameter sets, or by establishing an EB1 binding exclusion zone on the microtubule, as has been previously hypothesized.

Recent work has demonstrated that growing microtubule tips are less homogeneous than previously thought, such that they exhibit a wide range of protofilament lengths between the leading and lagging protofilaments, both in cells and in cell-free experiments (*Atherton et al., 2018*; *Cleary and Hancock, 2021*; *Coombes et al., 2013*; *Gudimchuk et al., 2020*; *Guesdon et al., 2016*; *Igaev and Grubmüller, 2022*). Here, a wide range of protofilament lengths at the growing microtubule end would lead to increased numbers of protofilament-edge sites on the exposed protofilament sides, which, in

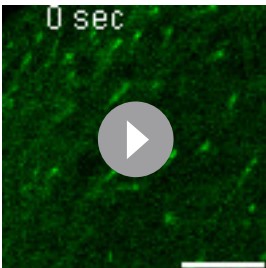

**Video 9.** Experimental EB1-GFP tip tracking in live LLC-PK1 cells without Designed Ankyrin Repeat Protein (DARPin) transfection. 5 μm scale bar.
https://elifesciences.org/articles/91719/figures#video9

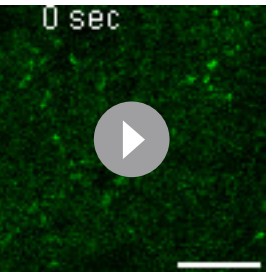

**Video 10.** Experimental EB1-GFP tip tracking in live LLC-PK1 cells with Designed Ankyrin Repeat Protein (DARPin) transfection. 5 μm scale bar.
https://elifesciences.org/articles/91719/figures#video10

our model, would increase the EB1 on-rate to the tip. In addition, in our model, other tip configurations, such as partial curvature (*Bechstedt et al., 2014*; *Farmer et al., 2021*), or flaring of the growing microtubule plus-end (*McIntosh et al., 2018*), would also contribute to EB1 tip tracking (*Figure 1—figure supplement 1C*). Here, partial curvature or flaring requires opening of the closed microtubule tube – indicating that protofilaments or groups of protofilaments are separated from each other. Importantly, separation between protofilaments means that the number of protofilament-edge sites would be enriched, as new protofilament sides would be exposed. Thus, the role of protofilament-edge sites in facilitating EB1 tip tracking could apply to a wide range of growing microtubule tip configurations.

Recently published work provides support for the importance of EB1 protofilament-edge site binding in the efficiency of EB1 tip tracking. Specifically, by using the microtubule polymerase protein XMAP215 in cell-free experiments, the range of protofilament lengths between the leading and lagging protofilaments at the growing microtubule plus-end was increased (*Farmer et al., 2021*). Importantly, an increase in EB1-GFP intensity at the growing microtubule tip was observed with increased XMAP215-induced tip taper (*Farmer et al., 2021*). We note that increased tip taper would likely correspond to an increase in the number of protofilament-edge sites along exposed protofilament sides, similar to our split comet phenotype (*Figure 3*). Thus, XMAP215 could increase the efficiency of EB1 tip tracking by adding new protofilament-edge sites to the growing microtubule plus-end. This suggests that EB1 recruitment, and by extension the recruitment of the +Tip Complex, could be sensitive to the number of protofilament-edge sites at the tip of the growing microtubule. Correspondingly, a recent report found that EB1, and thus CLASP2, is redistributed from the plus-end onto the microtubule lattice in cells subjected to stretch and compression cycles (*Li et al., 2023*). This result is consistent with the idea that microtubule bending could cause openings and holes in the lattice, leading to the creation of new protofilament-edge sites along the lattice, which in turn causes a redistribution of EB1 from the plus-end tip to the lattice (*Figure 4C*, blue).

In conclusion, we find that protofilament-edge sites are an important contributing factor for proper EB1 tip tracking along growing microtubule ends, and that EB1 tip tracking is suppressed by blocking the protofilament-edge sites at growing microtubule ends. Therefore, altering the number of exposed protofilament-edge sites at the growing microtubule tip, or along the microtubule lattice, may provide a new mechanism to regulate EB1 localization in cells.

## Materials and methods
### Simulation methods

The simulation was performed in MATLAB, and all code has been deposited in GitHub (copy archived at *Gonzalez, 2024a*).

The microtubule assembly portion of the simulation is based on work from the Goodson lab (*Margolin et al., 2012*). Briefly, this model allows microtubule protofilaments to grow independently via the addition of individual tubulin subunits, and to form and break lateral bonds with neighboring protofilaments, once individual tubulin subunits are longitudinally bound to the microtubule. All of the parameter values for the microtubule assembly simulation matched a previously published

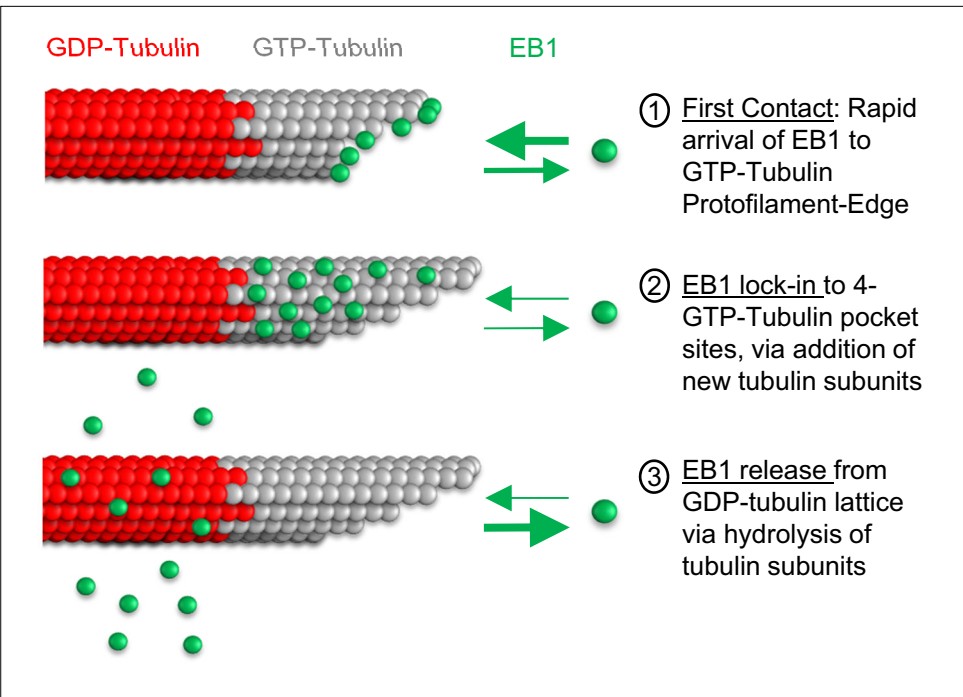

**Figure 7.** Model for EB1 tip tracking. Step 1 (top): EB1 binds rapidly to protofilament-edge sites, with a slower on-rate to 4-tubulin closed lattice binding sites. Step 2 (middle): Incorporation of new tubulin subunits 'locks in' EB1 bound at protofilament-edge sites as the binding pocket transitions from a protofilament-edge site to a 4-tubulin closed-lattice site. Step 3 (bottom): As the GTP-tubulin hydrolyzes to GDP-tubulin, the EB1 dissociates from the GDP closed-lattice binding site.

parameter set (*Margolin et al., 2012*; *Supplementary file 1*), with the exception of (1) the tubulin on-rate constant, which was lowered in order to match our (slow) experimental growth rates, and (2) one additional rule was added to ensure that the tip taper at the microtubule plus-end matched our experimental values (*Figure 1—figure supplement 1A and B*). Here, if the difference between the longest and the penultimate shortest protofilament exceeded 600 nm (75 dimers) (*Ogren et al., 2022*), the tubulin subunit off-rate and the lateral bond breakage rate were dramatically increased, quickly leading to a catastrophe event.

In the EB1 binding and unbinding portion of the simulation, the model allows for EB1 to bind to and dissociate from the microtubule independently of tubulin addition/dissociation. Here, EB1 molecules bind to protofilament-edge sites and closed-lattice sites with differential on-rates, and then dissociate from GTP- and GDP-tubulin binding sites with differential off-rates (see *Supplementary file 2*).

At the start of every time step in the simulation, the total execution time was calculated for each potential event, which included EB1 association/dissociation, tubulin association/dissociation, and lateral bond formation/breakage between protofilaments, using the equation:

$$time = \frac{-log\left(rand\right)}{k} \tag{2}$$

where *time* is the total execution time required for each potential event, *rand* represents the built-in MATLAB function that generates a uniformly distributed random number between 0 and 1, and *k* is the single-molecule rate constant for each potential event (see *Supplementary files 1 and 2*).

Then, once all of the total execution times were calculated for each potential event, the simulation executed only one event, with the shortest time.

After the event was executed, the time to GTP-tubulin hydrolysis was calculated using *Equation 2*. If the hydrolysis time was shorter than the time of the executed event, then one of the GTP tubulin dimers was randomly hydrolyzed. Otherwise, no hydrolysis occurred in that time step.

After every time step in the simulation, every tubulin dimer's hydrolysis state was recorded (GTP vs GDP), and the configuration of every potential EB1 binding site was also recorded (GTP vs GDP; Edge

vs Lattice; EB1 bound or not bound). After each time step in the simulation, the tubulin dimer and EB1 binding site states and configurations were updated based on activities during the time step (e.g. tubulin association/dissociation, tubulin hydrolysis, EB1 binding/dissociation, lateral bond breakage/ formation). Every thousand-time steps of the simulation (which averaged between 0.5 and 2 s of real-time in the simulation), the length of each microtubule protofilament, and the position of every bound EB1 were stored. This was repeated until the simulation ended, which in general was between $6 \times 10^4$ and $4.5 \times 10^5$ steps, depending on which simulation experiment was being performed. This stored data was then saved as excel files for further analysis (*Crosby, 2024*).

## Creating model-convolved images from the simulation

To visualize the output from the simulation, 15% of the tubulin dimers were randomly labeled with a red fluorophore and plotted on an image. Then, all occupied EB1 positions were labeled with a green fluorophore and plotted. Next, uniform random noise was added to the image. Finally, the simulated image was convolved using the microscope point spread function, as would be observed on our TIRF microscope (*Demchouk et al., 2011*; *Gardner et al., 2010*; *Reid et al., 2019*). This model-convolution process was performed using the stored data from every 1000 steps in each simulation, and allowed for the generation of simulated videos in which the simulated EB1 localization along growing microtubules could be observed (GitHub, copy archived at *Gonzalez, 2024b*).

## Analyzing line scans of simulated images for peak EB1 position and tip:lattice intensity

To quantify EB1 tip tracking in the simulation, line scans along the length of simulated microtubules were obtained using ImageJ. Then, each image was aligned along the peak EB1 position. Next, at least five separate simulation runs were averaged. To determine the end of the microtubule, the position where the intensity of the microtubule channel (red) was halfway between its maximum value and its minimum value was deemed as the microtubule end, and denoted with a relative position of zero nanometers. From here, the peak EB1 position could be determined by finding its relative position with respect to the microtubule end at position 0. In all simulated data that used line scans, N refers to the number of individual simulation growth events runs. To determine the tip:lattice intensity values, the maximum EB1 intensity at the tip was divided by the median EB1 intensity along the lattice, which was measured at 768–1024 nanometers distal of the microtubule plus-end.

## Analyzing line scans of simulated and experimental images for microtubule tapering

To quantify the simulated and experimental microtubule tip standard deviation (*Figure 1—figure supplement 1B* and *Figure 5C*), images were loaded into a previously described program to measure tip tapering (*Demchouk et al., 2011*). In short, the user-defined points along the start and end of the microtubule lattice, and then the intensity of the microtubule across this distance was measured. Then, this intensity was fitted to a Gaussian survival curve to determine the tip standard deviation of the microtubule (which includes the microscope point spread function as well as the underlying proto-filament length standard deviation). Each data point is the standard deviation for a single simulation run. This process was repeated for experimental microtubules.

## Simulation parameter testing

To determine the robustness of tip tracking in our simulation, we ran a series of simulations in which all parameters but one were held constant, and then the parameter being tested was altered in twofold increments. The range for this parameter testing spanned from 1/16 to 16-fold times the baseline parameter value for each key parameter in the simulation. Ten simulation runs were performed for each parameter set. In each case, a simulated video was generated, and a line scan was obtained, as described above. To determine the uncertainty for a given parameter across the range we tested, we calculated the sum of absolute errors between the simulated and experimental Mal3 line scan profiles (*Bieling et al., 2007*).

## Examining the effect of microtubule tip flaring and EB1 monomers on simulation tip tracking

In the simulation, the most distal lateral bond between protofilaments is tracked. To simulate flared microtubule tips, a rule was added in which all EB1 binding sites past the most distal lateral bond between protofilaments were considered protofilament-edge sites. Then, the effect of this rule on EB1 tip tracking was examined.

To examine the effect of EB1 monomer binding in our simulation, we course-grained monomer binding by increasing all EB1 off-rates by fourfold, as has been shown in the literature (*Song et al., 2020*). Then, EB1 tip tracking was examined with these faster off-rates.

## Split comet simulations

To generate split comets in our simulations, we increased the maximum taper length that was allowed before increasing $k_{lateral\ bond\ break}$ and $k_{off}$ ($T_{dimer}$ ~400 dimers long (~3 μM)). In addition, we lowered the $\pi_{break}$ value in the simulation from 10 to 6, and we increased the tubulin concentration *[GTP-tub]* in the simulation to 30 μM. The change to max taper length and $\pi_{break}$ values both encouraged longer tip tapers, and by extension a higher likelihood of split comets. The increase in the tubulin concentration reduced the catastrophe frequency, to increase the lifetime of split comets. Then, simulations were run as previously, and only growth events were analyzed that produced split comets (not all growth events produced split comets). When there were split comets, images were cropped during the same growth event for the brightest single-comet frame, the brightest split-comet frame, and for a seed-only frame (no dynamic microtubule, for background). Next, the smallest region that encapsulated all the split and single comets in both images was determined. The intensity across this region was summed for all three images (single comet, split comet, no comet). Then, the background intensity (no comets) was subtracted from the single comet and split comet values, allowing for a comparison of the summed intensity of EB1 along the simulated growing microtubule with a single comet and with split comets.

## Experimental methods

### Tubulin purification and labeling

Tubulin was purified and labeled as previously described (*Gell et al., 2010*).

### Preparation of GTP-analogue microtubules

GMPCPP microtubules were prepared as previously described *Gell et al., 2010*. In short, 3.9 μM rhodamine labeled tubulin was incubated with 1 mM GMPCPP with 1.1 mM $MgCl_2$ in BRB80 for 5 min on ice. Then, the solution was incubated at 37 °C for 2 hr or overnight. The GMPCPP microtubules were then spun with an airfuge (Beckman Coulter, 20 psi, 5 min) and further stabilized in 10 μM Taxol in BRB80. These microtubules were stored at 37 °C and used in experiments within four days after preparation. These microtubules were used as seeds for dynamic microtubule experiments and were used for Mal3 binding along GMPCPP microtubules.

### Purification of Mal3-GFP

The pETMM11-HIS6x-Mal3-GFP plasmid with a TEV cut site after the His6x tag was a kind gift from Dr. Thomas Surrey. The plasmid was transformed into Rosetta (DE3) pLysS *E. coli* and grown in 800 mL of LB + kan + cam at 37 °C to an OD of approximately 0.4. To induce protein expression, IPTG was added to 0.2 mM and the culture was mixed at 14 °C for 16 hr. Cells were centrifuged (30 min., 4 °C, 4400 × g) and resuspended in 25 mL lysis buffer (50 mM Tris pH7.5, 200 mM NaCl, 5% glycerol, 20 mM imidazole, 5 mM β-mercaptoethanol, 0.2% triton X-100), protease inhibitors (1 mM PMSF, 10 μM Pepstatin A, 10 μM E-64, 0.3 μM aprotinin), and DNAse I (1 U/mL). The cell suspension was sonicated on ice (90% power, 50% duty, 6 × 1 min). Cell lysates were centrifuged (1 hr, 4 °C, 14000 × g) and the soluble fraction was passed through 1 mL of Talon Metal Affinity Resin (Clontech #635509). The resin was washed for four times with 4 mL Wash Buffer (50 mM Tris pH7.5, 500 mM NaCl, 5% glycerol, 20 mM imidazole, 5 mM β-mercaptoethanol, 0.1 mM PMSF, 1 μM Pepstatin A, 1 μM E-64, 30 nM aprotinin) for 5 min each. Protein was eluted from the resin by mixing with 1 mL of Elution Buffer (50 mM Tris pH 7.5, 200 mM NaCl, 250 mM imidazole, 0.1 mM PMSF, 1 μM Pepstatin A, 1 μM E-64, 30 nM aprotinin) for 15 min followed by slow centrifugation through a fritted column to retrieve

eluate. To cleave the HIS6x tag, 10 units of GST-tagged TEV enzyme (TurboTEV, #T0102M, Accelagen) and 14 mM β-mercaptoethanol were added and the eluate was dialyzed into Brb80 overnight at 4 °C. To remove the TEV enzyme, the dialysate was mixed with 100 ul of glutathione-sepharose (GE Healthcare #17-0756-01) for 30 min. at 4 °C and spun (1 min, 2000 × g). The Mal3-GFP protein was quantified by band intensity on a coomassie-stained SDS PAGE protein gel.

## Purification of Mal3-mCherry

Purification of Mal3 from bacteria was based on the protocol described previously *Gerson-Gurwitz et al., 2011*; *Hepperla et al., 2014*. His6x-Mal3-mCherry is in the pETMM11 vector with a TEV cut site after the His6x tag, kindly provided by Dr. Thomas Surrey. This plasmid, in Rosetta (DE3) pLysS *E. coli*, was grown in 600 ml of TB + kan + cam at 37° to an OD of about 0.4, then IPTG was added to 0.2 mM to induce protein expression and growth continued at 14° for 20 hr. Cells were centrifuged and resuspended in 25 ml lysis buffer (50 mM Tris pH7.5, 200 mM NaCl, 5% glycerol, 20 mM imidazole, 5 mM B-mercaptoethanol, 0.2% triton X-100) plus protease inhibitors (1 mM PMSF, 10 uM Pepstatin A, 10 uM E-64, 0.3 uM aprotinin). DNAse I was added to 1 U/ml in the cell suspension and sonicated on ice at 90% power, 50% duty for six cycles (1 min. on, 1 min. off) to lyse. Lysate was centrifuged at 14000 × g, 4°, for 1.5 hr and the soluble fraction was mixed with 1 ml of Talon affinity resin (Clontech #635509) at 4° for 1 hr. Resin was poured into a small column and washed in 5 min. sequences with buffers with 0.1 x protease inhibitors: two times with lysis buffer, two times with lysis buffer/700 mM NaCl, one time with lysis buffer. Protein was eluted from the resin by mixing for 15 min with 2 ml of elution buffer (lysis buffer/250 mM imidazole)+0.1 x protease inhibitors, and slowly centrifuging the column to collect all the eluate. 10 units of TEV enzyme (TurboTEV, #T0102M, Accelagen) and B-mercaptoethanol to 14 mM was added and the eluate was dialyzed into Brb80 (80 mM PIPES pH6.9, 1 mM MgCl2, 1 mM EGTA) at 4°. The dialysate was mixed with four 100 ul of glutathione-sepharose (GE Healthcare #17-0756-01) for 30 min., 4° to remove the TEV enzyme, which has a GST tag. The Mal3-mCherry protein was quantified by band intensity on a coomassie-stained SDS PAGE protein gel.

## Purification of DARPin D1

A plasmid containing the D1-Darpin sequence with a 6-HIS tag, a generous gift from Dr. Andreas Plückthun and Dr. Benoît Gigant (*Pecqueur et al., 2012*) was grown in XL1-Blue bacteria in LB + ampicillin media to an A600 of 0.6, then IPTG was added to 1 mM and grown for 21 hr. at 18°. All subsequent steps were performed at 4°. The centrifuged cell pellet was resuspended in lysis buffer (50 mM Tris pH8 /10 mM imidazole/1 mM MgCl2 /0.3 mg/ml lysozyme with cOmplete EDTA-free protease inhibitor (Sigma #4693159001)), incubated on ice for 30 min., lysed by sonication on ice and cell debris was removed by centrifugation at 18000 × g for 30 min. The soluble lysate was passed over 1 ml Talon metal affinity resin column (https://www.takarabio.com/ #635502) three times, the resin was then washed with 10 volumes of wash buffer (50 mM Tris pH8 /10 mM imidazole/1 mM MgCl2) and eluted with 50 mM Tris/300 mM imidazole/1 mM MgCl$_2$ pH8. The eluted protein was dialyzed against Brb80 (80 mM PIPES/1 mM MgCl2 /1 mM EGTA pH6.9) and centrifuged to remove any precipitate. Purifed D1-Darpin was quantified by measuring band intensity on a Coomassie G-250-stained acrylamide protein gel.

## Creation of TIRF microscopy flow chambers for cell-free assays

Imaging flow chambers were constructed as in Section VII of *Gell et al., 2010*, with the following modifications: two narrow strips of parafilm replaced double-sided scotch tape as chamber dividers: following placement of the smaller coverslip onto the parafilm strips, the chamber was heated to melt the parafilm and create a seal between the coverslips; typically, only three strips of parafilm were used, resulting in two chambers per holder. Chambers were prepared with an anti-rhodamine antibody (Invitrogen A6397, RRID:AB_2536196) followed by blocking with Pluronic F127, as described in Section VIII of *Gell et al., 2010*. Microtubules were adhered to the chamber coverslip, and the chamber was flushed gently with warm BRB80. The flow chamber was heated to 28 °C using an objective heater on the microscope stage, and then 3–4 channel volumes of imaging buffer were flushed through the chamber. Microtubules were imaged on a Nikon TiE microscope using 488 nm and 561 nm lasers sent through a Ti-TIRF-PAU for Total Internal Reflectance Fluorescence (TIRF) illumination. An Andor

iXon3 EM-CCD camera fitted with or without a 2.5x projection lens depending on the experiment was used to capture images with high signal-to-noise and small pixel size (64 nm or 160 nm, respectively). Images were collected using TIRF with a Nikon CFI Apochromat 100x 1.49 NA oil objective.

## Cell-free microtubule assays

For the damaged GTP-analogue microtubule assays, GMPCPP microtubules were introduced into a flow chamber as described above and allowed to incubate for 3 min before flushing out any non-adhered microtubules with BRB80. Next, 10 mM $CaCl_2$ in warmed BRB80 was introduced into the chamber and incubated for 1–5 min, until obvious degradation occurred to the microtubules. The chamber was then washed with multiple chamber volumes of warmed BRB80. Next, the chamber was washed with one chamber volume of prewarmed imaging buffer (20 µg/mL glucose oxidase, 10 µg/mL catalase, 20 mM D-Glucose, 10 mM DTT, 80 µg/mL casein, 110 mM KCl, and 1% tween-20). Finally, a Mal3 reaction mixture with or without 1 uM DARPin (imaging buffer plus 123 nM Mal3-GFP) was introduced to the chamber and allowed to incubate for 15 min. Images of hundreds of non-overlapping fields of view were collected and used for downstream analysis.

For undamaged GTP-analogue microtubule assays, GMPCPP microtubules were introduced into a flow chamber as described above and allowed to incubate for 30 s to 3 min before flushing out any non-adhered microtubules with BRB80. Next, one chamber volume of prewarmed imaging buffer (20 µg/mL glucose oxidase, 10 µg/mL catalase, 20 mM D-Glucose, 10 mM DTT, 80 µg/mL casein, 110 mM KCl, and 1% tween-20) was added. Finally, a reaction mixture with or without 1 uM DARPin (imaging buffer plus 123 nM Mal3-GFP) was introduced to the chamber and allowed to incubate for 15 min. Finally, images of hundreds of non-overlapping fields of view were obtained and used for downstream analysis.

For the dynamic microtubule assays, GMPCPP microtubule seeds were introduced into a flow chamber as described above and allowed to incubate for ~3 min before flushing out any non-adhered microtubules with BRB80. Next, one chamber volume of prewarmed imaging buffer (20 µg/mL glucose oxidase, 10 µg/mL catalase, 20 mM D-Glucose, 10 mM DTT, 80 µg/mL casein, 110 mM KCl, and 1% tween-20) was added. Finally, a reaction mixture consisting of Imaging buffer plus 212 nM Mal3-mCherry, 11.5 µM of 12% green-labeled tubulin, and 1 mM GTP, with or without DARPin, was added. Time-lapse images were collected of dynamic microtubules growing from the GMPCPP stabilized seeds and were then used for quantification.

## Analyzing microtubule area bound by Mal3

To compare the binding of Mal3-GFP on undamaged and damaged microtubules in the presence and absence of DARPin, the total length of green (Mal3-GFP) occupancy was divided by the total length of the red microtubules on each image. This was accomplished by using a previously described semi-automated MATLAB analysis code (*Reid et al., 2017*). Briefly, first, automatic processing of the red microtubule channel was used to determine the microtubule-positive regions, which then allowed for the conversion of the red channel into a binary image with white microtubules and a black background. The green Mal3-GFP channel was then also pre-processed to smooth high-frequency noise and to correct for TIRF illumination heterogeneity. The green channel threshold was then manually adjusted to ensure visualization of all Mal3-GFP binding areas on each microtubule. Measurements of the total Mal3-GFP coverage area were then automatically collected from the identified microtubule regions. Finally, the total coverage area of Mal3-GFP was divided by the total microtubule area in each field of view. This experiment was replicated three times, as shown in the main text and in supplemental material.

## Analysis of Mal3 tip tracking

To determine the Tip Specificity (*Equation 1*), a custom MATLAB script was written that allowed the user to pick the brightest point of the comet, then a point on the microtubule lattice, behind the comet, and then a point alongside the comet for background. Then, a 4 × 4 pixel box was summed at the brightest point of the comet ($I_{tip}$), another 4 × 4 pixel box was summed at the point behind the comet ($I_{lattice}$), and a final 4 × 4 pixel box was summed alongside the tip ($I_{background}$). Finally, *Equation 1* was used to calculate the Tip Specificity (*S*) (GitHub, copy archived at *Gonzalez, 2024c*). Only the brightest comet per growth event was measured and every growth event in a field of view was

analyzed. This was completed over three replicate experiments, and pooled together for each condition. To analyze EB1-GFP comets in LLC-Pk1 cells, the brightest comets in each frame were cropped. These comets were then not cropped in later frames to ensure each comet was only analyzed once during its lifetime.

For the growth rate and time to catastrophe measurements, kymographs were made from representative growth events. Then, the growth rate was calculated from the kymograph by determining the length of the microtubule at the start and end of each growth event and dividing by the time required to reach the end of that growth event (GitHub, copy archived at *Gonzalez, 2024d*). The time to catastrophe was determined by examining growth events that started from the microtubule seed, and by measuring the elapsed time until a microtubule catastrophe event occurred.

To examine split comets, dynamic microtubules were examined to determine whether split comets were present. If there were split comets, then an image was cropped during the same growth event for the brightest single-comet frame, the brightest split-comet frame, and for a seed-only frame (no dynamic microtubule, for background). Next, the smallest region that encapsulated all of the split and single comets in both images was determined. The intensity across this region was summed for all three images (single comet, split comet, no comet). Then, the background intensity (no comets) was subtracted from the single comet and split comet values, allowing for a comparison of the summed intensity of Mal3-mCherry along the growing microtubule with a single comet and with split comets.

## Crystal structure diagram of DARPin and Mal3

To generate a structural schematic, the crystal structure of DARPin bound to tubulin (4drx) (*Pecqueur et al., 2012*) was aligned with the crystal structure of Mal3 bound to tubulin (4abo) (*Maurer et al., 2014*) where the β-tubulin with DARPin and Mal3 bound were used for the alignment in Chimera (*Pettersen et al., 2021*).

## Cloning of DARPin into a Turbo-RFP-P2A vector

An RFP-P2A-DARPin plasmid was generated by isolating the DARPin sequence from the DARPin bacterial expression vector (DARPin D1 in pDST67, *Pecqueur et al., 2012*) via a restriction digestion with the BamHI and HindIII sites and cloning it into FLAG-HA-mRFP-pcDNA3.1 vector (plasmid #52510, Addgene) using the same BamHI and HindIII sites (RFP-DARPin vector). Proper integration of the DARPin sequence was verified with sequencing. Next, we isolated a TurboRFP and P2A from a separate vector (plasmid # 78933, Addgene) via a restriction digestion with the NHEI and BamHI sites and cloned it into the RFP-DARPin vector with the BamHI and NHEI sites leading to a mammalian expression vector with TurboRFP and P2A N-terminal to DARPin (TurboRFP—P2a—DARPin vector). Proper integration of the Turbo-RFP and P2A sequence was verified with sequencing.

## Cell lines

The LLC-Pk1 cell line expressing EB1-GFP was a gift from Dr. Patricia Wadsworth (*Piehl et al., 2004*), and the cell line expressing GFP-Tubulin was a gift from Dr. Lynne Cassimeris (*Rusan et al., 2001*). The identities of the cell lines (non-human) were authenticated by microscopy observation and analysis.

## Culture and imaging of LLC-PK1 cells

The LLC-PK1 cell lines were grown in Optimem media (Thermo Fisher #31985070), 10% fetal bovine sera + penicillin/streptomycin at 37 °C and 5% $CO_2$. Cells were grown in 14 mm glass bottom dishes for visualization by microscopy. Cells were imaged with a laser scanning confocal microscope (Nikon Ti2, 488 nm laser line) fitted with a 100 x oil objective (Nikon N2 Apochromat TIRF 100 x Oil, 1.49 NA), which allowed for a 0.16 μm pixel size.

## Transfecting LLC-PK1 cells

LLC-PK1 cells were transfected with Lipofectamine 3000 following the manufacturer's protocol, except that the transfection was performed for 16 hr before imaging rather than 2–4 days before imaging. Immediately before imaging, cells were transferred into $CO_2$-independent imaging media.

## LLC-Pk1 microtubule growth rate analysis

To analyze EB1-GFP comet velocity, which was used as a proxy for the microtubule growth rate, we employed analysis software from the Danuser lab (*Applegate et al., 2011*). In short, we collected multiple 100 by 100-pixel movies of LLC-Pk1 cells treated with DMSO or transfected with DARPin from three separate biological replicates and loaded them into the Danuser code software, using constant parameters for thresholding and water shedding. We then allowed the program to identify, link, and track comets over time, which provided us with EB1 comet velocities across multiple cells. We next cut off any outlier values greater than 1 µm/s, which were likely artifacts from the analysis software. The growth rates were statistically analyzed using a student t-test.

## LLC-PK1 microtubule density analysis

To determine the microtubule density in LLC-Pk1 cells that were or were not transfected with DARPin, LLC-Pk1 cells overexpressing Tubulin-GFP were grouped by RFP expression, with RFP expression indicating a successful transfection of DARPin. Z-stacks were acquired across the volume of these cells using confocal microscopy. Then, maximum Z-projections were created, followed by the analysis of the area of the microtubules divided by the area of the cell, which was performed with a custom MATLAB script (GitHub, copy archived at *Gonzalez, 2024e*; *Goldblum et al., 2021*). Finally, this normalized value was compared using a Mann-Whitney U test.

## Materials availability

All materials generated during this study are available by contacting the Gardner lab at klei0091@umn.edu.

## Acknowledgements

The Gardner laboratory is supported by a National Institutes of Health grant NIGMS R35-GM126974. SJG was supported in part by the National Institute of Health Training Program T32GM140936. We thank members of the Gardner, Courtemanche, and Titus laboratories for helpful discussions. We thank Dr. Andreas Plückthun and Dr. Benoît Gigant for the generous gift of the DARPin construct, and Dr. Thomas Surrey for the kind gift of Mal3 constructs.

## Additional information

### Funding

| Funder | Grant reference number | Author |
| --- | --- | --- |
| National Institutes of Health | R35-GM126974 | Melissa K Gardner |
| National Institutes of Health | T32GM140936 | Samuel J Gonzalez |

The funders had no role in study design, data collection and interpretation, or the decision to submit the work for publication.

### Author contributions

Samuel J Gonzalez, Software, Formal analysis, Validation, Investigation, Visualization, Methodology, Writing - original draft, Writing – review and editing; Julia M Heckel, Formal analysis, Investigation, Writing – review and editing; Rebecca R Goldblum, Taylor A Reid, Software, Writing – review and editing; Mark McClellan, Resources, Methodology, Writing – review and editing; Melissa K Gardner, Conceptualization, Data curation, Formal analysis, Supervision, Funding acquisition, Investigation, Methodology, Project administration, Writing – review and editing

### Author ORCIDs

Samuel J Gonzalez http://orcid.org/0000-0002-5372-8068
Julia M Heckel https://orcid.org/0000-0002-5953-1717

Melissa K Gardner  https://orcid.org/0000-0001-5906-7363

**Decision letter and Author response**
Decision letter https://doi.org/10.7554/eLife.91719.sa1
Author response https://doi.org/10.7554/eLife.91719.sa2

## Additional files

### Supplementary files
• Supplementary file 1. Summary table with simulation parameters for the microtubule assembly portion of the simulation. Parameters determine the on and off rates of tubulin subunits from the microtubule tip, as well as parameters that control the hydrolysis rate of GTP-tubulin subunits within the lattice.

• Supplementary file 2. Summary table with simulation parameters for single-molecule EB1 dynamics. Parameters determine the on and off rates of EB1 molecules from the microtubule tip and lattice.

• Supplementary file 3. Summary table with model parameter comparisons between the EB1 on and off rates from the microtubule tip and lattice for the current study model, as compared to a model developed by *Maurer et al., 2014*. In addition, tubulin 'maturation rates' are compared, which define the EB1 binding zones in the Maurer et al model.

• MDAR checklist

### Data availability
All data generated or analyzed during this study are included in this manuscript and supporting files. Source data files have been provided for *Figures 1–6*.

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
