## [Editor Report]

This paper represents an important study for the microtubule cytoskeleton research community. By employing computational simulation, cell-free biophysical assays, and live-cell imaging, Gonzalez et al. convincingly reveal a mechanistic insight into the EB1 tip-tracking activity at the growing microtubule plus ends, preferential binding of GTP- over GDP-microtubule protofilaments does not fully explain the plus tip tracking of EB1. The authors show a binding preference of EB1 for protofilament edges over the closed lattice, which together with the nucleotide-state dependent dissociation rate of EB1 from the closed lattice successfully recapitulates the efficiency of EB1 tip tracking.

---

## [Decision Letter]

**Decision letter after peer review:**

[Editors’ note: the authors submitted for reconsideration following the decision after peer review. What follows is the decision letter after the first round of review.]

Thank you for submitting the paper "Rapid binding to protofilament edge sites facilitates tip tracking of EB1 at growing microtubule plus-ends" for consideration by *eLife*. Your article has been reviewed by 3 peer reviewers, and the evaluation has been overseen by a Reviewing Editor and a Senior Editor. The reviewers have opted to remain anonymous.

Comments to the Authors:

We are sorry to say that, after consultation with the reviewers, we have decided that this work will not be considered further for publication by *eLife*.

Our decision reflects the content of individual reviews and the outcome of the consultation session. All reviewers and the editor thought that the study was potentially important and could change the way how we think about the mechanism underlying the accumulation of End Binding (EB) proteins at growing microtubule plus ends, a topic of considerable interest to the cytoskeletal community. However, significant concerns were raised both about the experimental and computational aspects of the study. Given that this is the second manuscript developing the idea of EB proteins binding to protofilaments edges, compelling experimental evidence supporting the conclusions would be needed to make this study suitable for *eLife*.

1. Concerns about the experimental part. The conclusions strongly rely on the use of Eribulin, which the authors propose to bind to protofilament edges and directly interfere with EB binding to protofilament edges. The evidence for this is insufficiently compelling. First, contrary to what the authors claim (but do not illustrate anywhere in the manuscript), there seems to be no steric overlap between the binding sites of Eribulin (which binds to the longitudinal interface of β-tubulin) and EB CH domain (which binds predominantly to the lateral tubulin interfaces, see attached figure). The EM data included in the manuscript confirm the fluorescence microscopy data, previously published by Doodhi et al., that Eribulin binds to microtubule ends and not to shafts. These EM data do not have sufficient resolution to pinpoint the exact binding site. Another problem is that the effect of Eribulin on EB comets can be explained in different ways. Low concentrations of Eribulin, as well as most other microtubule-depolymerizing agents, indeed have limited effects on microtubule growth rates but trigger catastrophes, presumably by affecting microtubule tip structure. Changes in microtubule tip structure can affect EB binding. Therefore, a comparison of the effects of different concentrations of different compounds, with and without steric overlap with the EB binding site, could be a useful approach. For example, Halichondrin could be a drug candidate with a binding site strongly overlapping with that of EB.

2. Concerns about the computational part. While the simulations currently represent the strongest part of the manuscript, there were also some significant criticisms, as outlined in the individual reviews. In particular, the reviewers had questions about the sensitivity of the model output to the parameter values. They also thought that it would be important to prove that the simulations reproduce microtubule dynamics, including catastrophe frequencies, successfully, and consider alternative models of microtubule tip structure (with flared, rather than tapered protofilaments – a model that is gaining popularity in the field). The potential difference in the accumulation of EB monomers vs dimers also needs to be discussed.

Since addressing these concerns would require very significant efforts and their outcome appears uncertain at the moment, we return the paper to you. However, if you think that you can address all these concerns in full, we will be happy to reconsider this manuscript. It will then be treated as a new submission, but we will do our best to send it to the same reviewers. If you decide to resubmit to *eLife*, please provide a point-by-point rebuttal to all comments.

*Reviewer #1 (Recommendations for the authors):*

Gonzalez et al. have studied the molecular mechanism of end-tracking by EB proteins. In 2019, the Gardner lab published a study of end-tracking that used Brownian dynamics simulations to argue that EB binds more rapidly to "protofilament edge sites" (Reid et al. *eLife* 2019). Those simulations used static lattice structures and simulated the diffusion of EB into these edge sites. The present manuscript extends this line of inquiry with a simulation of dynamic microtubules that implements an accelerated rate of binding to protofilament edge sites. They show that the simulation matches experimental data for end-tracking, particularly with regard to the gap between the microtubule tip and the EB comet position (Figure 1C).

My first comment concerns the sensitivity of the model output to the parameter values. The authors write: "Even the most sensitive parameters had at least an 8-fold range of acceptable values", with the data shown in Figure S1. But I'm confused as to how this relates to Figure 2E and 2F, where end-tracking is lost when the edge-binding parameter is turned off. The lack of sensitivity that the authors state early in the manuscript seems in conflict with a lot of the rest of the paper, where they adjust parameters and show that the model breaks. Perhaps it's because I'm having difficulty relating the actual parameter values used in the model with the ranges used in Figure S1, for example. However, elsewhere in the paper, they say "the simulation predicts that reducing the protofilament-edge on-rate by 4-fold will lead to a dramatic loss of Mal3-GFP intensity at the tips of dynamic microtubules". So: does a 4-fold change in a parameter kill the model or is there an 8-fold range at which everything is fine? The authors need to clarify which parameters of the model are important.

My second comment about the model is that there is no validation that it reproduces microtubule dynamics successfully, although the simulation is well established in the Gardner lab so I'm sure they have considered these issues. But importantly: does the simulation accurately reproduce catastrophes? Presumably, the catastrophe frequency is related to the hydrolysis rate constant, and the hydrolysis rate constant will determine the relative size of the GTP-cap. Presumably, the size of the GTP cap is significant for the model's performance, especially for the relative significance of the closed-lattice on-rate vs. edge on-rate. If I understand correctly, if there is a larger GTP zone, then a higher on-rate to the closed lattice will shift the EB signal further away from the microtubule end.

The simulation is validated by a few different types of experimental data, most notably experiments using Eribulin. The authors use a relatively low concentration of Eribulin, which does not reduce the microtubule growth rate, but which does, in their hands, cause a modest reduction in Mal3 end "tip specificity" (Figure 4C and Figure 5B). This data, while promising, is a relatively weak anchor point for their computational work at this time. Only one Eribulin concentration is used in each experiment (80 nM for the in vitro work, 50 nM for the work in cells). In comparison, Doodhi et al. went as high as 250 nM Eribulin. At these high concentrations, the microtubule growth rate starts to decrease, but presumably, this effect can also be understood within the context of their computational framework. If they observed a dose-dependence of the Eribulin response, their argument would be strengthened.

The authors claim that Eribulin blocks the EB site at protofilament edges. This point would be much clearer to the reader if the authors created a structured figure panel for their paper, e.g., one that highlights the residues that interact with Eribulin alongside the residues that interact with EB.

Lastly, the paper assumes a structure for the microtubule end that is consistent with the lab's previous work and with many people's ideas in the field, namely that the end is tapered. It's worth noting, however, that the structure of the end is not a settled manner, with the McIntosh lab and their collaborators taking a decidedly different view of the end. While McIntosh's flared growing ends would have lots of edge sites, it's the lack of a taper that prevents a problem. Without some protofilaments being longer than others, the EB signal will not be displaced back from the end of the microtubule in the same way. The paper needs to address this issue for the reader so that a less-experienced reader (e.g., an early graduate student) will not have a false sense of a settled issue. Could a McIntosh model for the microtubule end make sense in terms of EB end-tracking as these authors understand it?

The raw data on the EM is very close-cropped (Figure 3B), so it's hard to see if the gold particles are consistently edge-bound or if the examples are just a lucky few where the gold particle happened to be near the side.

The Introduction includes a "reference dump", in which a single sentence is followed by a large number of references (in this case, 12). I sympathize with the desire to cite all of our colleagues, but I consider such reference dumps to be suboptimal because the reader does not really know why each paper is being cited.

*Reviewer #2 (Recommendations for the authors):*

This manuscript aims to explore and understand the mechanisms by which EB1-family proteins achieve their characteristic pattern of end-recognition. The work rests heavily on kinetic simulations but also incorporates experimental data to support assumptions and/or validate predictions. I found the work to be interesting. I think it is most convincing in its demonstration that differences in binding to 'complete' GTP- vs GDP-lattice sites cannot recapitulate observed aspects of EB comets – some end-specific recognition features are required. The authors postulate a particular kind of end-specific feature ('edge sites'), but it seems others might be possible. Some moderation in language and/or more explicit acknowledgment that other end-specific features may be operating might be helpful in this regard (and would not detract from the interest of the work).

The use of kinetic simulations is a strength of the work because it allows the authors to directly test different assumptions, and explore alternative models. The computational work is generally well-done, and it was particularly helpful to see results across a range of parameter values. The conclusion that distinguishing between 'closed' GTP- and GDP- lattice sites is not sufficient to recapitulate plus-end tracking is also interesting and considered a strength. The main weakness concerns whether the Eribulin data can be interpreted in the way the authors state. Additional weaknesses include a too-brief description of the modeling in the main text and too little quantitative engagement with prior work on EB comets.

The authors state that Eribulin can interfere with the EB binding site. My understanding from the Doodhi et al. paper cited is that Eribulin binds the plus-end of ab-tubulin and when bound at the end of a protofilament effectively blocks its elongation. I think at the very least the authors should add a figure panel to show a model of the eribulin and EB binding sites, to put things into structural context and provide better support for the statements that eribulin can bind to protofilament edge sites. An alternative view might be that Eribulin is doing something to change the shape of the microtubule end or the conformation of tubulin near the microtubule end, and these latter changes are influencing EB binding. Because the eribulin data provide the main experimental support for the claims that emerge from the model, this is an important aspect of the manuscript that needs some shoring up.

The essence of the underlying polymerization model is described in one sentence in the main text ("The tubulin assembly portion …"). This is too brief. The authors should expand the description somewhat to make the models and their assumptions more obvious for someone not interested in jumping to the methods section. It would also be nice to have some cartoons illustrating what sorts of end structures their simulations are generating (how tapered are they and is there detectable protofilament splaying), and how the model parameters relate to other models such as those previously used in the Gardner lab. For example, koff(GTP)/kon = 16 nM if I calculated correctly – does that correspond to a longitudinal interaction? If so, the affinity is rather strong relative to other models in the literature.

Finally, it would be helpful for the authors to more explicitly interpret their explicit simulations in light of simpler models like those proposed in the Maurer et al. work from the Surrey group, in which a relatively simple kinetic scheme could recapitulate observed features of EB comets. Can the authors make some more or less quantitative comparison between their results and these prior simpler schemes, both in terms of the basic reactions but also the quantitative parameters used in each model (association rates, for example)? Doing so would round out the manuscript and make it more appealing.

Overall I found the manuscript to be interesting – while on one hand, it might seem obvious to state that some end-specific binding feature is important for the end-localization of EB, much of the structural explanation for EB has focused on differences between GTP and GDP lattices, which the authors show is not sufficient.

I have two additional questions.

First – would the authors consider softening or doing more explaining around 'protofilament edge sites' and what that might encompass? It's a very specific phrase and made me wonder whether other end-specific features (partial curvature, say) might also suffice to give good-looking EB-localization in simulations. Basically, the authors are postulating an awfully specific mechanism given the supporting experimental data. So, I think it would be good to discuss this more, possibly raising (or even ruling out) alternatives. Do they think their results are general in the sense that they might also apply to CAMSAP proteins at the minus end?

Second – if EB associates more slowly to 'closed' sites on the lattice, should tubulin associate more slowly to EB-occupied 'edge sites', or are those closing events mainly happening by the kind of 'isomerization' reaction mimicking protofilament:protofilament pairing? These might be useful issues to add to a more fleshed-out description of the model and what it does and does not encompass.

The authors might also consider making their summary figure (currently 5F) a new standalone. I thought its impact was diminished by being combined with cellular data.

*Reviewer #3 (Recommendations for the authors):*

The authors investigate the mechanism by which tip tracking proteins EB recognize and bind microtubule tips. Earlier simulations from this group suggest that EB binds much faster at the edge of the microtubule where the lattice is not yet fully formed because reduced steric hindrance allows faster and easier landing of diffusing EBs on microtubule binding sites. Authors propose that if this acceleration in binding is more significant than the acceleration of detachment from these sites (which would also always happen because the site is not complete), the overall recruitment to the edge is more efficient than the recruitment to the closed GTP lattice itself.

Thus, the authors propose that in growing microtubules binding of EB occurs predominantly at the edge. As the microtubule elongates, these EB molecules get incorporated into the lattice of the GTP cap and detach when the lattice changes from GTP to GDP.

To test this idea, the authors use clever experiments. First, they show that the drug Eribulin recognizes incomplete (edge) EB binding sites and competes with EB for binding. Moderate concentrations of Eribulin do not reduce the microtubule growth rate but do reduce the relative number of EBs on the tips. This suggests that at least partially binding to the edge does facilitate EB loading to the microtubule tips. Authors take this a step further and argue that it is in fact always the edge where EBs bind and binding directly to the GTP cap does not play any significant role. To show this, the authors use simulations. They find that at a specific set of parameters binding of EBs at the edge can reproduce observed microscopic distributions of EBs on microtubule tips and predict that their experiments are fully explained by EB binding to the edge only.

I find experiments quite solid. I also find that the model needs improvement before it can explain events at the microtubule tips as it doesn't explain some of the most fundamental EB tip tracking properties. Therefore, using the simulations to prove that it is only the edge of the microtubule where EBs bind doesn't seem too convincing. Here are more detailed comments:

1. Simulations have many parameters. It is important to understand which parameters are estimated from experimental data and which are variables. Uncertainties in parameters and which parameters are more important and which are less should be better explained. For example, the ability of EB to bind better to the edge, critical for the conclusions of the paper, is the result of two rates. The on-rate, which is increased ~ 70 times, and the off-rate, which is increased ~10 times. Where did the latter number come from and what is the associated uncertainty? If it was close to 70, there would be no overall difference between the binding to the edge or binding directly to the cap. It should also be clarified for the rate related to the closed-lattice.

2. The model presented in the text and summarized in Figure 5F proposes how monomers of EB can track microtubule tips. However, there is a number of very convincing studies showing that monomers in fact cannot track microtubule tips. EB has to be a dimer to be able to recognize and track the tip. For example, if you dissociate dimers in real-time, they can no longer track microtubule tips (https://doi.org/10.1038/s41556-017-0028-5). It is confusing that authors first find parameters that would allow monomers to tip track and validate their simulations made for monomers using the experimental data, which should represent the behaviour of dimers. It makes validation arguably difficult. Before the model can be used to make predictions about where exactly EBs bind, it should be able to explain why EB monomers do not track microtubule tips and how EB dimers do. This seems like a big difference, so it is difficult to see if or how this more realistic model would lead to the same interpretation of the experimental data.

3. The simulations that show that just the edge binding alone is sufficient to account for the profiles of microtubules observed in microscopy experiments need to be better explained. We do know that GTP caps can be long (e.g. https://doi.org/10.7554/*eLife*.51992) and in growing microtubules, there should be a lot more EBs sitting on the GTP lattice as compared to the number of EBs sitting on the edge simply because there are more closed-lattice sites regardless of how EB ends up there. Therefore, the shape of the experimental profile should have a much stronger contribution from the EBs sitting on the closed lattice as opposed to those sitting on the edge. If this is true, why would simulations explain the data only assuming zero closed-lattice binding and not direct binding to the GTP cap? What about the opposite experiments? It is very likely, that one could find a set of closed-lattice off-rates that would explain experimental data by assuming only direct binding to the closed lattice and no binding to the edge whatsoever. Can these explain the experimental results?

4. One prediction from only edge binding may be that microtubules growing in the presence of GTPgS should have very specific EB comets. Since incorporation at the edge is expected to be the same, the brightness at the tips should be the same as for GTP microtubules, but the comet should be significantly longer and tail off at a specific distance as the closed-lattice off rate should remain that of GTP. However, if it is only closed-lattice binding there should be no specific comet seen on GTPgS microtubules. Maybe the EB profile in these experiments can be used to extract exactly how much binding can be attributed to the lattice and how much to the edge?

5. In growing microtubules majority of EBs are expected to be at the closed-lattice of the GTP cap simply because the number of these sites should be higher than the number of the edge sites. Let's say it is 10%, 50%, or 100% of EBs that sit on the closed-lattice are incorporated by the edge binding and the rest by direct GTP closed-lattice binding. Would that have an impact on the regulation of microtubule dynamic instability of other tip interactions? Are there any other potential implications?

[Editors’ note: further revisions were suggested prior to acceptance, as described below.]

Thank you for resubmitting your work entitled "Rapid binding to protofilament edge sites facilitates tip tracking of EB1 at growing microtubule plus-ends" for further consideration by *eLife*. Your revised article has been evaluated by Amy Andreotti (Senior Editor) and a Reviewing Editor.

The manuscript has been improved but there are some remaining issues that need to be addressed, as outlined below:

Essential revisions:

1) Provide a detailed analysis/simulation of the split Mal3 comets

*Reviewer #3 (Recommendations for the authors):*

Gonzalez et al. employ an interdisciplinary approach to dissecting the molecular mechanism by which EB1 tracks the growing microtubule plus ends. In particular, the authors propose that the rapid binding to a special feature, the 'protofilament edge' and the differential binding affinity for the close lattice in GTP or GDP state facilitates efficient tip tracking activity of EB1 at growing microtubule ends. Solid experiment data support the computational simulation. As the authors have thoroughly addressed the reviewers' questions, I only have a few comments that might further improve the clarity.

1. A more detailed analysis/simulation of the split Mal3 comets

The split EB1 comets (Figure 3) are a good opportunity to test the 'protofilament edge-binding' model. The authors quantify the summed intensity of Mal3 and show an ~80% increase in the split comets, supporting additional protofilament-edge binding sites at the growing microtubules with split comets. However, as the split comets are usually quite well separated, it is counterintuitive that the continuously exposed 'protofilament edge' can cause the split comets. Is it possible to simulate the split comets? Also, it appears that the split comet in Figure 3A tracks the depolymerizing microtubules. Is it common? What is the possible explanation?

2. The mechanism by which EB1 peak is behind the very tip of microtubules.

As EB1 binds to the protofilament edge with a 5~7-fold higher affinity than to the close lattice, the location of the EB1 peak seems dependent on the protofilament density (either tapered or flared). Have the authors examined the EB1 tip tracking on microtubules with different end structures? For example, how would the EB1 comet look on microtubules with blunt but flared ends?

3. When I read the manuscript, I wondered how this current model could improve our understanding of the EB1 tip-tracking activity in the context of the model proposed by Maurer et al. 2014. From my point of view, the major conceptual advance is that the rapid binding to the 'protofilament edge' can explain the behaviors of EB1 at the growing microtubule ends without introducing an 'exclusion zone' as proposed in Maurer's model. The authors should compare Maurer's model earlier in the manuscript rather than later in the discussion.

---

## [Author Response]

[Editors’ note: the authors resubmitted a revised version of the paper for consideration. What follows is the authors’ response to the first round of review.]

Comments to the Authors:We are sorry to say that, after consultation with the reviewers, we have decided that this work will not be considered further for publication by eLife.Our decision reflects the content of individual reviews and the outcome of the consultation session. All reviewers and the editor thought that the study was potentially important and could change the way how we think about the mechanism underlying the accumulation of End Binding (EB) proteins at growing microtubule plus ends, a topic of considerable interest to the cytoskeletal community. However, significant concerns were raised both about the experimental and computational aspects of the study. Given that this is the second manuscript developing the idea of EB proteins binding to protofilaments edges, compelling experimental evidence supporting the conclusions would be needed to make this study suitable for eLife.1. Concerns about the experimental part. The conclusions strongly rely on the use of Eribulin, which the authors propose to bind to protofilament edges and directly interfere with EB binding to protofilament edges. The evidence for this is insufficiently compelling. First, contrary to what the authors claim (but do not illustrate anywhere in the manuscript), there seems to be no steric overlap between the binding sites of Eribulin (which binds to the longitudinal interface of β-tubulin) and EB CH domain (which binds predominantly to the lateral tubulin interfaces, see attached figure). The EM data included in the manuscript confirm the fluorescence microscopy data, previously published by Doodhi et al., that Eribulin binds to microtubule ends and not to shafts. These EM data do not have sufficient resolution to pinpoint the exact binding site.

We agree that there seems to be no direct overlap between Eribulin and the EB CH domain. While diffusional hindrance to binding may explain why Eribulin acts to limit EB binding to protofilament-edge sites, even though there was perhaps not direct EB binding site overlap, we felt that this more in-depth topic would likely be better explored as future work. Therefore, we have now performed entirely new experiments, as follows:

1) We have employed a Designed Ankyrin Repeat Protein (DARPin) that exclusively targets protofilament-edge sites, and has direct overlap with the EB binding site (See Figure 4A) (Pecqueur et al., 2012). Consistent with previous reports, we found that DARPin blocked protofilament-edge sites on GTP-analogue stabilized microtubules (new Figure 4). Correspondingly, DARPin suppressed Mal3 tip tracking in cell-free dynamic microtubule experiments (Figure 5). Finally, DARPin suppressed EB1 tip tracking in cells (new Figure 6).

2) We have leveraged the appearance of “split comets” in our control experiments to further test the idea that EB1 binds rapidly to protofilament-edge sites, which increases the efficiency of tip tracking. In the canonical model in which EB1 tip tracking relies exclusively on an increased affinity for GTP tubulin relative to GDP tubulin, it is expected that, for a single microtubule growth event with a constant growth rate (and thus a constant GTP-cap size), the summed intensity of EB1 along a “split” comet would be equal to the summed intensity of EB1 in an intact comet. However, in a model with preferential EB1 binding to protofilament-edge sites, we predicted that the additional protofilamentedge binding sites in split comets, afforded by a large difference in protofilament lengths at the tip of growing microtubules, would lead to a net increase in the summed intensity of EB1. Consistent with this model, we observed a ~80% increase in summed Mal intensity on split comets relative to single comets (see Figure 3)

Another problem is that the effect of Eribulin on EB comets can be explained in different ways. Low concentrations of Eribulin, as well as most other microtubule-depolymerizing agents, indeed have limited effects on microtubule growth rates but trigger catastrophes, presumably by affecting microtubule tip structure. Changes in microtubule tip structure can affect EB binding. Therefore, a comparison of the effects of different concentrations of different compounds, with and without steric overlap with the EB binding site, could be a useful approach. For example, Halichondrin could be a drug candidate with a binding site strongly overlapping with that of EB.

Halichondrin is a natural drug, for which Eribulin is the synthetic substitute. Despite extensive efforts, we were not able find or purchase Halichondrin for our studies. However, we have now performed experiments over a range of DARPin concentrations. In addition, we have measured the effect of DARPin on microtubule growth rates, time to catastrophe, and microtubule tip structure. Finally, to test our hypothesis in the absence of other proteins, we have quantified Mal3 tip tracking intensity for “split comets”, which occur as a result of changes in tip structure, as described above (Figure 3). All details are described below, in response to individual reviewer concerns.

2. Concerns about the computational part. While the simulations currently represent the strongest part of the manuscript, there were also some significant criticisms, as outlined in the individual reviews. In particular, the reviewers had questions about the sensitivity of the model output to the parameter values. They also thought that it would be important to prove that the simulations reproduce microtubule dynamics, including catastrophe frequencies, successfully, and consider alternative models of microtubule tip structure (with flared, rather than tapered protofilaments – a model that is gaining popularity in the field). The potential difference in the accumulation of EB monomers vs dimers also needs to be discussed.

As described below, in response to individual reviewer comments, we have thoroughly addressed each of the reviewer questions.

Reviewer #1 (Recommendations for the authors):Gonzalez et al. have studied the molecular mechanism of end-tracking by EB proteins. In 2019, the Gardner lab published a study of end-tracking that used Brownian dynamics simulations to argue that EB binds more rapidly to "protofilament edge sites" (Reid et al. eLife 2019). Those simulations used static lattice structures and simulated the diffusion of EB into these edge sites. The present manuscript extends this line of inquiry with a simulation of dynamic microtubules that implements an accelerated rate of binding to protofilament edge sites. They show that the simulation matches experimental data for end-tracking, particularly with regard to the gap between the microtubule tip and the EB comet position (Figure 1C).My first comment concerns the sensitivity of the model output to the parameter values. The authors write: "Even the most sensitive parameters had at least an 8-fold range of acceptable values", with the data shown in Figure S1. But I'm confused as to how this relates to Figure 2E and 2F, where end-tracking is lost when the edge-binding parameter is turned off. The lack of sensitivity that the authors state early in the manuscript seems in conflict with a lot of the rest of the paper, where they adjust parameters and show that the model breaks. Perhaps it's because I'm having difficulty relating the actual parameter values used in the model with the ranges used in Figure S1, for example. However, elsewhere in the paper, they say "the simulation predicts that reducing the protofilament-edge on-rate by 4-fold will lead to a dramatic loss of Mal3-GFP intensity at the tips of dynamic microtubules". So: does a 4-fold change in a parameter kill the model or is there an 8-fold range at which everything is fine? The authors need to clarify which parameters of the model are important.

We agree that the sensitivity of the model parameter values was unclear in the previous version of the manuscript, and the wording was confusing. To simplify and clarify key model parameters, we have now established two key dimensionless variables that fundamentally control tip tracking in the model, as follows:

the ratio of on-rates of EB1 molecules at protofilament-edge sites relative to closed lattice sites, which is independent of the hydrolysis state of the associated tubulin molecules, andthe ratio of off-rates of EB1 molecules from closed-lattice GDP-tubulin sites relative to closed – lattice GTP-tubulin sites.

These two dimensionless variables control EB1 tip tracking in the model. Therefore, to succinctly address parameter sensitivity in the model, we have added three new panels to main text Figure 2 (see Figure 2G-I). These panels quantitatively demonstrate how changes in these two key dimensionless variables alter tip tracking in the model, the importance of these dimensionless variables relative to each other in the model, and, finally, a comparison of the dimensionless variable values to experimentally measured values. Following is the updated manuscript text regarding the new parameter sensitivity dimensionless variables (p. 12):

“To quantitatively interrogate the model parameter sensitivity, we defined two key dimensionless variables that control tip tracking in the model. First, as described above, the ratio of the on-rate of EB1 at protofilament-edge sites relative to closed-lattice sites, which is independent of the hydrolysis state of the associated tubulin molecules, directly alters the EB1 tip tracking efficiency in the model (Figure 2G). Importantly, clear tip tracking was observed using the experimentally measured on-rate ratio for protofilament-edge sites relative to closed-lattice sites (50-100:1, (Reid et al., 2019)) (Figure 2G, image B, grey dashed boxes).

Second, as has been previously described, the ratio of the off-rate of EB1 from closed-lattice GDP-tubulin sites, relative to closed-lattice GTP-tubulin sites, also influenced EB1 tip tracking in the model (Figure 2H; note that the model is comparatively insensitive to protofilament-edge off-rates, regardless of hydrolysis state Figure S2G-I, Figure S3D-F). Similar to the on-rate ratio, clear tip tracking was observed using the experimentally measured off-rate ratio for GDP-tubulin relative to GTP-tubulin (calculated as 6-12, based on K_D_ values reported in (Maurer et al., 2011)) (Figure 2H, image B, grey dashed boxes).

Finally, we evaluated the relative importance of the two dimensionless variables, one that dictates relative EB1 on-rates, and the other that dictates relative EB1 off-rates, in influencing simulated EB1 tip tracking (Figure 2I). We found that, in the absence of protofilament-edge binding, the experimentally observed range of closed-lattice GDP:GTP off-rate ratios did not robustly reproduce EB1 tip tracking (Figure 2I, top: typical images; bottom: blue bars). However, by including a 50:1 protofilament-edge to closedlattice on-rate ratio in the simulation, robust tip tracking was reproduced, with an increase in EB1 tip localization for a higher ratio of GDP:GTP off-rates (Figure 2I, red). Thus, both a hydrolysis-state dependent EB1 off-rate, as well as a rapid protofilament-edge EB1 on-rate, contribute to EB1 tip tracking in the model. However, the addition of rapid protofilament-edge on-rates increased the efficiency and robustness of simulated EB1 tip tracking. “

My second comment about the model is that there is no validation that it reproduces microtubule dynamics successfully, although the simulation is well established in the Gardner lab so I'm sure they have considered these issues. But importantly: does the simulation accurately reproduce catastrophes? Presumably, the catastrophe frequency is related to the hydrolysis rate constant, and the hydrolysis rate constant will determine the relative size of the GTP-cap. Presumably, the size of the GTP cap is significant for the model's performance, especially for the relative significance of the closed-lattice on-rate vs. edge on-rate. If I understand correctly, if there is a larger GTP zone, then a higher on-rate to the closed lattice will shift the EB signal further away from the microtubule end.

We have now recorded the time to catastrophe in the simulation (see Figure S1E). While the time to catastrophe is in the range of our experimentally observed values, catastrophe events occur somewhat more rapidly in the simulation as compared to experiment. Thus, the GTP-cap is likely not larger than would be experimentally observed, and we do not expect that the EB signal is shifted further away from the microtubule end due to a larger GTP zone in the simulation. We note that the catastrophe time is similar whether or not EB1 is present in the model, which is to be expected as EB1 in the model does not currently alter the hydrolysis rate or any other tubulin assembly parameters in the model. Finally, we have now verified that, under conditions of spontaneous catastrophe, the simulated EB1 molecules dissociate from shortening microtubule tips, as has been observed experimentally (Figure 1B).

The simulation is validated by a few different types of experimental data, most notably experiments using Eribulin. The authors use a relatively low concentration of Eribulin, which does not reduce the microtubule growth rate, but which does, in their hands, cause a modest reduction in Mal3 end "tip specificity" (Figure 4C and Figure 5B). This data, while promising, is a relatively weak anchor point for their computational work at this time. Only one Eribulin concentration is used in each experiment (80 nM for the in vitro work, 50 nM for the work in cells). In comparison, Doodhi et al. went as high as 250 nM Eribulin. At these high concentrations, the microtubule growth rate starts to decrease, but presumably, this effect can also be understood within the context of their computational framework. If they observed a dose-dependence of the Eribulin response, their argument would be strengthened.

We have shifted to examining a Designed Ankyrin Repeat Protein (DARPin) that exclusively targets protofilament-edge sites, and has direct overlap with the EB binding site (Pecqueur et al., 2012). In the updated manuscript, we have now performed experiments over a range of DARPin concentrations to observe a concentration-dependent reduction in Mal3 tip tracking (Figure 5). In addition, we show that the suppression of tip tracking at 1 μM DARPin is more substantial than would be predicted based on the small reduction in microtubule growth rate (Figure 5). Finally, we have now added a more readily accessible simulation prediction figure to complement these results (Figure 5). The manuscript text is as follows (p. 15-17):

“we ran simulations to quantitatively predict how EB1 tip tracking would be altered by suppressing its protofilament-edge on-rate (Figure 5A, left). Thus, we gradually reduced the protofilament-edge on-rate and generated simulated images to detect the relative localization of EB1-GFP at growing microtubule plus-ends (Figure 5A, center). To evaluate EB1-GFP localization to growing microtubule plusends in the simulation, we measured the EB1-GFP “Tip Specificity”. Here, we defined Tip Specificity (S) as:

S=(Itip−Ibackground)(Ilattice−Ibackground)(1)

Where I_tip_ is the EB1 intensity at the growing microtubule tip, I_lattice_ is the EB1 intensity on the microtubule lattice, and I_background_ is the EB1 intensity just outside of the growing microtubule tip. By definition, a lower Tip Specificity value indicates that there is less efficient tip tracking. In addition, a Tip Specificity value equal to one (e.g., S=1) means that the EB1 intensity at the growing microtubule tip is equal to the EB1 intensity along the length of the microtubule, and therefore EB1 is not tip tracking. We found that, in the simulation, Tip Specificity was correlated with the protofilament-edge on-rate. Specifically, a ~2-fold reduction in protofilament-edge on-rate led to a ~25% reduction in Tip Specificity (Figure 5A, right, grey dotted lines).

Thus, to test this simulation prediction, we performed a cell-free assay in which dynamic microtubules were grown from stabilized seed templates in the presence of Mal3-mCherry (Figure 5B, left). We visualized the growing microtubules using TIRF microscopy, in the presence of increasing concentrations of DARPin (Figure 5B, center). We found that Mal3 tip tracking was increasingly disrupted as the DARPin concentration was increased (p<<0.001, Kruskal Wallis) (Figure 5B, right). Interestingly, 1 μM DARPin led to a ~25% reduction in Tip Specificity, consistent with the simulation prediction of a 2-fold reduction in protofilament-edge on-rate (Figure 5B, right, grey dotted line).

We then asked whether the suppression of tip tracking in DARPin could be due to a drop in microtubule growth rate, leading to a reduced concentration of GTP-tubulin at the growing microtubule plus-end (Farmer et al., 2021; Maurer et al., 2014; Reid et al., 2019). We found that the suppression of tip tracking was more substantial than would be predicted based on the small changes in microtubule growth rate at 1 μM DARPin (Figure 5C, left, blue dotted line: control, purple markers: 1 μM DARPin). Furthermore, we found no significant increase in the time to catastrophe with increasing DARPin concentrations, suggesting that DARPin does not affect the GTP hydrolysis rate or the associated GTP-cap size (Figure 5C, center, Tukey’s post-hoc test).

Finally, we asked whether DARPin could indirectly disrupt Mal3 tip tracking by altering the configuration of the growing microtubule plus-end. Here, a more blunt microtubule tip structure could reduce the number of available protofilament-edge sites, and thus indirectly disrupt tip tracking. In contrast, a more extended, tapered tip structure would naturally allow for increased numbers of protofilament-edge sites, similar to the split comet phenotype as described above (Figure 3A), which increased Mal3 targeting to the growing microtubule tip. We found that 1 μM DARPin led to a ~40% increase in tip tapering at the growing microtubule end, which reflects a moderate increase in available protofilament-edge sites (C. E. Coombes et al., 2013; Demchouk et al., 2011). However, despite the increased availability of protofilament-edge sites, tip tracking was suppressed in DARPin (Figure 5B). Thus, DARPin does not suppress Mal3 tip tracking by indirectly reducing the number of available protofilament-edge sites. Rather, Mal3 is likely excluded from the protofilament-edge sites that are occupied by DARPin, which in turn suppresses tip tracking.”

The authors claim that Eribulin blocks the EB site at protofilament edges. This point would be much clearer to the reader if the authors created a structured figure panel for their paper, e.g., one that highlights the residues that interact with Eribulin alongside the residues that interact with EB.

While Eribulin binds protofilament-edge sites, it may not directly overlap the binding site of EB1. Thus, we decided to shift to DARPin D1, which binds protofilament-edge sites, and whose binding site more clearly overlaps that of EB1 (Maurer et al., 2014; Pecqueur et al., 2012; Pettersen et al., 2021). Further, we have now added a structured figure panel to the main text figures to illustrate this point (Figure 4A). Darpin directly overlaps with Mal3 at the growing microtubule ends based on crystal structures (4drx and 4abo) in ChimeraX (Maurer et al., 2012; Pecqueur et al., 2012; Pettersen et al., 2021).

Lastly, the paper assumes a structure for the microtubule end that is consistent with the lab's previous work and with many people's ideas in the field, namely that the end is tapered. It's worth noting, however, that the structure of the end is not a settled manner, with the McIntosh lab and their collaborators taking a decidedly different view of the end. While McIntosh's flared growing ends would have lots of edge sites, it's the lack of a taper that prevents a problem. Without some protofilaments being longer than others, the EB signal will not be displaced back from the end of the microtubule in the same way. The paper needs to address this issue for the reader so that a less-experienced reader (e.g., an early graduate student) will not have a false sense of a settled issue. Could a McIntosh model for the microtubule end make sense in terms of EB end-tracking as these authors understand it?

We have addressed this concern in two ways, as follows:

We have now performed additional simulations to approximate microtubule splaying in the model. Specifically, we assumed that, with a splayed end, all EB1 binding sites in front of the most distal lateral bond between two protofilaments would be considered a protofilament-edge site, in the absence of a tapered tip configuration. We found that the “microtubule splaying” approximation in the simulation had no discernible effect on tip tracking results (see Figure S1c). Further, we introduced excessive splaying into the simulation by moving the most distal lateral bond farther away from the growing microtubule tip, which led to increased EB1 targeting to the splayed growing microtubule plus-end (Figure S1c, right). This data has been added to the Results section on p. 7, as follows:

“It has been previously suggested that growing microtubule plus-ends could be “flared”, such that they have bent protofilaments that are curved (or flared) away from the central microtubule axis (McIntosh et al., 2018). Thus, we asked how a flared microtubule tip structure would affect tip tracking in our simulation. To approximate microtubule splaying in the model, we assumed that, with a splayed end, all EB1 binding sites in front of the most distal lateral bond between two protofilaments would be considered protofilament-edge sites. We found that the microtubule splaying approximation in the simulation had no discernible effect on tip tracking results (Figure S1C, left/center). Further, we introduced excessive splaying into the simulation by moving the most distal lateral bond farther away from the growing microtubule tip, which led to increased EB1 targeting to the splayed growing microtubule plus-end (Figure S1C, right).”

We have performed further analysis to better explain why the EB signal is displaced back from the end of the microtubule in the simulation. Despite the fact that EB molecules arrive quickly to protofilament-edge sites in the model, we found that the reason the peak EB1 location is slightly back from the distal tip of the microtubule is because there are substantially more EBs sitting on closed-lattice GTP sites, as compared to the number of EBs sitting on the protofilament-edge (Figure S1h). Specifically, in the previous manuscript version, we demonstrated that the peak EB1 location matched published literature (Figure 1C). However, we have now explicitly reported the fraction of EB1 molecules that are bound to GTP-Tubulin protofilament-edge sites, as compared to GTP-Tubulin closed-lattice sites, at any one time in the simulation (Figure S1H). Consistent with the idea that EB1 localization at the tip is primarily influenced by closed-lattice GTP-tubulin sites, we find that there are ~2-fold more EB1 molecules bound to closed-lattice GTP-tubulin sites, than to GTP-tubulin protofilament-edge sites, during the tiptracking simulation. We have incorporated this new data into the manuscript, as follows (p. 20-21):

“As described above, our model predicts that rapid protofilamentedge binding increases the efficiency of EB1 tip tracking. However, in the simulation, the peak EB1 location is slightly distal from the tip of the growing microtubule (Figure 1C), similar to previous reports (Maurer et al., 2014). We surmised that the peak EB1 location in the simulation is heavily influenced by EB1 molecules that are stably bound to GTP-tubulin closed-lattice sites on the growing microtubule tip. Indeed, by reporting the fraction of EB1 molecules that are bound to GTP-Tubulin protofilament-edge sites as compared to GTP-Tubulin closed lattice sites, we found that there are ~2-fold more EB1 molecules bound to closed-lattice GTP-tubulin sites, as compared to protofilament-edge sites, at any one time in the simulation (Figure S1H). “

The raw data on the EM is very close-cropped (Figure 3B), so it's hard to see if the gold particles are consistently edge-bound or if the examples are just a lucky few where the gold particle happened to be near the side.

We have now switched to using DARPin instead of Eribulin, and as a result, we have removed this figure.

The Introduction includes a "reference dump", in which a single sentence is followed by a large number of references (in this case, 12). I sympathize with the desire to cite all of our colleagues, but I consider such reference dumps to be suboptimal because the reader does not really know why each paper is being cited.

We have now carefully reviewed the references throughout the manuscript to ensure that papers are cited where appropriate.

Reviewer #2 (Recommendations for the authors):This manuscript aims to explore and understand the mechanisms by which EB1-family proteins achieve their characteristic pattern of end-recognition. The work rests heavily on kinetic simulations but also incorporates experimental data to support assumptions and/or validate predictions. I found the work to be interesting. I think it is most convincing in its demonstration that differences in binding to 'complete' GTP- vs GDP-lattice sites cannot recapitulate observed aspects of EB comets – some end-specific recognition features are required. The authors postulate a particular kind of end-specific feature ('edge sites'), but it seems others might be possible. Some moderation in language and/or more explicit acknowledgment that other end-specific features may be operating might be helpful in this regard (and would not detract from the interest of the work).

We have moderated language throughout the manuscript, and the following specific comments have been added to the discussion to address this caveat (p. 18):

“We note that, while we found that DARPin is not likely to indirectly suppress EB1 tip tracking by altering the plus-end tip conformation or the GTP-tubulin hydrolysis rate (Figure 5C), it remains possible that DARPin treatment could alter the conformation of tubulin that composes the microtubule tip to indirectly suppress EB1 tip tracking. Further, while the model leads to robust tip tracking by leveraging our previous experimental and simulation results that demonstrate the on-rate of EB1 molecules is rapid to protofilament-edge sites (Reid et al., 2019), we cannot exclude that another end-specific feature, not considered here, may be possible.”

The use of kinetic simulations is a strength of the work because it allows the authors to directly test different assumptions, and explore alternative models. The computational work is generally well-done, and it was particularly helpful to see results across a range of parameter values. The conclusion that distinguishing between 'closed' GTP- and GDP- lattice sites is not sufficient to recapitulate plus-end tracking is also interesting and considered a strength. The main weakness concerns whether the Eribulin data can be interpreted in the way the authors state. Additional weaknesses include a too-brief description of the modeling in the main text and too little quantitative engagement with prior work on EB comets.

A detailed response to each of these concerns is included below.

The authors state that Eribulin can interfere with the EB binding site. My understanding from the Doodhi et al. paper cited is that Eribulin binds the plus-end of ab-tubulin and when bound at the end of a protofilament effectively blocks its elongation. I think at the very least the authors should add a figure panel to show a model of the eribulin and EB binding sites, to put things into structural context and provide better support for the statements that eribulin can bind to protofilament edge sites.

Previous studies have found that Eribulin binds to protofilament-edge sites, however, a structural model suggested that Eribulin may not directly overlap the binding site of EB1. Thus, we decided to shift to a Designed Ankyrin Repeat Protein (DARPin) that exclusively targets protofilamentedge sites, and has direct overlap with the EB binding site (Pecqueur et al., 2012). We have now added a structured figure panel to the main text figures to illustrate this point (Figure R9, copied from Figure 4A). DARPin directly overlaps with Mal3 at the growing microtubule ends based on crystal structures (4drx and 4abo) in ChimeraX (Maurer et al., 2012; Pecqueur et al., 2012; Pettersen et al., 2021).

An alternative view might be that Eribulin is doing something to change the shape of the microtubule end or the conformation of tubulin near the microtubule end, and these latter changes are influencing EB binding. Because the eribulin data provide the main experimental support for the claims that emerge from the model, this is an important aspect of the manuscript that needs some shoring up.

While we have switched to DARPin rather than Eribulin, the comments regarding an alternative interpretation of our experimental results are still applicable. Thus, we have addressed concerns regarding an alternative interpretation of the DARPin experimental data in four ways, as follows:

One way in which EB1 tip tracking could be disrupted is if the DARPin increased the GTP-tubulin hydrolysis rate, and thus reduced the size of the GTP-cap at the growing microtubule plus-end. To address this concern, we performed experiments to measure the catastrophe frequency at the growing microtubule plus-end, in the presence of increasing DARPin concentrations. Here, we reasoned that if DARPin was reducing the size of the GTP-cap by increasing the hydrolysis rate, we would observe an increased catastrophe frequency with increased DARPin concentration. However, we did not observe a change in catastrophe frequency (Figure 5). The associated manuscript text is as follows (p. 16):

“We then asked whether the suppression of tip tracking in DARPin could be due to a drop in microtubule growth rate, leading to a reduced concentration of GTP-tubulin at the growing microtubule plus-end (Farmer et al., 2021; Maurer et al., 2014; Reid et al., 2019). We found that the suppression of tip tracking was more substantial than would be predicted based on the small changes in microtubule growth rate at 1 μM DARPin (Figure 5C, left, blue dotted line: control; purple markers: 1 μM DARPin). Furthermore, we found no significant increase in the time to catastrophe with increasing DARPin concentrations, suggesting that DARPin does not affect the GTP hydrolysis rate or the associated GTP-cap size (Figure 5C, center, Tukey’s post-hoc test).”

Secondly, since DARPin binds to protofilament-edge sites, we reasoned that DARPin could blunt the “tip taper” at the growing microtubule tip. Here, if DARPin reduced the overall number of protofilament-edge binding sites at the growing microtubule end by blunting its natural tip taper, this could indirectly suppress EB1 binding. Thus, we performed experiments to measure the taper at the growing microtubule tip see Figure 5C, as is described in the main text on p. 16, as follows:

“Finally, we asked whether DARPin could indirectly disrupt Mal3 tip tracking by altering the configuration of the growing microtubule plus-end. Here, a more blunt microtubule tip structure could reduce the number of available protofilament-edge sites, and thus indirectly disrupt tip tracking. In contrast, a more extended, tapered tip structure would naturally allow for increased numbers of protofilament-edge sites, similar to the split comet phenotype as described above (Figure 3A), which increased Mal3 targeting to the growing microtubule tip. We found that 1 μM DARPin led to a ~40% increase in tip tapering at the growing microtubule end, which reflects a moderate increase in available protofilament-edge sites (Figure 5C, right) (C. E. Coombes et al., 2013; Demchouk et al., 2011). However, despite the increased availability of protofilament-edge sites, tip tracking was suppressed in DARPin (Figure 5B). Thus, DARPin does not suppress Mal3 tip tracking by indirectly reducing the number of available protofilament-edge sites. Rather, Mal3 is likely excluded from the protofilament-edge sites that are occupied by DARPin, which in turn suppresses tip tracking.”

So that our experimental results did not rely exclusively on the DARPin results, we have now also leveraged the appearance of “split comets” in our control experiments. In the canonical model in which tip tracking relies exclusively on a higher EB1 affinity for GTP-tubulin relative to GDP-tubulin, it is expected that, for a single microtubule growth event with a constant growth rate (and thus a constant GTP-cap size), the summed intensity of EB1 along a split comet would be equal to the summed intensity of EB1 in an intact comet. However, in a model with preferential EB1 binding to protofilament-edge sites, we predicted that the additional protofilamentedge binding sites afforded by a large difference in protofilament lengths at the tip of growing microtubules with “split comets” would lead to a net increase in the summed intensity of EB1. Consistent with this model, we observed an ~80% increase in summed EB1 intensity on split comets relative to single comets (see Figure 3). The new manuscript text is as follows (p.12-13):

“It has been previously reported that EB1-GFP can split into multiple comets that track the growing microtubule end (Doodhi et al., 2016). Thus, a “split comet” refers to the phenomenon in which there are two (or more) distinct EB1 puncta that track a growing microtubule end (Doodhi et al., 2016; Farmer et al., 2021) (Figure 3A, right; orange arrow: pre-split; magenta/blue arrows: post-split). A split comet likely occurs when one or more protofilaments lag behind the growing microtubule tip, thus producing leading and lagging GTP-caps (Figure 3A, left-bottom). In the canonical model in which tip tracking relies exclusively on a higher EB1 affinity for GTP-tubulin relative to GDP-tubulin, it is expected that, for a single microtubule growth event with a constant growth rate (and thus a constant GTP-cap size), the total summed intensity of EB1-GFP at split-comet tips would be expected to be similar to the intensity of EB1-GFP at single-comet tips. However, in a model with preferential EB1 binding to protofilament-edge sites, we predicted that the additional protofilament-edge binding sites afforded by a large difference in protofilament lengths at the tip of growing microtubules with “split comets” would lead to a net increase in the summed intensity of EB1 (Figure 3A, left-bottom) (Farmer et al., 2021). Thus, if EB1 binds to protofilament-edge sites, we predicted that there would be an increase in the summed EB1-GFP intensity at growing microtubule tips with split comets, due to the increased number of protofilament-edge sites that are available to recruit EB1.

To test this idea, we examined growth events that had split comets, and measured the summed Mal3 (yeast EB1-homolog) intensity before and after the comet split on single growing microtubule (Figure 3B top: orange box: pre-split; Figure 3B middle: magenta/blue box: post-split). We subtracted the background across the same area before and after the comet split (Figure 3B, bottom). We found that split comets had an ~80% increase in the summed intensity of Mal3 at the growing microtubule tip relative to the single comets on the same microtubule growth events (Figure 3C, paired t-test, p<<0.0001). Thus, these results suggest that an increase in protofilament-edge sites during splitcomet growth events lead to an increase in EB1 recruitment to the microtubule plus-end.”

Finally, we agree that it is not possible to rule out every possible explanation for changes in tip tracking with DARPin treatment relating to the shape of the microtubule end or the conformation of tubulin that composes the microtubule tip. Therefore, we have added the following text to the discussion to address this issue (P. 18):

“We found that DARPin blocks EB1 binding at protofilament-edge sites on stabilized microtubules, and importantly, this blocking of EB1 binding to protofilament-edge sites led to a disruption of EB1 tip tracking in dynamic microtubule cell-free assays, and in cells. We conclude that the rapid binding of EB1 to protofilament-edge sites facilitates the tip tracking of EB1 at growing microtubule ends. We note that, while we found that DARPin is not likely to indirectly suppress EB1 tip tracking by altering the plus-end tip conformation or the GTP-tubulin hydrolysis rate (Figure 5C), it remains possible that DARPin treatment could alter the conformation of tubulin that composes the microtubule tip to indirectly suppress EB1 tip tracking. Further, while the model leads to robust tip tracking by leveraging our previous experimental and simulation results that demonstrate the on-rate of EB1 molecules is rapid to protofilament-edge sites (Reid et al., 2019), we cannot exclude that another end-specific feature, that was not considered here, may be possible.”

The essence of the underlying polymerization model is described in one sentence in the main text ("The tubulin assembly portion …"). This is too brief. The authors should expand the description somewhat to make the models and their assumptions more obvious for someone not interested in jumping to the methods section.

We have now expanded on the microtubule assembly portion in the main text, as follows (p. 5-6):

“The microtubule assembly portion of the simulation utilized a previously published model, in which individual tubulin subunits were allowed to arrive and depart from the growing microtubule plus-end (Margolin et al., 2011, 2012). Once a tubulin subunit arrived to the growing microtubule plus-end, a longitudinal bond was immediately formed with its penultimate tubulin dimer. Then, lateral bonds were stochastically formed in subsequent time steps (Margolin et al., 2011, 2012). Finally, lattice-incorporated GTP-tubulin subunits were stochastically hydrolyzed to GDP-tubulin. In general, the on-rate of new tubulin subunits to the microtubule plus-end depended on the simulated tubulin concentration, and the off-rate of an individual tubulin subunit from the plus-end depended on its hydrolysis state and bonding state, where a GTP-tubulin subunit with two lateral bonds had the lowest off-rate in the simulation. All of the parameter values for the microtubule assembly simulation matched a previously published parameter set (Margolin et al., 2012) (see Table S1), with the exception of (1) the tubulin on-rate constant, which was lowered in order to match our (slow) experimental growth rates, and (2) one additional rule was added to ensure that the tip taper at the microtubule plus-end matched our experimental values (Figure S1A,B). Here, if the difference between the longest and the penultimate shortest protofilament exceeded 600 nm (75 dimers), the tubulin subunit off-rate and the lateral bond breakage rate were dramatically increased, quickly leading to a catastrophe event.”

It would also be nice to have some cartoons illustrating what sorts of end structures their simulations are generating (how tapered are they and is there detectable protofilament splaying),

We have addressed this issue in the following two ways, to address both a cartoon addition and protofilament splaying:

1) To demonstrate the range of tip structures that are generated by the simulation, we have added a new figure (see Figure S1A,B), which illustrates a typical tip structure in the simulation, as well as a histogram of tip standard deviation values at the point in the simulation in which EB1 tip specificity measurements were taken. In addition, a new Video 2, which is an animation showing the typical tip structures during a growth event of one microtubule, has been added to the manuscript submission. Finally, we have added the following text to compare simulated and experimental tip tapering measurements (p. 7):

“To ensure that the configuration of the microtubule plus-end was similar between experiment and simulation, we compared the fitted tip standard deviation in simulated microtubule images to our experimental values. Here, the “tip standard deviation” reflects the range of protofilament lengths at the tip of the growing microtubule, such that a “tapered tip” would have a large tip standard deviation. We found that the average tip standard deviation of our simulated microtubules was 191 ± 6 nm (mean ± SEM), similar to our experimental measurements of 180 ± 17 nm (Figure S1A, B; Video 2, mean ± SEM).”

2) The current simulation is based on a 2D model, published by (Margolin et al., 2012). In this model, while lateral bonds are made independently of subunit addition (e.g., there is a delay between subunit addition to the end of the lattice, and the subunit making a lateral bond), the model does not explicitly incorporate protofilament splaying, as was observed in the 3D model previously used by the Gardner laboratory. For the EB1 simulation in the current work, a simplified, course grained model was used, due to the computational intensity of the full 3D microtubule dynamics model, which would have increased the computational load in the EB1 single-molecule tip tracking simulation. Therefore, by definition, the simulated tip structures do not show obvious protofilament splaying. However, we have now leveraged the delayed lateral bonding aspect of the microtubule dynamics model (Margolin et al., 2012) to perform additional simulations that approximate microtubule splaying in the model. Specifically, we assumed that, with a splayed end, all EB1 binding sites in front of the most distal lateral bond between two protofilaments would be considered a protofilament-edge site. We found that, by including the approximation of protofilament splaying in the model, there was no discernable effect on tip tracking results (see Figure S1C). Further, we introduced excessive splaying into the simulation by moving the most distal lateral bond farther away from the growing microtubule tip, which led to increased EB1 targeting to the splayed growing microtubule plus-end (Figure S1c, right). This data has been added to the Results section on p. 7, as follows:

“It has been previously suggested that growing microtubule plus-ends could be “flared”, such that they have bent protofilaments that are curved (or flared) away from the central microtubule axis (McIntosh et al., 2018). Thus, we asked how a flared microtubule tip structure would affect tip tracking in our simulation. To approximate microtubule splaying in the model, we assumed that, with a splayed end, all EB1 binding sites in front of the most distal lateral bond between two protofilaments would be considered protofilament-edge sites. We found that the microtubule splaying approximation in the simulation had no discernible effect on tip tracking results (Figure S1C, left/center). Further, we introduced excessive splaying into the simulation by moving the most distal lateral bond farther away from the growing microtubule tip, which led to increased EB1 targeting to the splayed growing microtubule plus-end (Figure S1C, right).”

and how the model parameters relate to other models such as those previously used in the Gardner lab. For example, koff(GTP)/kon = 16 nM if I calculated correctly – does that correspond to a longitudinal interaction? If so, the affinity is rather strong relative to other models in the literature.

As noted above, the model used in this work was as published by (Margolin et al., 2012). In order to clarify the model parameters and their origin, we have now added a new Table S1, in which all of the microtubule assembly model parameters are listed, along with a reference to establish their origin. All of the parameter values used in the current work match a parameter set in the Margolin et al. 2012 paper, with the exception of (1) the tubulin on-rate constant, which was lowered in order to match our (slow) experimental growth rates, and (2) the addition of three new parameters to establish a rule that limits excessive microtubule taper lengths. Here, if the difference between the longest and the penultimate shortest protofilament exceeded 600 nm (75 dimers), the tubulin off-rate and the lateral bond breakage rate were dramatically increased, quickly leading to a catastrophe event. This rule was implemented to ensure that the model did not depend on excessively large taper lengths to recapitulate EB1 tip tracking. Further, this rule also ensured that a relatively strong longitudinal interaction energy would not lead to excessively long tip taper lengths, as very long taper lengths could have the potential to bias our EB1 simulation results.

Finally, it would be helpful for the authors to more explicitly interpret their explicit simulations in light of simpler models like those proposed in the Maurer et al. work from the Surrey group, in which a relatively simple kinetic scheme could recapitulate observed features of EB comets. Can the authors make some more or less quantitative comparison between their results and these prior simpler schemes, both in terms of the basic reactions but also the quantitative parameters used in each model (association rates, for example)? Doing so would round out the manuscript and make it more appealing.

In the Maurer et al. work from the Surrey group, a constant length microtubule template was employed, with three EB1 binding “zones”. Our new model incorporates microtubule assembly dynamics, in addition to single-molecule EB1 on/off dynamics, and in which an EB1 binding “exclusion zone” is not employed. Rather, EB1 tip tracking develops naturally as a result of the EB1 binding rules for each individually simulated EB1 molecule. Regardless, our model was able to reproduce the offset in the peak EB1 localization away from the distal end of the microtubule, as was first described in the Maurer paper (Figure 1C). In order to directly compare the two models, a new table with parameter value comparisons was added to supplemental material (see Table S3), and we have also added extensive text to the discussion, as follows (p. 34):

“Previously, a model was developed to explain the peak position of EB1 on the growing microtubule tip, which is slightly offset from the distal tip of the growing microtubule (Figure 1C) (Maurer et al., 2014). Because both our currently described model and the previously described model were able to reproduce the localization of EB1 on the microtubule, we sought to compare and contrast the described mechanisms in each of the two models.

In the previously described Maurer et al. model (Maurer et al., 2014), a model was developed that relied on a constant length microtubule template with three EB1 binding “zones”. Here, explicit tubulin assembly dynamics were not included in the model, but rather a constant length microtubule template was employed, in which there was an EB1 binding “exclusion zone” at the tip of the microtubule. A tubulin subunit maturation rate was included in the model, which led to a second zone, slightly distal from the tip of the microtubule, in which EB1 binding was allowed. Finally, a second tubulin subunit maturation rate was employed, which led to a third zone, far from the tip and along the microtubule lattice, in which EB1 disassociation was allowed. The second tubulin subunit maturation rate likely corresponds to GTP-tubulin to GDP-tubulin hydrolysis, which is similar both in the magnitude of the hydrolysis rates employed, and in the EB1 off-rates employed, between the Maurer model and our newly described model (Table S3).

Thus, the primary difference between the two models was in the binding of EB1 to the microtubule lattice. Here, we predict that the key features of the Maurer model that involved exclusion of EB1 binding to the distal tip of the microtubule, along with a binding zone just behind the distal tip, are incorporated into our newly described model by the ability of newly arriving tubulin subunits to “lock in” protofilament-edge bound EB1 molecules into a stable 4-tubulin pocket. Specifically, in our new model, EB1 molecules arrive rapidly to easily accessible protofilament-edge sites at the growing tip of the microtubule (Figure 7, step 1). However, the number of protofilament-edge sites are few relative to closed lattice sites. Thus, the EB1-GFP signal at the distal tip of the microtubule remains low. Then, upon new tubulin subunit addition, protofilament-edge bound EB1 molecules are “locked in” to a stable, 4-GTPtubulin pocket (Figure 7, step 2), leading to a low EB1 off-rate, and thus a high concentration of EB1 on GTP-tubulin closed lattice sites, slightly distal from the growing microtubule tip. Thus, in our new model,

EB1 accumulates on the GTP-cap at the growing microtubule end, with a peak EB1 position slightly distal from the growing microtubule tip, as has been previously reported (Maurer et al., 2014; Roth et al., 2019). Finally, upon the hydrolysis of GTP-tubulin to GDP-tubulin, the affinity of EB1 for the GDP-tubulin subunits is reduced, and EB1 dissociates from the microtubule (Figure 7, step 3). Therefore, the higher affinity of EB1 for GTP relative to GDP tubulin also contributes to EB1 localization just distal to the growing microtubule ends (Figure 2G-I). In summary, we predict that the primary difference between our new model, and the previous Maurer et al. model, may be in (1) the inclusion of tubulin assembly dynamics, and (2) the rapid EB1 binding to protofilament-edge sites. These features eliminate the requirement for an explicit EB1 binding “exclusion zone” at the tip of the microtubule, and naturally lead to a decrease in signal at the distal tip of the microtubule.”

Overall I found the manuscript to be interesting – while on one hand, it might seem obvious to state that some end-specific binding feature is important for the end-localization of EB, much of the structural explanation for EB has focused on differences between GTP and GDP lattices, which the authors show is not sufficient.I have two additional questions.First – would the authors consider softening or doing more explaining around 'protofilament edge sites' and what that might encompass? It's a very specific phrase and made me wonder whether other end-specific features (partial curvature, say) might also suffice to give good-looking EB-localization in simulations. Basically, the authors are postulating an awfully specific mechanism given the supporting experimental data. So, I think it would be good to discuss this more, possibly raising (or even ruling out) alternatives.

We have addressed this concern in three different ways, as follows:

We have now performed additional simulations to approximate microtubule splaying and/or protofilament curvature in the model. Specifically, we assumed that, with a splayed end, all EB1 binding sites in front of the most distal lateral bond between two protofilaments would be considered a protofilament-edge site, without a tapered tip configuration. We found that the “microtubule splaying” approximation in the simulation had no discernible effect on tip tracking results see Figure S1c. Further, we introduced excessive splaying into the simulation by moving the most distal lateral bond farther away from the growing microtubule tip, which led to increased EB1 targeting to the splayed growing microtubule plus-end (Figure S1c, right). This data has been added to the Results section on p. 7, as follows:

“It has been previously suggested that growing microtubule plus-ends could be “flared”, such that they have bent protofilaments that are curved (or flared) away from the central microtubule axis (McIntosh et al., 2018). Thus, we asked how a flared microtubule tip structure would affect tip tracking in our simulation. To approximate microtubule splaying in the model, we assumed that, with a splayed end, all EB1 binding sites in front of the most distal lateral bond between two protofilaments would be considered protofilament-edge sites. We found that the microtubule splaying approximation in the simulation had no discernible effect on tip tracking results (Figure S1C, left/center). Further, we introduced excessive splaying into the simulation by moving the most distal lateral bond farther away from the growing microtubule tip, which led to increased EB1 targeting to the splayed growing microtubule plus-end (Figure S1C, right).”

We agree that the term “protofilament-edge” sounds quite specific, however, we felt that this description is perhaps most illustrative of the type of binding site that is most accessible by EB1. However, to address this concern, have now added text to the introduction to more clearly explain (and generalize) the term “protofilament-edge”, and to clarify the various potential EB1 binding configurations that are encompassed by this term, as follows (p. 4):

“Recent work has demonstrated that EB1 can bind to a partial binding pocket composed of 2-3 tubulin subunits, either at the distal tip of a protofilament, along the edge of an exposed protofilament, or at lattice openings within the microtubule (Reid et al., 2019). We describe these exposed, partial binding pockets as “protofilament-edge” sites. Specifically, we use the term “protofilament-edge” to describe any partial EB1 binding site on the microtubule lattice, as opposed to closed (4-tubulin) binding sites. Importantly, we recently reported that the arrival rate of EB1 to 2-tubulin protofilament-edge sites was ~70-fold faster than to closed 4-tubulin pockets, due to a reduced diffusional steric hindrance to binding (Reid et al., 2019). Here, a partial EB1 binding site on the microtubule lattice led to a dramatic reduction in the diffusional steric hindrance that EB1 encounters in order to become properly oriented and then to slide into a closed, 4-tubulin binding pocket. In other words, the expanded physical access that is afforded by EB1 binding to a partial, 2tubulin binding pocket (as compared to a closed 4-tubulin binding pocket) led to a ~70-fold increase in the EB1 on-rate. Because protofilament-edge sites are present at growing microtubule plus-ends (Atherton et al., 2018; Gudimchuk et al., 2020; Guesdon et al., 2016), we hypothesized that this large difference in EB1 arrival rates could have important repercussions for the efficiency of EB1 tip tracking at growing microtubule plus-ends. We thus predicted that the rapid binding of EB1 to protofilament-edge sites at the growing microtubule plus-end could increase the efficiency of EB1 plus-end tip tracking.”

Finally, we have added text to the discussion regarding alternative configurations that could contribute to EB1 binding, as follows (p. 21):

“Recent work has demonstrated that growing microtubule tips are less homogeneous than previously thought, such that growing microtubule tips exhibit a wide range of protofilament lengths between the leading and lagging protofilaments, both in cells and in cell-free experiments (Atherton et al., 2018; Cleary and Hancock, 2021; C. E. Coombes et al., 2013; Gudimchuk et al., 2020; Guesdon et al., 2016; Igaev and Grubmüller, 2022). Here, a wide range of protofilament lengths at the growing microtubule end would lead to increased numbers of protofilament-edge sites, which, in our model, would increase the EB1 on-rate to the tip. In addition, in our model, other tip configurations, such as partial curvature (Bechstedt et al., 2014; Farmer et al., 2021), or flaring of the growing microtubule plus-end (McIntosh et al., 2018), would also contribute to EB1 tip tracking (Figure S1C). Here, partial curvature or flaring requires opening of the closed microtubule tube – indicating that protofilaments or groups of protofilaments are separated from each other. Importantly, separation between protofilaments means that the number of protofilament-edge sites would be enriched, as new protofilament sides would be exposed. Thus, the role of protofilament-edge sites in facilitating EB1 tip tracking could apply to a wide range of growing microtubule tip configurations.”

Do they think their results are general in the sense that they might also apply to CAMSAP proteins at the minus end?

CAMSAP appears to bind between α- and β-tubulin subunits rather than along the exposed α-tubulin at the minus end (Atherton et al., 2017). Therefore, while CAMSAP could potentially bind to the sides of exposed protofilaments at the minus-end, it may not rapidly bind at the extreme distal tips of minus ends as a result of a diffusional steric hindrance model.

Second – if EB associates more slowly to 'closed' sites on the lattice, should tubulin associate more slowly to EB-occupied 'edge sites', or are those closing events mainly happening by the kind of 'isomerization' reaction mimicking protofilament:protofilament pairing? These might be useful issues to add to a more fleshed-out description of the model and what it does and does not encompass.

The tubulin dynamics model does not currently encompass any potential effects of tubulin binding to EB1-occupied edge sites, as the GTP-tubulin on-rate in our simulation is constant regardless of whether or not an EB1 is already bound. Further, our simple 2D tubulin assembly model does not account for protofilament-protofilament pairing. However, to address this concern, we have added a discussion of this caveat to the Discussion section, as follows (p. 18):

“The tubulin assembly portion of the model was built on earlier work, in which individual tubulin subunits were allowed to arrive and depart from the growing microtubule plus-end (Margolin et al., 2011, 2012) (see Methods). Future work will involve examining the effects of a slower tubulin association rate to EB1occupied protofilament edge sites, and whether EB1 binding to protofilament edge sites could assist in neighboring protofilament zippering on flared microtubule tips. Further, microtubule targeting drugs that suppress the kinetics of tubulin assembly at the growing microtubule plus-end, such as Taxol (Castle et al., 2017), could potentially disrupt EB1 tip tracking by slowing the capture and “lock in” of EB1 to 4tubulin pocket binding sites (Figure 7, step 2), an idea that could be explored in future work.”

The authors might also consider making their summary figure (currently 5F) a new standalone. I thought its impact was diminished by being combined with cellular data.

We appreciate the reviewer’s comment and have incorporated their feedback. This is now Figure 7.

Reviewer #3 (Recommendations for the authors):The authors investigate the mechanism by which tip tracking proteins EB recognize and bind microtubule tips. Earlier simulations from this group suggest that EB binds much faster at the edge of the microtubule where the lattice is not yet fully formed because reduced steric hindrance allows faster and easier landing of diffusing EBs on microtubule binding sites. Authors propose that if this acceleration in binding is more significant than the acceleration of detachment from these sites (which would also always happen because the site is not complete), the overall recruitment to the edge is more efficient than the recruitment to the closed GTP lattice itself.Thus, the authors propose that in growing microtubules binding of EB occurs predominantly at the edge. As the microtubule elongates, these EB molecules get incorporated into the lattice of the GTP cap and detach when the lattice changes from GTP to GDP.To test this idea, the authors use clever experiments. First, they show that the drug Eribulin recognizes incomplete (edge) EB binding sites and competes with EB for binding. Moderate concentrations of Eribulin do not reduce the microtubule growth rate but do reduce the relative number of EBs on the tips. This suggests that at least partially binding to the edge does facilitate EB loading to the microtubule tips. Authors take this a step further and argue that it is in fact always the edge where EBs bind and binding directly to the GTP cap does not play any significant role. To show this, the authors use simulations. They find that at a specific set of parameters binding of EBs at the edge can reproduce observed microscopic distributions of EBs on microtubule tips and predict that their experiments are fully explained by EB binding to the edge only.I find experiments quite solid. I also find that the model needs improvement before it can explain events at the microtubule tips as it doesn't explain some of the most fundamental EB tip tracking properties. Therefore, using the simulations to prove that it is only the edge of the microtubule where EBs bind doesn't seem too convincing. Here are more detailed comments:1. Simulations have many parameters. It is important to understand which parameters are estimated from experimental data and which are variables. Uncertainties in parameters and which parameters are more important and which are less should be better explained. For example, the ability of EB to bind better to the edge, critical for the conclusions of the paper, is the result of two rates. The on-rate, which is increased ~ 70 times, and the off-rate, which is increased ~10 times. Where did the latter number come from and what is the associated uncertainty? If it was close to 70, there would be no overall difference between the binding to the edge or binding directly to the cap. It should also be clarified for the rate related to the closed-lattice.

We have addressed this concern in three different ways, as follows:

1) The sensitivity of the parameter values, and, importantly, the identification of the parameters that are critical to tip tracking in the model, were unclear in the previous version of the manuscript. To simplify and clarify key model parameters, we have now established two key dimensionless variables that fundamentally control tip tracking in the model,:

The ratio of on-rates of EB1 molecules at protofilament-edge sites relative to closed lattice sites, which is independent of the hydrolysis state of the associated tubulin molecules, andThe ratio of off-rates of EB1 molecules from closed-lattice GDP-tubulin sites relative to closed –lattice GTP-tubulin sites.

These two dimensionless variables control EB1 tip tracking in the model. Therefore, to succinctly address parameter sensitivity in the model, we have added three new panels to main text Figure 2 (Figure 2G-I). These panels quantitatively demonstrate how changes in these two key dimensionless variables alter tip tracking in the model, the importance of these dimensionless variables relative to each other in the model, and, finally, a comparison of the dimensionless variable values to experimentally measured values. Following is the updated manuscript text regarding the new parameter sensitivity dimensionless variables (p. 12):

“To quantitatively interrogate the model parameter sensitivity, we defined two key dimensionless variables that control tip tracking in the model. First, as described above, the ratio of the on-rate of EB1 to protofilament-edge sites relative to closed-lattice sites, which is independent of the hydrolysis state of the associated tubulin molecules, directly alters the EB1 tip tracking efficiency in the model (Figure 2G). Importantly, clear tip tracking was observed using the experimentally measured on-rate ratio for protofilament-edge sites relative to closed-lattice sites (50-100:1, (Reid et al., 2019)) (Figure 2G, image B, grey dashed boxes).

Second, as has been previously described, the ratio of the off-rate of EB1 from closed-lattice GDPtubulin sites, relative to closed-lattice GTP-tubulin sites, also influenced EB1 tip tracking in the model (Figure 2H; note that the model is comparatively insensitive to protofilament-edge off-rates, regardless of hydrolysis state Figure S2G-I, Figure S3D-F). Similar to the on-rate ratio, clear tip tracking was observed using the experimentally measured off-rate ratio for GDP-tubulin relative to GTP-tubulin (calculated as 6-12, based on K_D_ values reported in (Maurer et al., 2011)) (Figure 2H, image B, grey dashed boxes).

Finally, we evaluated the relative importance of the two dimensionless variables, one that dictates relative EB1 on-rates, and the other that dictates relative EB1 off-rates, in influencing simulated EB1 tip tracking (Figure 2I). We found that, in the absence of protofilament-edge binding, the experimentally observed range of closed-lattice GDP:GTP off-rate ratios did not robustly reproduce EB1 tip tracking (Figure 2I, top: representative simulated images; bottom: blue bars). However, by including a 50:1 protofilament-edge to closed-lattice on-rate ratio in the simulation, robust tip tracking was reproduced, with an increase in EB1 tip localization for a higher ratio of GDP:GTP offrates (Figure 2I, red). Thus, both a hydrolysis-state dependent EB1 off-rate, as well as a rapid protofilament-edge EB1 on-rate, contribute to EB1 tip tracking in the model. However, the addition of rapid protofilament-edge on-rates increased the efficiency and robustness of simulated EB1 tip tracking.”

2) Second, to estimate the uncertainty of each EB1 dynamics model parameter, the effect of changing each parameter value on the success of simulated tip tracking was plotted over a broad range of values for each parameter (see Figure S2 and S3). These tables are referenced throughout the manuscript.

3) Finally, to ensure that the value used for each parameter in the simulation was similar to previously reported experimental values, we have now included updated parameter tables for both the tubulin assembly model, and the EB1 dynamics model. The EB1 dynamics parameter table (Table S2), which also includes a bond energy justification for GTP-tubulin protofilament-edge off-rates relative to GTP-tubulin closed-lattice off-rates. All of the parameter values used in the simulation are similar to previously reported values, with the exception of the off-rate from protofilament-edge sites, which has not been experimentally measured, but was constrained relative to the closed-lattice off-rate using bond energy arguments (See Table S2). Table S2 and associated manuscript text are as follows (from p. 6):

“All parameter values in the EB1 dynamics model were constrained by previously published experimental values (Maurer et al., 2011, 2014; Reid et al., 2019), with the exception of the off-rate from protofilament-edge sites, which has not been experimentally measured, but was constrained using bond energy arguments (See Table S2). To evaluate the uncertainty of each model parameter in impacting simulation results, the success of simulated tip tracking was plotted over a broad range of values for each parameter (see Figure S2 and S3).”

2. The model presented in the text and summarized in Figure 5F proposes how monomers of EB can track microtubule tips. However, there is a number of very convincing studies showing that monomers in fact cannot track microtubule tips. EB has to be a dimer to be able to recognize and track the tip. For example, if you dissociate dimers in real-time, they can no longer track microtubule tips (https://doi.org/10.1038/s41556-017-0028-5). It is confusing that authors first find parameters that would allow monomers to tip track and validate their simulations made for monomers using the experimental data, which should represent the behaviour of dimers. It makes validation arguably difficult. Before the model can be used to make predictions about where exactly EBs bind, it should be able to explain why EB monomers do not track microtubule tips and how EB dimers do. This seems like a big difference, so it is difficult to see if or how this more realistic model would lead to the same interpretation of the experimental data.

In the model, we have employed experimentally determined on and off rates for EB1 (See Table S2). Therefore, because the relevant experiments were performed using EB1 in its normal state as a dimer, the baseline simulations represent the simulation results for EB1 dimers. To determine how the model results would be impacted by including monomers in the model, rather than dimers, we turned to previous work, which has demonstrated that the EB1 monomer off-rates from microtubules are ~4-fold faster than the off-rates for dimers (Song et al., 2020) Thus, we increased all off-rates in the model by 4-fold from their baseline values, and thus ran “monomer” simulations. We found that EB1 tip tracking was decreased by ~3-fold in the monomer simulations (Figure S1G). These results are consistent with previous work, which demonstrates that EB1 monomers tip track less effectively than their dimer counterparts (Komarova et al., 2009; Skube et al., 2010). We have now clarified that the baseline simulation parameters represent EB1 dimers, and included a discussion of the monomer simulation results, as follows, p. 8-9:

“Previous work has demonstrated that EB1 monomers tip track less effectively than their dimer counterparts (Komarova et al., 2009; Skube et al., 2010). In the model, we employed experimentally determined on and off rates for EB1 (See Table S2). Therefore, because the relevant experiments were performed using EB1 in its normal state as a dimer, the baseline simulations represent the simulation results for EB1 dimers. To determine how the model results would be impacted by including monomers in the model, rather than dimers, we turned to previous work, which demonstrated that the EB1 monomer off-rates from microtubules were ~4-fold larger than the off-rates for dimers (Song et al., 2020). Thus, we increased all off-rates in the model by 4-fold from their baseline values, and thus ran “monomer” simulations. We found that EB1 tip tracking was decreased by ~3-fold in the monomer simulations (Figure S1G), consistent with previous reports (Komarova et al., 2009; Skube et al., 2010).”

3. The simulations that show that just the edge binding alone is sufficient to account for the profiles of microtubules observed in microscopy experiments need to be better explained. We do know that GTP caps can be long (e.g. https://doi.org/10.7554/eLife.51992) and in growing microtubules, there should be a lot more EBs sitting on the GTP lattice as compared to the number of EBs sitting on the edge simply because there are more closed-lattice sites regardless of how EB ends up there. Therefore, the shape of the experimental profile should have a much stronger contribution from the EBs sitting on the closed lattice as opposed to those sitting on the edge.

We apologize for the lack of clarity in our previous manuscript version. We have addressed this comment in four ways, as follows:

We agree that the reason the peak EB1 location is slightly back from the distal tip of the microtubule is indeed because there are more EBs sitting on the GTP lattice as compared to the number of EBs sitting on the edge. In the previous manuscript version, we demonstrated that the peak EB1 location matched published literature (Figure 1C). However, we have now explicitly reported the fraction of EB1 molecules that are bound to protofilament-edge sites, as compared to GTP-Tubulin closed-lattice sites, at any one time in the simulation (Figure S1H). Consistent with the idea that EB1 localization at the tip is primarily influenced by binding to closed-lattice GTP-tubulin sites, we find that there are ~2-fold more EB1 molecules bound to closed-lattice GTP-tubulin sites, than to GTP-tubulin protofilament-edge sites, during the tiptracking simulation. We have incorporated this new data into the manuscript, as follows (p. 20-21):

“As described above, our model predicts that rapid protofilamentedge binding increases the efficiency of EB1 tip tracking. However, in the simulation, the peak EB1 location is slightly distal from the tip of the growing microtubule (Figure 1C), similar to previous reports (Maurer et al., 2014). We surmised that the peak EB1 location in the simulation is heavily influenced by EB1 molecules that are stably bound to GTP-tubulin closed-lattice sites on the growing microtubule tip. Indeed, by reporting the fraction of EB1 molecules that are bound to GTP-Tubulin protofilament-edge sites as compared to GTP-Tubulin closed lattice sites, we found that there are ~2-fold more EB1 molecules bound to closed-lattice GTP-tubulin sites, as compared to protofilament-edge sites, at any one time in the simulation (Figure S1H).”

However, a key aspect of the simulation is that EB1 molecules arrive rapidly to protofilament-edge sites at the tip of the growing microtubule (~70-fold faster on-rate to protofilament-edge sites relative to closed-lattice sites). To understand what fraction of EB1 was initially incorporated into the microtubule via protofilament-edge sites, we have now run simulations in which we recorded the initial binding location of EB1, to determine the fraction of EB1 that initially bound to protofilamentedge sites as compared to closed-lattice positions (Figure S1I). We found that ~50% of all EB1 binding events occurred at protofilament-edge sites, with ~50% binding directly to closed-lattice sites (Figure S1I).

However, importantly, we found that the EB1 molecules that initially bound to protofilament-edge sites were heavily concentrated at the growing microtubule tip, while EB1 molecules that bound to closed-lattice sites were less concentrated near the tip of the growing microtubule (Figure S1J). This is because closed-lattice binding occurs throughout the microtubule, rather than specifically near to the growing microtubule plus-end, although the increased EB1 offrate from GDP-tubulin subunits relative to GTP-tubulin subunits still leads to an increased presence of EB1 near the growing microtubule tip. A description of the new Figure S1I,J has been added to the manuscript, p. 21, as follows:

“….To test this idea, we ran simulations in which we recorded the initial binding location of EB1, to determine the fraction of EB1 molecules that initially bound to protofilament-edges as compared to closed-lattice positions (Figure S1I). We found that ~50% of all EB1 binding events occurred at protofilament-edge sites, with ~50% binding directly to closed-lattice sites (Figure S1I). However, importantly, the EB1 molecules that initially bound to protofilament-edge sites were heavily concentrated at the growing microtubule tip, while EB1 molecules that bound to closed-lattice sites were more uniformly distributed throughout the microtubule (Figure S1J, S1A, and Viseo 2). This is because closed-lattice binding occurs throughout the microtubule, rather than specifically near to the growing microtubule plus-end, although the increased EB1 off-rate from GDP-tubulin subunits relative to GTP-tubulin subunits still leads to an increased presence of EB1 near the growing microtubule tip (Figure S1J).”

To explain this result, we propose that because the on-rate of new tubulin molecules is also rapid (arrival rate for tubulin ~85 s^-1^ at 10 μM tubulin), the simulated EB1 molecules that bind to protofilament-edge sites are quickly “locked in” to a closed-lattice GTP-tubulin binding configuration (see Figure 7). Thus, while most of the EB1 molecules bound to GTP-tubulin subunits are indeed bound to closed-lattice GTP-tubulin sites (Figure S1h), these EB1 molecules likely originated as arrivals to protofilamentedge sites (Figure S1h). To demonstrate this point, we have now included a simulation output cartoon (see Figure S1A) and a new video (Video 2), both of which show the origin of each bound EB1 molecule. Here, it is clear that the majority of EB1 molecules at the tip of the growing microtubule originally arrived to protofilament-edge sites, regardless of whether they are currently bound to GTP-tubulin closed lattice sites or to GTP-tubulin protofilament-edge sites (Figure S1a, green). We have incorporated this new data into the manuscript, as follows (p. 21):

“…a key aspect of the simulation is that EB1 molecules arrive rapidly to protofilament-edge sites at the tip of the growing microtubule. We propose that, because the on-rate of new tubulin molecules is also rapid (arrival rate for tubulin ~85 s^-1^ at 10 μM tubulin), the simulated EB1 molecules that bind to protofilament-edge sites are quickly “locked in” to a closed-lattice GTP-tubulin binding configuration (Figure 7). Thus, while most of the EB1 molecules at the microtubule tip are indeed bound to closed-lattice GTP-tubulin sites (Figure S1H), these EB1 molecules likely originated as arrivals to protofilament-edge sites (Figure S1A).”

As a clarification, our model does not assume that EB1 only binds to protofilament edge sites. While the on-rate constant to protofilament-edge sites is ~70-fold higher than for closed-lattice sites, as sites does not specifically enrich GTP-tubulin binding at the growing tip of the microtubule, we find that an increased on-rate to closed-lattice sites does not efficiently enrich EB1 specifically at the growing tip of the microtubule (Figure 2A). This clarification has been added to the manuscript, p. 10-11, as follows:

“To quantitatively dissect the relative role of closed-lattice binding on EB1 localization to growing microtubules, we ran simulations over a range of EB1 closed-lattice on-rates, while keeping all other EB1 on-rates and off-rates constant and set to their baseline values, including rapid EB1 protofilament-edge binding (Table S2). We found that a low EB1 closed-lattice on-rate led to a clear EB1 puncta at the tip of the microtubule (Figure 2C, left-bottom). Here, EB1 accumulation is dominated by protofilament-edge binding. However, increasing the EB1 closed-lattice on-rate by 32fold led to a ~1.6-fold increase in EB1 intensity at the microtubule tip, but, importantly, also led to a ~25-fold increase in EB1 intensity along the length of the microtubule (Figure 2C, center), even in the presence of EB1 protofilament-edge binding. By plotting the ratio of Tip:Lattice EB1 intensity (see Methods), we found that, with increasing EB1 closed-lattice on-rates, the EB1 intensity at the microtubule tip was decreased relative to the lattice (Figure 2C, right). Thus, the efficiency of simulated EB1 tip tracking was reduced with faster EB1 binding to closed-lattice sites, due to increased EB1 accumulation along the length of the microtubule.”

If this is true, why would simulations explain the data only assuming zero closed-lattice binding and not direct binding to the GTP cap? What about the opposite experiments?

We agree that the shape of the experimental profile has a much stronger contribution from the EBs sitting on the GTP-tubulin closed-lattice sites as opposed to those sitting on the edge (see above). Further, because there are a much larger quantity of available closed-lattice binding sites than there are protofilament-edge sites, we did not feel that simulations assuming zero closed-lattice binding would represent a physically relevant scenario. Therefore, to clarify, we did not perform simulations assuming zero closed-lattice binding.

However, we did perform simulations over a range of on-rates for protofilament-edge binding, assuming an EB1 closed-lattice on-rate at its baseline value (Figure 2b). In addition, we performed simulations in which the protofilament-edge binding was assumed to be zero, and so only direct binding to closed lattice sites was allowed (Figure 2a).

In the simulation, we assume that EB1 binding can distinguish structural features (edge vs lattice), due to diffusional steric hindrance to binding (Reid et al. 2019). However, the closed-lattice EB1 on-rate to GTP or GDP tubulin subunits within the microtubule was identical, and so EB1 could bind at any closed-lattice position along the microtubule, regardless of the hydrolysis state of the tubulin subunit. Thus, increasing the direct EB1 closed-lattice on-rate tended to increase binding of EB1 over the entire length of the microtubule, rather than enriching EB1 at the microtubule tip (Figure 2a).

It is very likely, that one could find a set of closed-lattice off-rates that would explain experimental data by assuming only direct binding to the closed lattice and no binding to the edge whatsoever. Can these explain the experimental results?

We have addressed this question in the following two ways:

1) In the simulation, the off-rate of EB1 molecules bound to closed-lattice sites depended on the hydrolysis state of the tubulin subunit to which they were bound. Specifically, the previously reported difference in affinity of EB1 for GTP-tubulin as compared to GDP-tubulin (Kd) leads to an experimental ratio of GDP:GTP off-rates equal to 9 (Maurer et al., 2011). We incorporated this ratio into the model via a ~6-12-fold ratio of GDP:GTP off-rates for all simulations (Figure R2h, center).

In addition, we have now run new simulations for a 6-12 fold ratio of GDP:GTP off-rates, but with the protofilament-edge on-rate equal to zero (Figure R2i, right, blue). We found that over the experimentally observed range of GDP:GTP off-rates, robust tip tracking was not observed in the absence of protofilament-edge binding (Figure R2i, right, blue). In contrast, robust EB1 tip tracking was observed over a wide range of GDP:GTP off-rates, when combined with a 50:1 protofilamentedge:closed-lattice on-rate ratio (Figure R2h, center). Following is the updated manuscript text regarding this analysis (p. 10):

“To quantitatively interrogate the model parameter sensitivity, we defined two key dimensionless variables that control tip tracking in the model. First, as described above, the ratio of the on-rate of EB1 to protofilament-edge sites relative to closed-lattice sites, which is independent of the hydrolysis state of the associated tubulin molecules, directly alters the EB1 tip tracking efficiency in the model (Figure 2G). Importantly, clear tip tracking was observed using the experimentally measured on-rate ratio for protofilament-edge sites relative to closed-lattice sites (50-100:1, (Reid et al., 2019)) (Figure 2G, image B, grey dashed boxes).

Second, as has been previously described, the ratio of the off-rate of EB1 from closed-lattice GDPtubulin sites, relative to closed-lattice GTP-tubulin sites, also influenced EB1 tip tracking in the model (Figure 2H; note that the model is comparatively insensitive to protofilament-edge off-rates, regardless of hydrolysis state Figure S2G-I, Figure S3D-F). Similar to the on-rate ratio, clear tip tracking was observed using the experimentally measured off-rate ratio for GDP-tubulin relative to GTP-tubulin (calculated as 6-12, based on K_D_ values reported in (Maurer et al., 2011)) (Figure 2H, image B, grey dashed boxes).

Finally, we evaluated the relative importance of the two dimensionless variables, one that dictates relative EB1 on-rates, and the other that dictates relative EB1 off-rates, in influencing simulated EB1 tip tracking (Figure 2I). We found that, in the absence of protofilament-edge binding, the experimentally observed range of closed-lattice GDP:GTP off-rate ratios did not robustly reproduce EB1 tip tracking (Figure 2I, top: representative images; bottom: blue bars). However, by including a 50:1 protofilament-edge to closed-lattice on-rate ratio in the simulation, robust tip tracking was reproduced, with an increase in EB1 tip localization for a higher ratio of GDP:GTP off-rates (Figure 2I, red). Thus, both a hydrolysis-state dependent EB1 off-rate, as well as a rapid protofilament-edge EB1 on-rate, contribute to EB1 tip tracking in the model. However, the addition of rapid protofilamentedge on-rates increased the efficiency and robustness of simulated EB1 tip tracking.”

2) While we ran a specific set of physically relevant parameter values to test whether we could find a set of closed-lattice off-rates that would explain experimental data by assuming only direct binding to the closed lattice and no binding to the protofilament-edge, we feel that a more important point regarding a protofilament-edge binding model was poorly described in the previous version of the manuscript. Here, it is important to emphasize that both rapid binding of EB1 to protofilamentedges (50:1 edge:lattice), and a high closed-lattice GDP:GTP tubulin off-rate ratio (9:1 GDP:GTP), are important for robust tip tracking in the model. Because both of these factors contribute to tip tracking, this leads to a very robust model, that does not require a narrow set of parameter values for either effect, in order to reproduce experimental results. This is clear in Figure R24 (above, copied from Figure 2G-I), in which tip tracking is evident over a large range of values for the protofilamentedge on-rates. Thus, differential GDP:GTP off-rates, together with protofilament-edge binding, robustly lead to EB1 tip tracking, without requiring narrow parameter sets or rules for a binding exclusion zone on the microtubule, as has been previously hypothesized. This point has been added to the Discussion section, p. 22, as follows:

“While protofilament-edge binding is a key aspect of our model, it is important to emphasize that both rapid binding of EB1 to protofilament-edges (50-100:1 edge:lattice), as well as a differential GDP- to GTP-tubulin off-rate (6-12:1 GDP:GTP), were important for robust tip tracking in the model. Because both of these factors contribute to tip tracking, this leads to a highly robust model, that does not require a narrow set of parameter values for either effect, in order to reproduce experimental results (Figure 2G-I). Thus, differential GDP- to GTP-tubulin off-rates, together with rapid protofilament-edge binding, robustly led to EB1 tip tracking, without requiring narrow parameter sets or an EB1 binding exclusion zone on the microtubule, as has been previously hypothesized.”

4. One prediction from only edge binding may be that microtubules growing in the presence of GTPgS should have very specific EB comets. Since incorporation at the edge is expected to be the same, the brightness at the tips should be the same as for GTP microtubules, but the comet should be significantly longer and tail off at a specific distance as the closed-lattice off rate should remain that of GTP. However, if it is only closed-lattice binding there should be no specific comet seen on GTPgS microtubules. Maybe the EB profile in these experiments can be used to extract exactly how much binding can be attributed to the lattice and how much to the edge?

GTPγS microtubules are somewhat of an enigma. Previous labs have performed EB binding experiments with GTPγS microtubules, but with widely varying results. For example, the Surrey lab found that Mal3 bound with ~3 fold higher affinity along the length of GTPγS microtubules, relative to the tip (Maurer et al., 2011). However, the Straube lab found that EB1,2, and 3 all bound to GTPγS with a 1.3-4fold lower affinity along the length of the microtubule, relative to the growing tips (Roth et al., 2019). Finally, in our previous work, we demonstrated that the lattice structure tended to be highly damaged in GTPγS microtubules, with large sections of open microtubules along the length of the microtubule (Figure 1, Reid et al., 2017). These open and damaged areas would be highly enriched in protofilamentedge sites, and so could facilitate rapid EB1 binding along the length of the lattice. Thus, due to the widely varying results using GTPγS microtubules, we were uncertain as to how to proceed with new experiments and simulations. We have therefore decided that this exploration may be more appropriately deferred for future work.

5. In growing microtubules majority of EBs are expected to be at the closed-lattice of the GTP cap simply because the number of these sites should be higher than the number of the edge sites. Let's say it is 10%, 50%, or 100% of EBs that sit on the closed-lattice are incorporated by the edge binding and the rest by direct GTP closed-lattice binding. Would that have an impact on the regulation of microtubule dynamic instability of other tip interactions? Are there any other potential implications?

In our simulation, ~50% of all microtubulebound EBs were initially incorporated through protofilament-edge binding (Figure S1i, left). However, analysis of growing microtubules in the simulation indicates that the majority of EBs that are near to the growing microtubule tip were initially incorporated by edge binding (Figure S1j, right). This result suggests that the relative number of protofilament-edge sites on the growing microtubule tip would strongly influence the targeting of EB proteins to the growing microtubule plus-end, and therefore proteins that are targeted to the plus-end via EB proteins. We now have added to following comments to the Discussion section, p. 23:

Recently published work provides support for the importance of EB1 protofilament-edge site binding in the efficiency of EB1 tip tracking. Specifically, by using the microtubule polymerase protein XMAP215 in cell-free experiments, the range of protofilament lengths between the leading and lagging protofilaments at the growing microtubule plus-end was increased (Farmer et al., 2021). Importantly, an increase in EB1-GFP intensity at the growing microtubule tip was observed with increasing XMAP215induced tip taper (Farmer et al., 2021). We note that increased tip taper would likely correspond to an increase in the number of protofilament-edge sites at the growing microtubule end, similar to our split comet phenotype (Figure 3). Thus, XMAP215 could increase the efficiency of EB1 tip tracking by adding new protofilament-edge sites to the growing microtubule plus-end. This suggests that EB1 recruitment, and by extension the recruitment of the +Tip Complex, could be sensitive to the number of protofilamentedge sites at the tip of the growing microtubule. Correspondingly, a recent report found that EB1, and thus CLASP2, is redistributed from the plus-end to the microtubule lattice in cells subjected to stretch and compression cycles (Li et al., 2023). This result is consistent with the idea that microtubule bending could cause openings and holes in the lattice, leading to the creation of new protofilament-edge sites along the lattice, which in turn causes a redistribution of EB1 from the plus-end tip to the lattice.

References

Atherton, J., Jiang, K., Stangier, M. M., Luo, Y., Hua, S., Houben, K., van Hooff, J. J. E., Joseph, A.-P., Scarabelli, G., Grant, B. J., Roberts, A. J., Topf, M., Steinmetz, M. O., Baldus, M., Moores, C. A., and Akhmanova, A. (2017). A structural model for microtubule minus-end recognition and protection by CAMSAP proteins. Nature Structural and Molecular Biology, 24(11), 931–943. https://doi.org/10.1038/nsmb.3483

Atherton, J., Stouffer, M., Francis, F., and Moores, C. A. (2018). Microtubule architecture in vitro and in cells revealed by cryo-electron tomography. Acta Crystallographica Section D Structural Biology, 74(6), 572–584. https://doi.org/10.1107/S2059798318001948

Cleary, J. M., and Hancock, W. O. (2021). Molecular mechanisms underlying microtubule growth dynamics. Current Biology, 31(10), R560–R573. https://doi.org/10.1016/j.cub.2021.02.035

Coombes, C. E., Yamamoto, A., Kenzie, M. R., Odde, D. J., and Gardner, M. K. (2013). Evolving Tip Structures Can Explain Age-Dependent Microtubule Catastrophe. Current Biology, 23(14), Article 14. https://doi.org/10.1016/j.cub.2013.05.059

Demchouk, A. O., Gardner, M. K., and Odde, D. J. (2011). Microtubule Tip Tracking and Tip Structures at the Nanometer Scale Using Digital Fluorescence Microscopy. Cellular and Molecular Bioengineering, 4(2), 192–204. https://doi.org/10.1007/s12195-010-0155-6

Doodhi, H., Prota, A. E., Rodríguez-García, R., Xiao, H., Custar, D. W., Bargsten, K., Katrukha, E. A., Hilbert, M., Hua, S., Jiang, K., Grigoriev, I., Yang, C.-P. H., Cox, D., Horwitz, S. B., Kapitein, L. C., Akhmanova, A., and Steinmetz, M. O. (2016). Termination of Protofilament Elongation by Eribulin Induces Lattice Defects that Promote Microtubule Catastrophes. Current Biology, 26(13), 1713– 1721. https://doi.org/10.1016/j.cub.2016.04.053

Farmer, V., Arpağ, G., Hall, S. L., and Zanic, M. (2021a). XMAP215 promotes microtubule catastrophe by disrupting the growing microtubule end. Journal of Cell Biology, 220(10), e202012144. https://doi.org/10.1083/jcb.202012144

Farmer, V., Arpağ, G., Hall, S. L., and Zanic, M. (2021b). XMAP215 promotes microtubule catastrophe by disrupting the growing microtubule end. Journal of Cell Biology, 220(10), e202012144. https://doi.org/10.1083/jcb.202012144

Gudimchuk, N. B., Ulyanov, E. V., O’Toole, E., Page, C. L., Vinogradov, D. S., Morgan, G., Li, G., Moore, J. K., Szczesna, E., Roll-Mecak, A., Ataullakhanov, F. I., and Richard McIntosh, J. (2020). Mechanisms of microtubule dynamics and force generation examined with computational modeling and electron cryotomography. Nature Communications, 11(1), 3765. https://doi.org/10.1038/s41467-020-17553-2

Guesdon, A., Bazile, F., Buey, R. M., Mohan, R., Monier, S., García, R. R., Angevin, M., Heichette, C., Wieneke, R., Tampé, R., Duchesne, L., Akhmanova, A., Steinmetz, M. O., and Chrétien, D. (2016). EB1 interacts with outwardly curved and straight regions of the microtubule lattice. Nature Cell Biology, 18(10), Article 10. https://doi.org/10.1038/ncb3412

Igaev, M., and Grubmüller, H. (2022). Bending-torsional elasticity and energetics of the plus-end microtubule tip. Proceedings of the National Academy of Sciences, 119(12), e2115516119. https://doi.org/10.1073/pnas.2115516119

Komarova, Y., De Groot, C. O., Grigoriev, I., Gouveia, S. M., Munteanu, E. L., Schober, J. M., Honnappa, S., Buey, R. M., Hoogenraad, C. C., Dogterom, M., Borisy, G. G., Steinmetz, M. O., and Akhmanova, A. (2009). Mammalian end binding proteins control persistent microtubule growth. Journal of Cell Biology, 184(5), 691–706. https://doi.org/10.1083/jcb.200807179

Margolin, G., Goodson, H. V., and Alber, M. S. (2011). Mean-field study of the role of lateral cracks in microtubule dynamics. Physical Review E, 83(4), Article 4. https://doi.org/10.1103/PhysRevE.83.041905

Margolin, G., Gregoretti, I. V., Cickovski, T. M., Li, C., Shi, W., Alber, M. S., and Goodson, H. V. (2012). The mechanisms of microtubule catastrophe and rescue: Implications from analysis of a dimer-scale computational model. Molecular Biology of the Cell, 23(4), Article 4. https://doi.org/10.1091/mbc.e11-08-0688

Maurer, S. P., Bieling, P., Cope, J., Hoenger, A., and Surrey, T. (2011). GTPγS microtubules mimic the growing microtubule end structure recognized by end-binding proteins (EBs). Proceedings of the National Academy of Sciences, 108(10), 3988–3993. https://doi.org/10.1073/pnas.1014758108

Maurer, S. P., Cade, N. I., Bohner, G., Gustafsson, N., Boutant, E., and Surrey, T. (2014). EB1 Accelerates Two Conformational Transitions Important for Microtubule Maturation and Dynamics. Current Biology, 24(4), 372–384. https://doi.org/10.1016/j.cub.2013.12.042

Maurer, S. P., Fourniol, F. J., Bohner, G., Moores, C. A., and Surrey, T. (2012). EBs Recognize a NucleotideDependent Structural Cap at Growing Microtubule Ends. Cell, 149(2), 371–382. https://doi.org/10.1016/j.cell.2012.02.049

Pecqueur, L., Duellberg, C., Dreier, B., Jiang, Q., Wang, C., Plückthun, A., Surrey, T., Gigant, B., and Knossow, M. (2012). A designed ankyrin repeat protein selected to bind to tubulin caps the microtubule plus end. Proceedings of the National Academy of Sciences, 109(30), 12011–12016. https://doi.org/10.1073/pnas.1204129109

Pettersen, E. F., Goddard, T. D., Huang, C. C., Meng, E. C., Couch, G. S., Croll, T. I., Morris, J. H., and Ferrin, T. E. (2021). UCSF CHIMERAX: Structure visualization for researchers, educators, and developers. Protein Science, 30(1), 70–82. https://doi.org/10.1002/pro.3943

Reid, T. A., Coombes, C., and Gardner, M. K. (2017). Manipulation and quantification of microtubule lattice integrity. Biology Open, bio.025320. https://doi.org/10.1242/bio.025320

Reid, T. A., Coombes, C., Mukherjee, S., Goldblum, R. R., White, K., Parmar, S., McClellan, M., Zanic, M., Courtemanche, N., and Gardner, M. K. (2019). Structural state recognition facilitates tip tracking of EB1 at growing microtubule ends. *ELife*, 8, e48117. https://doi.org/10.7554/*eLife*.48117

Roth, D., Fitton, B. P., Chmel, N. P., Wasiluk, N., and Straube, A. (2019). Spatial positioning of EB family proteins at microtubule tips involves distinct nucleotide-dependent binding properties. Journal of Cell Science, 132(4), jcs219550. https://doi.org/10.1242/jcs.219550

Skube, S. B., Chaverri, J. M., and Goodson, H. V. (2010). Effect of GFP tags on the localization of EB1 and EB1 fragments in vivo. Cytoskeleton, 67(1), 1–12. https://doi.org/10.1002/cm.20409

Song, Y., Zhang, Y., Pan, Y., He, J., Wang, Y., Chen, W., Guo, J., Deng, H., Xue, Y., Fang, X., and Liang, X. (2020). The dimeric organization that enhances the microtubule end-binding affinity of EB1 is susceptible to phosphorylation. Journal of Cell Science, jcs.241216. https://doi.org/10.1242/jcs.241216

VanBuren, V., Cassimeris, L., and Odde, D. J. (2005). Mechanochemical Model of Microtubule Structure and Self-Assembly Kinetics. Biophysical Journal, 89(5), 2911–2926. https://doi.org/10.1529/biophysj.105.060913

[Editors’ note: what follows is the authors’ response to the second round of review.]

The manuscript has been improved but there are some remaining issues that need to be addressed, as outlined below:Essential revisions:1) Provide a detailed analysis/simulation of the split Mal3 comets

As detailed below, we have now added simulation results for the split Mal3 comets, and provided a detailed analysis of simulation results relative to the experimental results.

Reviewer #3 (Recommendations for the authors):Gonzalez et al. employ an interdisciplinary approach to dissecting the molecular mechanism by which EB1 tracks the growing microtubule plus ends. In particular, the authors propose that the rapid binding to a special feature, the 'protofilament edge' and the differential binding affinity for the close lattice in GTP or GDP state facilitates efficient tip tracking activity of EB1 at growing microtubule ends. Solid experiment data support the computational simulation. As the authors have thoroughly addressed the reviewers' questions, I only have a few comments that might further improve the clarity.1. A more detailed analysis/simulation of the split Mal3 cometsThe split EB1 comets (Figure 3) are a good opportunity to test the 'protofilament edge-binding' model. The authors quantify the summed intensity of Mal3 and show an ~80% increase in the split comets, supporting additional protofilament-edge binding sites at the growing microtubules with split comets. However, as the split comets are usually quite well separated, it is counterintuitive that the continuously exposed 'protofilament edge' can cause the split comets. Is it possible to simulate the split comets? Also, it appears that the split comet in Figure 3A tracks the depolymerizing microtubules. Is it common? What is the possible explanation?

We thank the reviewer for this suggestion, and have now added new simulation results for split comets. Interestingly, the simulations also show separated EB1 puncta in the split comets (see Figure R2 below), perhaps due to stochastic incorporation of increased concentrations of EB1 at different locations along the growing microtubule tip. In these simulations, we adjusted the tubulin assembly parameters to allow for a longer tip taper length (≤ ~3 μm as compared to ≤ ~600 nm for the remainder of the paper simulations; see methods). Similar to experimental results, the simulated split comets demonstrated a higher total EB1 intensity relative to single comets within the same growth event (Figure R2, see panel C).

With regards to the experimental split comets tracking depolymerizing ends, this is not always the case. It may be that EB1 occasionally appears to track depolymerizing ends when there are split comets because remnants of EB1 remain behind on extremely tapered tips, providing the appearance of tiptracking as microtubules begin depolymerizing.

Following is the new text in the Results section on p. 13‐14:

“We first tested this prediction using our simulation. Thus, we asked whether there was an increase in the summed EB1‐GFP intensity at growing microtubule tips with split comets. To generate split comets in the simulation, we altered the microtubule assembly simulation parameters to allow for an increase in taper at the growing microtubule tips (from ≤ ~600 nm in our standard simulation, to ≤ ~3 μm in the split comet simulation (see methods)). By increasing the taper at the microtubule tip, the simulation was able to recapitulate split comets (Figure 3A, right (orange arrow: pre‐split; cyan arrows: post‐split)). We then asked whether there was an increase in the summed EB1‐GFP intensity on individual growing microtubule tips after an EB1 comet split, relative to prior to the split. Thus, we measured the total intensity of EB1‐GFP both before and after the comet split on individual simulated growing microtubules (Figure 3B, top: pre‐split; middle: post‐split). We subtracted the green background intensity both before and after the comet split (Figure 3B, bottom). We found that the split comets had a ~40% increase in the summed intensity of EB1‐GFP at the growing tip, relative to single comets on the same growth events (Figure 3C, paired t‐test, p<0.001). Therefore, consistent with our prediction, the simulation data indicates that an increase in protofilament‐edge sites on the sides of exposed protoflaments during split‐comet growth events leads to an increase in EB1 recruitment to the microtubule plus‐end.

Next, to test this prediction experimentally, we examined experimental microtubule growth events with split comets (Figure 3D, right; orange arrow: pre‐split; cyan arrows: post‐split). We measured the summed Mal3‐mCherry (yeast EB1‐homolog) intensity both before and after the comet split on individual growing microtubules (Figure 3E; top: pre‐split; middle: post‐split). We subtracted the green background intensity both before and after the comet split (Figure 3E, bottom). We found that split comets had an ~80% increase in the summed intensity of Mal3 at the growing microtubule tip relative to the single comets on the same microtubule growth events (Figure 3F, paired t‐test, p<<0.0001). Thus, the experimental results are consistent with the simulation results, and suggest that an increase in protofilament‐edge sites on the sides of exposed protofilaments during split‐comet growth events lead to an increase in EB1 recruitment to the microtubule plus‐end.”

2. The mechanism by which EB1 peak is behind the very tip of microtubules.As EB1 binds to the protofilament edge with a 5~7-fold higher affinity than to the close lattice, the location of the EB1 peak seems dependent on the protofilament density (either tapered or flared). Have the authors examined the EB1 tip tracking on microtubules with different end structures? For example, how would the EB1 comet look on microtubules with blunt but flared ends?

We have addressed this question in three different ways, as follows:

We agree that the location of the EB1 peak seems dependent on the protofilament‐edge density (either tapered or flared). As noted above, in further examining our simulation results, most of the EB1 arrivals are to the sides of the protofilaments, since the number of EB1 binding sites at the very tip of the microtubule is explicitly limited by the number of protofilaments in the microtubule (13 binding sites) (Figure 1‐ Figure supplement 1A). As the reviewer #2 notes, since protofilament‐edge sites on the sides of exposed protofilaments go deeper into the lattice, this effect may contribute to a peak EB1 location that is slightly distal from the tip of the growing microtubule. We have now updated our discussion comments on p. 21, as follows:

“…the number of EB1 binding sites at the tip of each protofilament is explicitly limited by the number of protofilaments in the microtubule (13 binding sites). Thus, EB1 binding to numerous protofilament‐edge sites along exposed protofilament sides that are distal from the tip of the microtubule may also contribute to the peak EB1 location.”

In regards to simulated EB1 tip tracking on microtubules with different end structures, the largest disruption to microtubule end structures was generated in the new “split comet” simulations (see above). Here, by substantially increasing the taper at the tip of the growing microtubule (≤ ~3 μm), the EB1 comet was greatly extended in length, and altered in configuration, thus shifting the location of EB1 binding. However, simulations in the remainder of the manuscript were limited to a tip taper of ≤ ~600 nm. To determine whether more subtle changes to the tip structure altered the location of the EB1 peak, we compared the peak EB1 position for simulated microtubules with tip tapers of ~200‐400 nm, to those with tip tapers of 400‐600 nm. We found that there was no significant difference in these two groups (Figure 1‐ Figure supplement 1B Right). Thus, while the protofilament‐edge density does alter the configuration and location of the EB1 comet for large changes in tip configuration (eg, for the split comet simulations), the location of the EB1 peak was robust to small changes in tip configuration. These comments have been added to the discussion, p. 21, as follows:

“Thus, EB1 binding to numerous protofilament‐edge sites along exposed protofilament sides that are distal from the tip of the microtubule may also contribute to the peak EB1 location. This idea is consistent with results from the “split comet” simulations (Figure 3A,B). Here, by substantially increasing the taper at the tip of the simulated growing microtubule (≤ ~3 μm), the EB1 comet was greatly extended in length, and altered in configuration, thus shifting the location of EB1 binding (Figure 3B). However, the location of the simulated EB1 peak position was insensitive to small changes in tip taper (Figure 1‐ Figure supplement 1B).”

Finally, the reviewer was curious as to how the EB1 comets looked on microtubules with blunt, but flared, ends. We examined the effect of flared ends on EB1 tip tracking by assuming that all protofilaments without lateral bonds were flared, and that these flared protofilaments had EB1 protofilament‐edge binding sites on their exposed protofilament sides. We found that simulated tip tracking was nearly identical for tapered tips (≤ 600 nm taper), or blunt tips with protofilament flaring (Figure 1—figure supplrment 1C, left and center). In addition, we introduced increased flaring into the simulation by reducing the lateral bond creation rate. We found that increased protofilament flaring at the tip led to very efficient tip tracking when the simulated EB1 was allowed to target flared protofilament‐edges (Figure 1—figure supplrment 1C, right, magenta), as compared to simulations in which EB1 did not bind to protofilament‐edges on flared protofilaments (Figure 1—figure supplrment 1C, right, blue). Thus, flared ends in the simulation behaved similarly to tapered tips, both in EB1 intensity and peak EB1 location. The associated manuscript text in regards to this analysis is as follows (p. 8)

“It has been previously suggested that growing microtubule plus‐ends could be “flared”, such that they have bent protofilaments that are curved (or flared) away from the central microtubule axis (McIntosh et al., 2018). Thus, we asked how a flared microtubule tip structure would affect tip tracking in our simulation. To approximate microtubule tip flaring in the model, we assumed that, with a flared end, all EB1 binding sites in front of the most distal lateral bond would be considered protofilament‐edge sites. We found that the microtubule flaring approximation in the simulation had no discernible effect on EB1 tip tracking (Figure 1‐ Figure supplement 1C, left/center). Further, we introduced increased tip flaring into the simulation by moving the most distal lateral bond farther away from the growing microtubule tip, which led to increased EB1 targeting to the flared growing microtubule plus‐end (Figure 1‐ Figure supplement 1C, right). Thus, flared microtubule tips in the simulation behaved similarly to tapered tips, both in EB1 intensity and in peak EB1 location.”

3. When I read the manuscript, I wondered how this current model could improve our understanding of the EB1 tip-tracking activity in the context of the model proposed by Maurer et al. 2014. From my point of view, the major conceptual advance is that the rapid binding to the 'protofilament edge' can explain the behaviors of EB1 at the growing microtubule ends without introducing an 'exclusion zone' as proposed in Maurer's model. The authors should compare Maurer's model earlier in the manuscript rather than later in the discussion.

In addition to the comparison of our new model to the Maurer 2014 model in the discussion near the end of the paper, we have now included additional references to the Maurer model in the Results section, as follows (p. 7):

“Importantly, our model with EB1 protofilament‐edge binding reproduced the peak EB1 position without requiring a predetermined EB1 “exclusion zone”, as has been previously hypothesized (Maurer et al., 2014). Rather, EB1 tip tracking in our current model depended solely on EB1 on/off rates and a growing microtubule plus‐end.”